

# Multi-band D-TRILEX approach to materials with strong electronic correlations

Matteo Vandelli[1,2,3,⋆], Josef Kaufmann[4], Mohammed El-Nabulsi[1], Viktor Harkov[1,5],
Alexander I. Lichtenstein[1,2,5] and Evgeny A. Stepanov[6,†]

**1** Institute of Theoretical Physics, University of Hamburg, Hamburg, Germany
**2** The Hamburg Centre for Ultrafast Imaging, Hamburg, Germany
**3** Max Planck Institute for the Structure and Dynamics of Matter,
Center for Free Electron Laser Science, Hamburg, Germany
**4** Institute of Solid State Physics, TU Wien, 1040 Vienna, Austria
**5** European X-Ray Free-Electron Laser Facility, Schenefeld, Germany
**6** CPHT, CNRS, Ecole Polytechnique, Institut Polytechnique de Paris,
F-91128 Palaiseau, France

⋆ matteo.vandelli94@gmail.com, † evgeny.stepanov@polytechnique.edu

## Abstract

We present the multi-band dual triply irreducible local expansion (D-TRILEX) approach to interacting electronic systems and discuss its numerical implementation. This method is designed for a self-consistent description of multi-orbital systems that can also have several atoms in the unit cell. The current implementation of the D-TRILEX approach is able to account for the frequency- and channel-dependent long-ranged electronic interactions. We show that our method is accurate when applied to small multi-band systems such as the Hubbard-Kanamori dimer. Calculations for the extended Hubbard, the two-orbital Hubbard-Kanamori, and the bilayer Hubbard models are also discussed.



# 1 Introduction

Understanding the effect of electronic correlations in materials is currently a very active topic of research. Strong interaction between electrons is responsible for their non-trivial collective behavior and for the formation of a variety of different states of matter, as for instance Mott insulating [1,2] and unconventional superconducting [3–5] phases. Usually, the low-energy physics of materials with strong electronic correlations is determined by a subspace of electronic bands that lie near the Fermi energy. In several situations, this correlated subspace can be effectively reduced to one band, which, for instance, is a standard approximation for cuprate superconductors [3, 6–9]. However, in most of the cases an accurate description of realistic materials with strong electronic correlations requires to take into account several bands that originate from different orbitals and/or atoms in the unit cell. Prominent examples of materials where the interplay of orbital degrees of freedom and strong correlations is believed to be of crucial importance are vanadates [10–13], ruthenates [14–20], nikelates [21–27], and iron-based superconductors [28–37]. Even in the case of cuprates the question whether an effective three-band model should be used instead of a single-band one is still under debate [38–47].

The need in addressing strong electronic correlations in a multi-band framework sets an outstanding challenge for theoretical material science. The *state-of-the-art* method for predicting and describing properties of correlated materials is dynamical mean field theory (DMFT) [48–50]. DMFT is particularly successful, because it allows for a non-perturbative description of local correlation effects by mapping the interacting system onto an effective local impurity problem. However, this approach is not able to account for non-local correlations, since it is based on the assumption of the locality of the electronic self-energy.

Two main routes have been proposed to go beyond the local picture provided by DMFT, while still taking advantage of the non-perturbative description of local electronic correlations. The first route consists in considering a finite cluster of lattice sites instead of a single-site impurity problem, which allows for taking into account spatial correlation effects within the cluster [50–56]. However, these methods are usually based on quantum Monte Carlo solvers that often suffer from a fermionic sign problem in multi-orbital calculations [57], or on the

exact diagonalization (ED) method [3,58], the complexity of which scales exponentially with the number of orbitals and sites. As the result, in the multi-orbital case the cluster methods are able to account for only short-range electronic correlations [59–62]. Diagrammatic extensions of DMFT provide the second way of taking into account the non-local correlation effects [63]. The key idea of these approaches is to use the DMFT impurity problem as a reference system for a diagrammatic expansion in order to describe the non-local electronic correlations in the form of the most relevant Feynman diagrams.

Particular examples of such theories are the $GW$+DMFT [64–70], the dual fermion (DF) [71–76], the dual boson (DB) [77–83], the dynamical vertex approximation (DΓA) [84–88], the triply irreducible local expansion (TRILEX) [89–92], and the dual TRILEX (D-TRILEX) [20, 93–97] methods. Among them, DF, DB, and DΓA have the most sophisticated diagrammatic structures that allow for a very accurate description of both, local and non-local correlation effects [98]. At the same time, these methods generally suffer from high computational costs that limit the application of these approaches in multi-band setups. For instance, the diagrammatic expansion in DF, DB, and DΓA involves the exact local four-point vertex function of the reference system. Evaluating this frequency-dependent object in a multi-band case is very expensive numerically, because it contains four external points that have independent frequency and band indices. Additionally, using the four-point vertex in a diagrammatic expansion is frequently hindered by the need of inverting the Bethe-Salpeter equation (BSE), which requires large computational resources both in terms of time for the calculation and of memory consumption for storing the full momentum and frequency dependent vertices. For this reason, among the three methods only the DΓA [86–88] and the second-order DF [99] approaches have been extended to the multi-band case so far.

On the contrary, $GW$+DMFT and TRILEX methods have a much simpler diagrammatic structure compared to DF, DB, and DΓA, which makes the former very attractive for multi-band calculations. Currently, $GW$+DMFT is intensively used in many realistic calculations [12, 64, 100–105]. However, among all non-local correlations this method considers only charge fluctuations and thus neglects important magnetic effects. Furthermore, $GW$+DMFT does not take into account vertex corrections that are crucial for an accurate description of magnetic, optical and transport properties [20, 105–113]. The TRILEX method partially cures these drawbacks by considering both, the charge and magnetic fluctuations, and also by introducing vertex corrections in the form of the local Hedin's vertex [114] of the DMFT impurity problem. The diagrammatic expansion based on the three-point vertices is still relatively simple, because it does not require to invert the BSE in momentum and frequency space. This is a clear computational advantage over DF, DB, and DΓA. However, TRILEX is affected by a double-counting (Fierz ambiguity [115–117]) problem when the charge and magnetic fluctuations are taken into account simultaneously [92]. Additionally, vertex corrections in TRILEX are included in diagrams in an asymmetric way. This diagrammatic structure leads to inconsistent results in a strong-coupling limit [118] and does not have a correct symmetry in the orbital [20] space.

In order to resolve the aforementioned issues of $GW$+DMFT and TRILEX, the D-TRILEX approach was recently developed [93, 94]. D-TRILEX has a very similar diagrammatic structure to TRILEX. However, it was derived following a rather different route, namely as an approximation of the DB theory. For this reason, D-TRILEX has the same degree of internal consistency as the DB approach. In particular, it treats both charge and magnetic fluctuations without facing the Fierz ambiguity issue, and also possesses a desired symmetry for the vertex corrections that is missing in the original formulation of the TRILEX approach. Initially, the D-TRILEX method was formulated and implemented in the single-orbital form [93]. In this context, it retains a high degree of accuracy compared with its parental DB theory [94]. Specifically, this approach captures both, the reduction of the critical value of the critical interaction for the Mott transition with respect to the DMFT prediction [93] and the pseudogap formation

in the Slater regime of a single-orbital Hubbard model [94]. Additionally, D-TRILEX correctly reproduces the momentum differentiation in the electronic self-energy predicted by an exact benchmark in a model relevant for cuprates [94]. These successes of the method achieved in a single-orbital context made it desirable to extend the D-TRILEX approach to a multi-band framework. An early attempt in this direction was the application of a simplified version of the method to the study of magnetic fluctuations in a three-orbital model for perovskites [20]. The promising results obtained in that case motivated us to derive a general formulation of the multi-band D-TRILEX theory and to develop a numerical implementation that does not rely on a specific model. The introduced approach provides a consistent formulation of a diagrammatic expansion on the basis of an arbitrary interacting reference problem. In particular, considering a finite cluster as the reference problem allows one to combine the diagrammatic and cluster ways of taking into account the non-local correlation effects within the multi-band D-TRILEX computational scheme.

In this work, we provide a detailed formulation of the D-TRILEX method in a multi-orbital and multi-site framework. We start by deriving a general multi-band fermion-boson action in Section 2. In Section 3, we show expressions for the self-energy and the polarization operator in D-TRILEX approximation, and related them to physical quantities. Section 4 contains the description of the computational workflow used in our implementation. There, we highlight several snippets that improve the convergence. In Section 5 we discuss several applications of our method that illustrate various capabilities of the developed multi-band D-TRILEX approach. Finally, Section 6 is devoted to conclusions.

## 2 Effective fermion-boson action in the multi-band framework

We start with a general action of a multi-band extended Hubbard model

$$
\begin{aligned}
\mathcal{S} = & -\sum_{\substack{k,\{l\},\\ \sigma\sigma'}} c^*_{k\sigma l}\left[(i\nu+\mu)\delta_{\sigma\sigma'}\delta_{ll'}-\varepsilon^{\sigma\sigma'}_{\mathbf{k},ll'}\right]c_{k\sigma'l'} + \frac{1}{2}\sum_{\substack{q,\{l\},\\ \{k\},\{\sigma\}}} U^{pp}_{l_1l_2l_3l_4}c^*_{k\sigma l_1}c^*_{q-k,\sigma'l_2}c_{q-k',\sigma'l_4}c_{k'\sigma l_3} \\
& + \frac{1}{2}\sum_{\substack{q,\{l\},\\ \varsigma=d,m}} V^{\varsigma}_{q,l_1l_2,l_3l_4}\rho^{\varsigma}_{-q,l_1l_2}\rho^{\varsigma}_{q,l_4l_3} + \sum_{\substack{q,\{l\},\\ \vartheta=s,t}} V^{\vartheta}_{q,l_1l_2,l_3l_4}\rho^{*\vartheta}_{q,l_1l_2}\rho^{\vartheta}_{q,l_3l_4}.
\end{aligned}
\tag{1}
$$

In this expression, $c^{(*)}_{k\sigma l}$ is the Grassmann variable that describes the annihilation (creation) of an electron with momentum $\mathbf{k}$, fermionic Matsubara frequency $\nu$, and spin projection $\sigma \in \{\uparrow,\downarrow\}$. The label $l$ numerates the orbital and the site within the unit cell. To simplify notations, we use a combined index $k \in \{\mathbf{k},\nu\}$. Summations over momenta and frequencies are defined as:

$$
\sum_k = \frac{1}{\beta}\sum_\nu\frac{1}{N_k}\sum_{\mathbf{k}},
\tag{2}
$$

where $\beta = T^{-1}$ is the inverse temperature and $N_k$ is the number of $\mathbf{k}$-points in the discretized Brillouin zone (BZ). The single-particle part of the lattice action (first term in Eq. (1)) contains the chemical potential $\mu$ and the single-particle Hamiltonian term $\varepsilon^{\sigma\sigma'}_{\mathbf{k},ll'}$ that has the following structure in the spin space: $\varepsilon^{\sigma\sigma'}_{\mathbf{k},ll'} = \varepsilon_{\mathbf{k},ll'}\delta_{\sigma\sigma'} + i\vec{\gamma}_{\mathbf{k},ll'}\cdot\vec{\sigma}_{\sigma\sigma'}$. The diagonal part in the spin space $\varepsilon_{\mathbf{k},ll'}$ of this matrix contains the momentum- and orbital-space representation of the hopping amplitudes between different lattice sites, and may also account for the effect of the crystal field splitting (CFS) and of the external electric field. The non-diagonal contribution in spin space $\vec{\gamma}_{\mathbf{k},ll'}$ describes the effect of the external magnetic field and the spin-orbit coupling

(SOC), that is usually expressed in the Rashba form [119]. The latter corresponds to a Fourier transform of the effective spin-dependent hopping amplitudes [120]. $\vec{\sigma} = \{\sigma^x, \sigma^y, \sigma^z\}$ is a vector of Pauli matrices.

The on-site Coulomb potential is written in the conventional (particle-particle) form

$$U^{pp}_{l_1 l_2 l_3 l_4} = \int dr dr' \psi^*_{l_1}(r) \psi^*_{l_2}(r') V(r-r') \psi_{l_3}(r) \psi_{l_4}(r'), \tag{3}$$

where $V(r-r')$ is the screened Coulomb interaction and $\psi_l(r)$ are localized on-site basis functions. The local interaction can also be rewritten in the particle-hole representation using the following relation $U^{ph}_{l_1 l_2 l_3 l_4} = U^{pp}_{l_1 l_4 l_2 l_3}$ (see Appendix B). The remaining part of the interaction $V^r_q$ in Eq. (1) is written in the channel representation $r \in \{\varsigma, \vartheta\}$, where $\varsigma \in \{d, m\}$ denotes charge ($d$) and magnetic ($m \in \{x, y, z\}$) channels, and $\vartheta \in \{s, t\}$ depicts singlet ($s$) and triplet ($t$) channels. This interaction can have an arbitrary momentum $\mathbf{q}$ and bosonic Matsubara frequency $\omega$ dependence as depicted by a combined index $q \in \{\mathbf{q}, \omega\}$. Usually, $V^r_q$ corresponds to the non-local interaction. However, it may also contain the frequency-dependent part of the local interaction that is not included in the $U_{l_1 l_2 l_3 l_4}$ term. Composite fermionic variables $\rho^r_{q, l_1 l_2}$ for the considered bosonic channels describe fluctuations of corresponding densities around their average values $\rho^r_{q, l_1 l_2} = n^r_{q, l_1 l_2} - \langle n^r_{q, l_1 l_2} \rangle$. The orbital-dependent charge and magnetic densities can be introduced as follows

$$n^d_{q, l_1 l_2} = \sum_{k, \sigma} c^*_{k+q, \sigma l_1} c_{k \sigma l_2}, \qquad \vec{n}^m_{q, l_1 l_2} = \sum_{k, \{\sigma\}} c^*_{k+q, \sigma l_1} \vec{\sigma}_{\sigma \sigma'} c_{k \sigma' l_2}. \tag{4}$$

Densities for the particle-particle channel $n^{(*)\vartheta}_{q, l_1 l_2}$ are defined in Appendix A.1.

In this work, the initial lattice problem (1) is addressed within the dual triply irreducible local expansion (D-TRILEX) formalism [93, 94]. The D-TRILEX approach allows one to construct a self-consistent diagrammatic expansion on the basis of a generic reference system. The reference system is introduced to account for some (usually local or short-range) part of electronic correlations numerically exactly, and its particular form depends on the considered lattice problem [76]. For instance, in DMFT-based calculations the reference system corresponds to a single-impurity [48], several isolated impurities [121, 122], or a finite cluster [50–56] problems. In the Hubbard-I approximation, the DMFT local impurity problem can be reduced to an atomic problem [123–125]. It is also possible to build the D-TRILEX diagrammatic expansion on the basis of the impurity problem of the extended dynamical mean field theory (EDMFT) [126–130] by introducing a bosonic hybridization function (see Appendix A.1). The latter accounts for the effect of the non-local interaction on the local electronic correlations and could play an important role when the non-local interactions are strong. Alternatively, in the spirit of the cluster perturbation theory, one can consider a finite plaquette as a reference system [76, 131]. The limit of an infinite plaquette as a reference system corresponds to the exact solution of the problem. For this reason, we expect the accuracy of the D-TRILEX method to improve with enlarging the cluster similarly to what has been shown for the TRILEX approach [92]. Indeed, as the spatial size of the reference problem is increased, the range of electronic correlations that are treated within the exactly-solved cluster reference problem is also increased. Additionally, using a cluster reference system allows for the study of broken symmetry phases. In this regard, instead of viewing the cluster methods and the multi-band D-TRILEX theory as competing approaches, one could consider D-TRILEX as a method to improve the cluster solution of the problem by diagrammatically adding long-range correlations that are not captured by a finite cluster when the computational costs prevent a further increase of the cluster's size.

As has been mentioned in Introduction, the D-TRILEX method was originally developed for a single-band case. In order to extend this formalism to multi-band systems, we follow

the derivation presented in Refs. [93, 94] and introduce an effective partially bosonized dual action written in terms of fermion $f$ and boson $b$ variables (see Appendix A.1 for details)

$$
\mathcal{S}_{fb} = -\sum_{k,\{l\}}\sum_{\sigma\sigma'} f^*_{k\sigma l}\big[\tilde{\mathcal{G}}_k^{-1}\big]^{\sigma\sigma'}_{ll'} f_{k\sigma'l'} - \frac{1}{2}\sum_{q,\{l\}}\sum_{\varsigma\varsigma'} b^\varsigma_{-q,l_1 l_2}\big[\tilde{\mathcal{W}}_q^{-1}\big]^{\varsigma\varsigma'}_{l_1 l_2, l_3 l_4} b^{\varsigma'}_{q,l_4 l_3}
$$
$$
-\sum_{q,\{l\}}\sum_{\vartheta\vartheta'} b^{*\vartheta}_{q,l_1 l_2}\big[\tilde{\mathcal{W}}_q^{-1}\big]^{\vartheta\vartheta'}_{l_1 l_2, l_3 l_4} b^{\vartheta'}_{q,l_3 l_4} + \mathcal{F}[f,b]. \tag{5}
$$

It is important to emphasize that this action describes only those correlation effects that are not taken into account by the reference problem. The bare dual fermionic Green's function has the following form

$$
\tilde{\mathcal{G}}^{\sigma_1\sigma_2}_{k,l_1 l_2} = \sum_{\{l'\},\{\sigma'\}} B^{\sigma_1\sigma_1'}_{\nu,l_1 l_1'}\Big[\big((\varepsilon_{\mathbf{k}}-\Delta_\nu)^{-1}-g_\nu\big)^{-1}\Big]^{\sigma_1'\sigma_2'}_{l_1'l_2'} B^{\sigma_2'\sigma_2}_{\nu,l_2'l_2}, \tag{6}
$$

where $\Delta^{\sigma\sigma'}_{\nu,ll'}$ and $g^{\sigma\sigma'}_{\nu,ll'} = -\langle c_{\nu\sigma l} c^*_{\nu\sigma'l'}\rangle$ are respectively the fermionic hybridization and the Green's function of the reference system. In this expression, the scaling factors $B^{\sigma\sigma'}_{\nu,ll'}$ appear as a consequence of a certain freedom in the Hubbard-Stratonovich transformation of the initial action (1) and can be chosen arbitrarily (see Appendix A.1). In the original formulation of the D-TRILEX theory [93, 94] and other dual methods [71–73, 77–80, 82, 83] the choice $B^{\sigma\sigma'}_{\nu,ll'} = g^{\sigma\sigma'}_{\nu,ll'}$ ensures that the interaction of the effective dual action corresponds to vertex functions of the reference system. However, working with these vertices is not very convenient, because they have a numerical noise at large frequencies and also delta-functions appearing in the imaginary-time space [83]. To avoid these problems, in the multi-orbital D-TRILEX implementation we exclude these scaling factors by setting $B^{\sigma\sigma'}_{\nu,ll'} = \delta_{ll'}\delta_{\sigma\sigma'}$. This simplifies the expression (6) for the bare dual fermionic Green's function to

$$
\tilde{\mathcal{G}}^{\sigma\sigma'}_{k,ll'} = \Big[\big((\varepsilon_{\mathbf{k}}-\Delta_\nu)^{-1}-g_\nu\big)^{-1}\Big]^{\sigma\sigma'}_{ll'}. \tag{7}
$$

Note that within the convention chosen here, the dimension of the dual Green's function (7) does not correspond to [1/Energy] dimension of a physical Green's function.

The bosonic propagator (renormalized interaction) of the partially bosonized dual action (5) is the following (see Appendix A.1)

$$
\tilde{\mathcal{W}}^{rr'}_{q,l_1 l_2, l_3 l_4} = \mathcal{W}^{rr'}_{q,l_1 l_2, l_3 l_4} - \bar{u}^r_{l_1 l_2, l_3 l_4}\delta_{rr'}. \tag{8}
$$

Here, $\bar{u}^\varsigma_{l_1 l_2, l_3 l_4} = \frac{1}{2} U^\varsigma_{l_1 l_2, l_3 l_4}$ and $\bar{u}^\vartheta_{l_1 l_2, l_3 l_4} = U^\vartheta_{l_1 l_2, l_3 l_4}$ are the corrections that prevent the double counting of the interaction between different channels (see Appendix B). The renormalized interaction $\mathcal{W}^{rr'}_q$ of extended DMFT [126–130] can be obtained from the corresponding Dyson equation

$$
\big[\mathcal{W}_q^{-1}\big]^{rr'}_{l_1 l_2, l_3 l_4} = \Big[\big(U^r + V^r_q\big)^{-1}\Big]_{l_1 l_2, l_3 l_4}\delta_{rr'} - \Pi^{\text{imp}\,rr'}_{\omega,l_1 l_2, l_3 l_4}, \tag{9}
$$

that involves the polarization operator $\Pi^{\text{imp}\,rr'}_{\omega,l_1 l_2, l_3 l_4}$ of the reference (impurity) problem and the bare interaction in the channel representation (see Appendix B)

$$
U^d_{l_1 l_2 l_3 l_4} = \frac{1}{2}\Big(2U^{ph}_{l_1 l_2 l_3 l_4} - U^{ph}_{l_1 l_3 l_2 l_4}\Big) = \frac{1}{2}\Big(2U^{pp}_{l_1 l_4 l_2 l_3} - U^{pp}_{l_1 l_4 l_3 l_2}\Big), \tag{10}
$$

$$
U^m_{l_1 l_2 l_3 l_4} = -\frac{1}{2}U^{ph}_{l_1 l_3 l_2 l_4} = -\frac{1}{2}U^{pp}_{l_1 l_4 l_3 l_2}, \tag{11}
$$

$$
U^s_{l_1 l_2 l_3 l_4} = \frac{1}{2}\Big(U^{ph}_{l_1 l_3 l_4 l_2} + U^{ph}_{l_1 l_4 l_3 l_2}\Big) = \frac{1}{2}\Big(U^{pp}_{l_1 l_2 l_3 l_4} + U^{pp}_{l_1 l_2 l_4 l_3}\Big), \tag{12}
$$

$$
U^t_{l_1 l_2 l_3 l_4} = \frac{1}{2}\Big(U^{ph}_{l_1 l_3 l_4 l_2} - U^{ph}_{l_1 l_4 l_3 l_2}\Big) = \frac{1}{2}\Big(U^{pp}_{l_1 l_2 l_3 l_4} - U^{pp}_{l_1 l_2 l_4 l_3}\Big). \tag{13}
$$

The interacting term of the effective action (5) (see Appendix A.1)

$$
\begin{aligned}
\mathcal{F}[f,b] = \sum_{q,\{k\}} \sum_{\{v\},\{\sigma\}} \sum_{\{l\},\varsigma/\vartheta} \Bigg\{ &\Lambda^{\sigma\sigma'\varsigma}_{v\omega,l_1,l_2,l_3l_4} f^*_{k\sigma l_1} f_{k+q,\sigma',l_2} b^{\varsigma}_{q,l_4l_3} \\
&+ \frac{1}{2} \left( \Lambda^{\sigma\sigma'\vartheta}_{v\omega,l_1,l_2,l_3l_4} f^*_{k\sigma l_1} f^*_{q-k,\sigma',l_2} b^{\vartheta}_{q,l_3l_4} + \Lambda^{*\sigma\sigma'\vartheta}_{v\omega,l_1,l_2,l_3l_4} b^{*\vartheta}_{q,l_3l_4} f_{q-k,\sigma',l_2} f_{k\sigma l_1} \right) \Bigg\}
\end{aligned}
\tag{14}
$$

contains only the momentum-independent three-point interaction vertex function $\Lambda^{(*)}_{v\omega}$ of the reference system. The explicit expression of $\Lambda^{(*)}_{v\omega}$ can be found in Eqs. (51)–(53). The four-point (fermion-fermion) vertex function (92) is eliminated from the theory by using a partially bosonized approximation for the interaction [93, 94, 132–134].

## 3 D-TRILEX approach

The introduced effective fermion-boson action (5) allows for the calculation of observable quantities by using diagrammatic techniques. This goal can be achieved either by performing approximated diagrammatic expansions or by applying exact numerical methods such as the diagrammatic Monte Carlo (DiagMC) scheme [94]. In this section, we discuss the simplest diagrammatic approximation, namely the D-TRILEX method, that represents a feasible approach for actual calculations in the multi-band case. From here on, we neglect the non-local fluctuations in the particle-particle ($\vartheta$) channel due to their small contribution to the D-TRILEX diagrammatic structure [94].

### 3.1 Diagrammatic expansion in the dual space

To be consistent with applications discussed in Section 5, in the main text of the paper we restrict ourselves to a paramagnetic regime and do not consider the spin-orbit coupling. A general (spin-dependent) form of the D-TRILEX equations is shown in Appendix A.3. In the paramagnetic case all single-particle quantities are diagonal in the spin space and do not depend on the spin projection. For instance, the bare dual Green's function (7) becomes $\tilde{\mathcal{G}}^{\sigma\sigma'}_{k,ll'} = \tilde{\mathcal{G}}_{k,ll'} \delta_{\sigma\sigma'}$. Consequently, the two-particle quantities are diagonal in the channel indices, as for example holds true for the bare bosonic propagator (8) $\tilde{\mathcal{W}}^{\varsigma\varsigma'}_q = \tilde{\mathcal{W}}^{\varsigma}_q \delta_{\varsigma\varsigma'}$. One can also introduce spin-independent three-point vertex functions for the charge $\Lambda^d_{v\omega} = \Lambda^{\uparrow\uparrow d}_{v\omega}$ and magnetic $\Lambda^{m=x,y,z}_{v\omega} = \Lambda^{\uparrow\uparrow z}_{v\omega}$ channels (see e.g. Refs. [83, 94]). The dressed Green's function $\tilde{G}_k$ and the renormalized interaction $\tilde{W}^{\varsigma}_q$ of the effective partially bosonized dual problem (5) can be found via Dyson equations

$$
\left[ \tilde{G}^{-1}_k \right]_{ll'} = \left[ \tilde{\mathcal{G}}^{-1}_k \right]_{ll'} - \tilde{\Sigma}_{k,ll'},
\tag{15}
$$

$$
\left[ \left( \tilde{W}^{\varsigma}_q \right)^{-1} \right]_{l_1l_2,l_3l_4} = \left[ \left( \tilde{\mathcal{W}}^{\varsigma}_q \right)^{-1} \right]_{l_1l_2,l_3l_4} - \tilde{\Pi}^{\varsigma}_{q,l_1l_2,l_3l_4}.
\tag{16}
$$

The dual self-energy $\tilde{\Sigma}$ in the D-TRILEX approximation consists of the tadpole and $GW$-like diagrams $\tilde{\Sigma}_k = \tilde{\Sigma}^{TP}_k + \tilde{\Sigma}^{GW}_k$. The explicit expressions for these contributions are

$$
\left[ \tilde{\Sigma}^{TP}_v \right]_{l_1l_7} = 2 \sum_{k',\{l\}} \left[ \Lambda^d_{v,\omega=0} \right]_{l_1,l_7,l_3l_4} \left[ \tilde{\mathcal{W}}^d_{q=0} \right]_{l_3l_4,l_5l_6} \left[ \Lambda^d_{v',\omega=0} \right]_{l_8,l_2,l_6l_5} \left[ \tilde{G}_{k'} \right]_{l_2l_8},
\tag{17}
$$

$$
\left[ \tilde{\Sigma}^{GW}_k \right]_{l_1l_7} = - \sum_{q,\{l\},\varsigma} \left[ \Lambda^{\varsigma}_{v\omega} \right]_{l_1,l_2,l_3l_4} \left[ \tilde{G}_{k+q} \right]_{l_2l_8} \left[ \tilde{W}^{\varsigma}_q \right]_{l_3l_4,l_5l_6} \left[ \Lambda^{\varsigma}_{v+\omega,-\omega} \right]_{l_8,l_7,l_6l_5}.
\tag{18}
$$

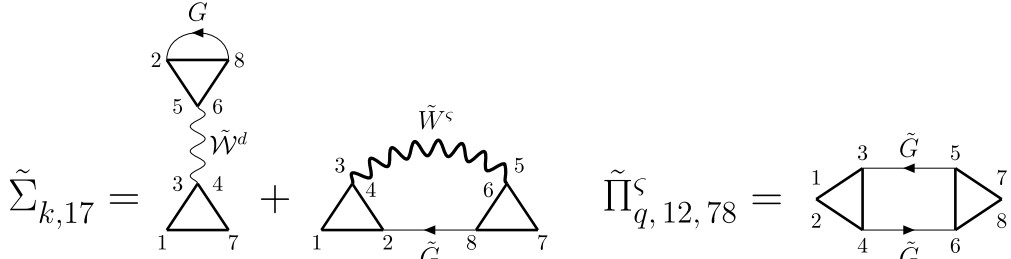

Figure 1: Diagrammatic representation for the dual self-energy $\tilde{\Sigma}$ (left) and the dual polarization operator $\tilde{\Pi}$ (right). The two processes contributing to $\tilde{\Sigma}$ are expressed mathematically in Eqs. (17) and (18). The expression for $\tilde{\Pi}$ is explicitly written in Eq. (19). Wavy lines represent dual bosonic propagators and straight lines depict dual Green's function, as explicitly indicated in the Figure. Triangles represent three-point vertex functions $\Lambda^\varsigma_{\nu\omega}$. Numbers correspond to band indices. The bosonic end of the triangle can be identified by the fact that it carries two band indices.

The polarization operator $\tilde{\Pi}$ of the D-TRILEX approach is following

$$\left[\tilde{\Pi}^{\varsigma}_q\right]_{l_1 l_2, l_7 l_8} = 2 \sum_{k,\{l\}} \left[\Lambda^{\varsigma}_{\nu+\omega,-\omega}\right]_{l_4,l_3,l_2 l_1} \left[\tilde{G}_k\right]_{l_3 l_5} \left[\tilde{G}_{k+q}\right]_{l_6 l_4} \left[\Lambda^{\varsigma}_{\nu\omega}\right]_{l_5,l_6,l_7 l_8} . \tag{19}$$

Note that the D-TRILEX diagrams (17)-(19) represent the leading contribution to the self-energy and the polarization operator of the partially bosonized dual action (5) in both the weak and the strong coupling limits independently from the dimensionality of the problem. Indeed, at weak coupling the D-TRILEX diagrammatic expansion is a perturbative expansion in terms of the renormalised interaction (8). On the other hand, in the strong coupling limit the small parameter of the diagrammatic expansion is the bare dual Green's function (7), which is purely non-local. The diagrams for $\tilde{\Sigma}$ and $\tilde{\Pi}$ are also shown in Fig. 1. These expressions illustrate that in the D-TRILEX approach the single- and two-particle quantities are treated self-consistently, which allows one to account for the effect of collective electronic fluctuations onto the electronic spectral function and *vice versa* [94–97].

## 3.2 Relation between physical and dual quantities

The D-TRILEX diagrammatic expansion introduced in Section 3.1 is performed in the dual space (5) that describes electronic correlations beyond the ones of the reference system. This formulation of the theory allows one to avoid double-counting of correlation effect that are already taken into account by the reference problem. The single- and two-particle quantities for the initial lattice problem (1) can be obtained from the dual quantities using the following exact relations (see Appendix A.2). The most convenient relation for the practical calculation of the lattice Green's function $G_{k,ll'}$ involves the dual self-energy $\tilde{\Sigma}$

$$\left[G_k^{-1}\right]_{ll'} = \left[\left(g_\nu + \tilde{\Sigma}_k\right)^{-1}\right]_{ll'} + \Delta_{\nu,ll'} - \varepsilon_{\mathbf{k},ll'} . \tag{20}$$

An alternative way to get $G_{k,ll'}$ requires calculating the lattice self-energy

$$\Sigma_{k,ll'} = \Sigma^{\text{imp}}_{\nu,ll'} + \sum_{l_1} \tilde{\Sigma}_{k,ll_1} \left[\left(\mathbb{1} + g_\nu \cdot \tilde{\Sigma}_k\right)^{-1}\right]_{l_1 l'} . \tag{21}$$

The lattice Green's function can then be obtained from the standard Dyson equation for the initial lattice action (1)

$$\left[G_k^{-1}\right]_{ll'} = (i\nu + \mu)\delta_{ll'} - \varepsilon_{\mathbf{k},ll'} - \Sigma_{k,ll'} . \tag{22}$$

Calculating $G_{k,ll'}$ be means of Eq. (20) has an advantage, because this expression does not involve the self-energy of the reference problem $\Sigma^{\text{imp}}_{\nu,ll'}$. The latter is not a correlation function and is usually calculated by inverting the corresponding Dyson equation for the Green's function $g_{\nu,ll'}$ of the reference problem

$$\Sigma^{\text{imp}}_{\nu,ll'} = (i\nu + \mu)\delta_{ll'} - \Delta_{\nu,ll'} - \left[g_\nu^{-1}\right]_{ll'}. \tag{23}$$

Consequently, $\Sigma^{\text{imp}}_{\nu,ll'}$ obtained in this way contains big numerical noise at large frequencies $\nu$. This problem can be cured by employing improved estimator methods that consist in computing higher-order correlation functions [135–137]. However, in multi-band calculations this procedure is numerically expensive. For this reason, it is preferable to compute the lattice Green's function using Eq. (20), and the lattice self-energy (21) separately.

Contrary to the self-energy, the polarization operator of the reference system $\Pi^{\text{imp}\varsigma}_\omega$ is not strongly affected by noise at large frequencies, because it has the same dimension as the susceptibility of the reference system $\chi^\varsigma_{\omega,l_1l_2,l_3l_4} = -\langle\rho^\varsigma_{\omega,l_2l_1}\rho^\varsigma_{-\omega,l_3l_4}\rangle$. Indeed, the polarization operator is defined through the corresponding Dyson equation as

$$\left[\left(\Pi^{\text{imp}\varsigma}_\omega\right)^{-1}\right]_{l_1l_2,l_3l_4} = \left[\left(\chi^\varsigma_\omega\right)^{-1}\right]_{l_1l_2,l_3l_4} + U^\varsigma_{l_1l_2,l_3l_4}. \tag{24}$$

For this reason, the lattice susceptibility $X^\varsigma_q$ is convenient to obtain directly from the Dyson equation (see Appendix A.2)

$$\left[\left(X^\varsigma_q\right)^{-1}\right]_{l_1l_2,l_3l_4} = \left[\left(\Pi^\varsigma_q\right)^{-1}\right]_{l_1l_2,l_3l_4} - \left[U^\varsigma + V^\varsigma_q\right]_{l_1l_2,l_3l_4}, \tag{25}$$

that involves the polarization operator of the lattice problem

$$\Pi^\varsigma_{q,l_1l_2,l_3l_4} = \Pi^{\text{imp}\varsigma}_{\omega,l_1l_2,l_3l_4} + \sum_{l',l''}\tilde{\Pi}^\varsigma_{q,l_1l_2,l'l''}\left[\left(\mathbb{1} + \bar{u}^\varsigma \cdot \tilde{\Pi}^\varsigma_q\right)^{-1}\right]_{l'l'',l_3l_4}. \tag{26}$$

Importantly, as shown in Appendix A.2, the divergence in the lattice susceptibility $X^\varsigma_q$ occurs at the same time as in the renormalized interaction $\tilde{W}^\varsigma_q$ that enters the self-energy (18). This allows the D-TRILEX approach to capture the formation of the pseudogap in the electronic spectral function in the paramagnetic regime in the vicinity of a symmetry broken phase in the system [95, 97].

Finally, the polarization operator in the D-TRILEX approach (19) has the same structure as the exchange interaction $\mathcal{J}^\varsigma$ between charge and/or magnetic densities derived in Refs. [132–134] in the many-body framework. This fact allows for a direct calculation of the exchange interaction within the D-TRILEX scheme using the following relation

$$\mathcal{J}^\varsigma_{q,l_1l_2,l_3l_4} = \sum_{\{l'\}}\left[\left(\Pi^{\text{imp}\varsigma}_\omega\right)^{-1}\right]_{l_1l_2,l'_1l'_2}\left[\tilde{\Pi}^\varsigma_q\right]_{l'_1l'_2,l'_3l'_4}\left[\left(\Pi^{\text{imp}\varsigma}_\omega\right)^{-1}\right]_{l'_3l'_4,l_3l_4}. \tag{27}$$

This quantity is computed at the first iteration of the D-TRILEX self-consistent cycle, because instead of the dressed dual Green's functions $\tilde{G}$ the dual polarization operator (19) in the expression for the exchange interaction contains the bare dual Green's functions $\tilde{\mathcal{G}}$ (see Ref. [134]). Note that in Eq. (27) the inverse of the polarization operator of the reference problem appears to the left and right of the dual polarization operator due to a different definition of the vertex function used in Refs. [132–134].

We stress that the obtained relations between the dual and the lattice quantities are valid for any form of the self-energy and the polarization operator of the partially bosonized dual action (5). Focusing on the D-TRILEX approach, it accounts for the leading diagrammatic contributions in both the weak and the strong coupling limits, as previously mentioned. Specifically,

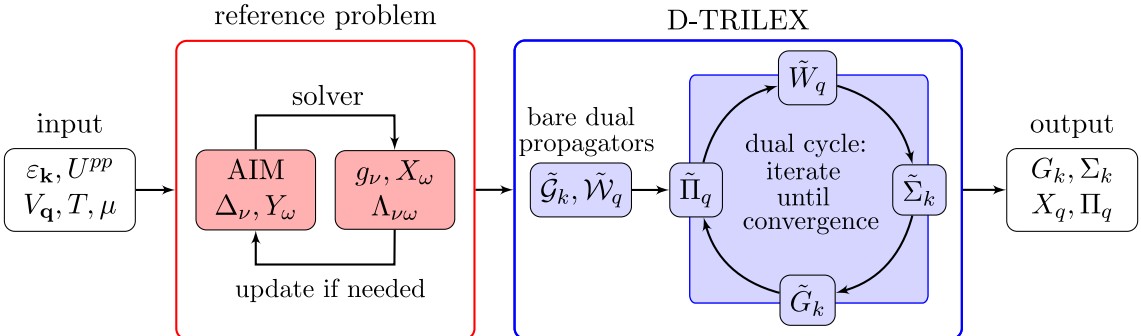

Figure 2: Workflow of the D-TRILEX method. The input consists in the parameters of the electronic lattice problem (Init.1-2 in the main text). The red box indicates the solution of the reference impurity problem (Init.3), that in some case has to be updated until self-consistency is reached (for instance in DMFT, EDMFT and cluster DMFT). The blue box contains the operations performed in the dual space, i.e. the calculation of the bare dual propagators (Init.4) and the self-consistency cycle on the dual quantities (St.2). The output consists in the Green's function, self-energy, susceptibility and polarization operator of the lattice problem, obtained by applying the exact relations between lattice and dual quantities (St.3).

the D-TRILEX polarization operator (19) becomes the leading contribution (the second-order in terms of the dual Green's functions) to the susceptibility (apart from the impurity polarization operator) in the strong coupling limit. More elaborate contributions have at least four dual Green's functions in their structure, so they can be disregarded and D-TRILEX becomes accurate in the regime close to the atomic limit also at the two-particle level. As an additional confirmation of this fact, the expression in Eq. (27) gives the correct result for the exchange interaction $\sim t^2/U$ in the atomic limit [132]. For the sake of completeness, we note that the D-TRILEX solution based on the DMFT reference problem is exact for arbitrary interaction strength in the limit of infinite dimensions at the single-particle level, but not at the two particle level, because it uses the partially bosonized approximation (see Appendix B) instead of the exact four-point vertex function. However, based on the results presented in the current and previous works [94], we stress that there is no correlation between the exactness of the theory in the limit of infinite dimensions and the accuracy of the theory in finite dimensions.

## 4 Computational workflow

In this section, we offer a detailed description of the current implementation (*available upon reasonable request from the corresponding authors*). We also put some emphasis in the discussion of the main issues faced when performing actual calculations.

### 4.1 Structure of the calculation

The computational workflow is divided into several parts, as shown in Fig 2. The first step (St.1) consists in solving the reference system, e.g. the DMFT impurity problem. This produces the inputs necessary for the initialization of the diagrammatic part of the calculation. Hence, the inner steps are denoted with (Init.). The second step (St.2) takes care of the self-consistent dressing of the dual Green's function $\tilde{G}$ and the renormalized interaction $\tilde{W}$. The inner steps (I.) of the self-consistent diagrammatic iteration are highlighted below. After the dressed dual quantities are determined, the single- and two-particle quantities for the initial

(lattice) problem are evaluated at the third step (St.3). The computational workflow has the following form:

(St.1) Input initialization:

  (Init.1) Specify the single-particle term $\varepsilon_{\mathbf{k},ll'}$, the interactions $U^{pp/ph}_{l_1l_2l_3l_4}$ and $V^{\varsigma}_{q,l_1l_2,l_3l_4}$, and the temperature $T$ that enter the initial action (1).

  (Init.2) Define the chemical potential $\mu$ and the hybridization function $\Delta_{\nu,ll'}$.

  (Init.3) Solve the reference system and get the corresponding Green's function $g_{\nu,ll'}$, the susceptibility $\chi^{\varsigma}_{\omega,l_1l_2,l_3l_4}$, and the vertex function $\Lambda^{\varsigma}_{\nu\omega,l_1,l_2,l_3l_4}$.

  (Init.4) Compute the bare fermionic $\tilde{\mathcal{G}}_{k,ll'}$ and bosonic $\tilde{\mathcal{W}}^{\varsigma}_{q,l_1l_2,l_3l_4}$ propagators of the effective partially bosonized dual action (5) according to Eqs. (7) and (8), respectively.

(St.2) Self-consistent calculation of D-TRILEX diagrams:

  (I.1) Compute the dual polarization operator $\tilde{\Pi}$ using Eq. (19).

  At the first iteration only: Compute the exchange interaction $\mathcal{J}^{\varsigma}_{q,l_1l_2,l_3l_4}$ via Eq. (27).

  (I.2) Compute the dual renormalized interaction $\tilde{W}$ using Eq. (16).

  (I.3) Compute the diagrams $\tilde{\Sigma}^{\text{TP}}$ (17) and $\tilde{\Sigma}^{GW}$ (18) for the dual self-energy.

  (I.4) Compute the dressed dual Green's function $\tilde{G}$ using Eq. (15).

  (I.5) If the desired accuracy $\delta$ for the self-consistent condition is reached, go to (St.3). Otherwise go back to (I.1).

(St.3) Evaluation of lattice quantities:

  Compute the dressed Green's function $G_{k,ll'}$ (20), the self-energy $\Sigma_{k,ll'}$ (21), the susceptibility $X^{\varsigma}_{q,l_1l_2,l_3l_4}$ (25), and the polarization operator $\Pi^{\varsigma}_{q,l_1l_2,l_3l_4}$ (26) for the lattice problem (1). From these quantities determine the orbital-resolved average density $\langle n_l \rangle$ (99) and the average energy $\langle E \rangle$ of the system (100) and (101). If one aims at the specific density $\langle n \rangle$, it is possible to update the chemical potential $\mu$ and go back to the beginning of the outer loop. In that case, go to (St.1 of Init.2) and fix the new $\mu$ and update the hybridization function $\Delta_{\nu,ll'}$ if needed.

## 4.2  Details of the calculation

The complexity of the diagrammatic part of the D-TRILEX calculation is estimated as

$$\mathcal{O}(N_\nu N_\omega)\, \mathcal{O}\left( N_{\text{imp}}^2 \times \sum_{i=1}^{N_{\text{imp}}} N_{l_i}^8 \right) \mathcal{O}(N_k \log N_k)\,, \tag{28}$$

where $N_{\nu(\omega)}$ is the number of fermionic (bosonic) Matsubara frequencies, $N_{\text{imp}}$ is the number of impurities in the reference system, $N_{l_i}$ is the number of orbitals for the $i$-th impurity and $N_k$ is the total number of $\mathbf{k}$-points. In this context, $N_{\text{imp}}$ is the number of independent impurities in the unit cell of the reference problem. Note that the case of $N_{\text{imp}} > 1$ corresponds to a collection of impurities, as explained in Ref. [122], and not to a cluster of $N_{\text{imp}}$ sites. If the impurities are all identical, then the reference system reduces to a single site impurity problem. If some of them are different, it is sufficient to solve an impurity problem only for the non-equivalent ones. In the multi-impurity case, fluctuations between the impurities are taken into account diagrammatically in the framework of D-TRILEX approach. On the other hand, a cluster reference system corresponds to a multi-orbital problem with $N_{\text{imp}} = 1$. In this case,

$N_l$ is the total number of orbitals and sites of the considered cluster. The separation between orbitals and sites that we introduce is useful to reduce the computational complexity when addressing problems with several atoms in the unit cells.

The scaling as a function of **k**-points is determined from the fact that we utilise the fast-Fourier transform (FFT) algorithm for computing convolutions in momentum space. This shows that the multi-impurity calculation has a quadratic scaling with respect to the number of impurities. In our current implementation, the local Coulomb matrix is considered as a non-sparse matrix within each site subspace, hence the scaling to the 8th power in the number of orbitals. However, before running the actual calculations, we introduced a check to assess which components of the vertices are zero. These components are automatically skipped in order to avoid unnecessary calculations and to automatically take advantage of a possible sparsity of the Coulomb matrix, effectively reducing the complexity (28) in most cases. The summation over frequencies and band indices can be efficiently parallelized both in a shared-memory framework (as done in the current implementation) and in an message-passing interface (MPI) framework.

To measure the accuracy at the $n$-th iteration of the self-consistent cycle, we use the relative Frobenius norm of the Green's function $F = ||\tilde{G}_n - \tilde{G}_{n-1}||/||\tilde{G}_{n-1}||$ as a metric, where $||...||$ is the square root of the squared sum over all the components of the array. If $F$ is smaller than some predefined accuracy value $\delta$, the self-consistent cycle stops. The cycle stops also if a specified maximum number of iterations is reached. The stability of the bosonic Dyson equation (16) can be problematic in regimes of parameters, where one or more of the eigenvalues of the quantity $\tilde{\Pi} \cdot \tilde{\mathcal{W}}$ become equal or larger than 1. In particular, this happens when the system is close to a phase transition or if the correlation length in some channel of instability exceeds a critical value. This issue appears in similar forms in other diagrammatic extensions of DMFT (see, e.g., Ref. [138]). In one- and two-dimensional systems, where Mermin-Wagner theorem forbids the breaking of continuous symmetries [139], the issue can be mitigated by imposing that the eigenvalues $\lambda_i$ of the $\tilde{\Pi} \cdot \tilde{\mathcal{W}}$ matrix in the orbital space for a physical meaningful solution are always smaller than 1. In our implementation, we check whether any eigenvalue $\lambda_i \geq 1$ ($i \in \{k, \varsigma\}$). If this happens, the eigenvalue can be rescaled as described in Ref. [138] in order to improve convergence.

Several strategies can be used to improve the stability of the self-consistent procedure in the general case. The first strategy is implemented when updating the D-TRILEX self-energy. The updated dual self-energy at the $n$-th iteration is computed as $\tilde{\Sigma}_n = (1 - \xi)\tilde{\Sigma}_{n-1} + \xi\tilde{\Sigma}$ for $\xi \in (0, 1)$, where $\tilde{\Sigma}_{n-1}$ is the value of the dual self-energy computed at the previous $(n-1)$ iteration, and $\tilde{\Sigma}$ is computed using the propagators $\tilde{G}_{n-1}$ and $\tilde{W}_{n-1}$ obtained at the previous iteration. This procedure was shown to improve stability in $GW$-like theories [140]. A similar mixing scheme can be applied to the dual polarization. To the same aim, we also introduce multiplicative factors for the dual self-energy and the dual polarization at the first iteration. Of course, no rescaling is expected to work in the presence of the symmetry breaking due to a true phase transition. The latter case should be addressed using a suitable cluster or multi-impurity reference problem.

It is worth mentioning that the efficiency of the whole scheme is strongly affected by the computational cost of the impurity solver. In our tests, the time needed to solve the impurity problem and to obtain the required correlation functions of the reference system using continuous time quantum Monte Carlo solvers [141–144] always exceeds the computational cost for the diagrammatic part of the calculation, even by several orders of magnitude. For example, a single iteration of the self-consistent diagrammatic cycle for a two-orbital case discussed in Section 5.3 takes only few minutes.

# 5 Application to relevant physical systems

In this section, we apply the above described D-TRILEX approach to several model systems to illustrate various capabilities of the method. We take the impurity problem of DMFT as a reference system for these particular calculations. The DMFT results are obtained using the w2dynamics package [145]. It should be noted, that this choice for the reference system is not always optimal (see e.g. Ref. [94]), in particular in the case of a dimer [99]. On the other hand, this choice allows one to consistently investigate the effect of non-local collective electronic fluctuations that are taken into account beyond the local DMFT approximation.

In the following, we consider model systems, where the on-site Coulomb interaction term in the Hamiltonian is parametrized in the Kanamori form [146, 147] as:

$$H_U = U \sum_l n_{l\uparrow} n_{l\downarrow} + \sum_{l \neq l'} \left\{ U' n_{l\uparrow} n_{l'\downarrow} + \frac{1}{2}(U' - J) \sum_\sigma n_{l\sigma} n_{l'\sigma} - J c_{l\uparrow}^\dagger c_{l\downarrow} c_{l'\downarrow}^\dagger c_{l'\uparrow} + J c_{l\uparrow}^\dagger c_{l\downarrow}^\dagger c_{l'\downarrow} c_{l'\uparrow} \right\}. \tag{29}$$

Here, $c_{l\sigma}^{(\dagger)}$ is the annihilation (creation) operator for an electron at the orbital $l$ with the spin projection $\sigma \in \{\uparrow, \downarrow\}$. $n_{l\sigma} = c_{l\sigma}^\dagger c_{l\sigma}$ is the local spin-dependent density. In the notation of Eq. (1), the non-zero components of the Kanamori interaction are

$$
\begin{aligned}
U_{llll}^{pp} &= U && \text{intraorbital density-density}, \\
U_{ll'll'}^{pp} &= U' && \text{interorbital density-density}, \\
U_{ll'l'l}^{pp} &= J && \text{pair hopping}, \\
U_{lll'l'}^{pp} &= J && \text{spin flip}.
\end{aligned} \tag{30}
$$

We also fix $U' = U - 2J$ to ensure rotational invariance [147]. In the single-orbital case, the on-site Coulomb interaction reduces to a Hubbard form given by the first term in Eq. (30).

This section is organized as follows. In Section 5.1 we discuss the performance of the D-TRILEX approach in application to the Hubbard-Kanamori dimer system, for which the exact numerical solution can be obtained using the exact diagonalization (ED) method. In Section 5.2 we study the effect of the non-local interaction $V_{\mathbf{q}}^d$ for the case of the extended Hubbard model on a square lattice. The obtained D-TRILEX result is compared to the one of the dual boson diagrammatic Monte Carlo (DiagMC@DB) approach [83]. In Section 5.3 we analyze electronic correlations in the framework of a two-orbital Hubbard-Kanamori model on a square lattice. Finally, in Section 5.4 we show the results for a bilayer square lattice Hubbard model as a particular example of a multi-impurity calculation.

## 5.1 Hubbard-Kanamori dimer as a benchmark system

The first case-study we discuss is a two-site model, also known as dimer. Due to a small size of this system, the exact solution for the dimer problem for small number of orbitals can be achieved by ED. This makes the dimer an ideal platform to benchmark various approximate methods. To test our multi-orbital D-TRILEX implementation, we consider a particular case of a Hubbard-Kanamori dimer, where each of the two identical sites has two degenerate orbitals. The single-particle part of the corresponding Hamiltonian reads:

$$H_0 = -t \sum_{l,\sigma} \sum_{j \neq j'} c_{j\sigma l}^\dagger c_{j'\sigma l}. \tag{31}$$

The single-particle Hamiltonian (31) can be diagonalized in the site-space. After that, the dimer problem can be effectively considered as a periodic system with the dispersion

$$\varepsilon_{\mathbf{k},ll'} = -2t \cos(\mathbf{k}) \delta_{ll'}, \tag{32}$$

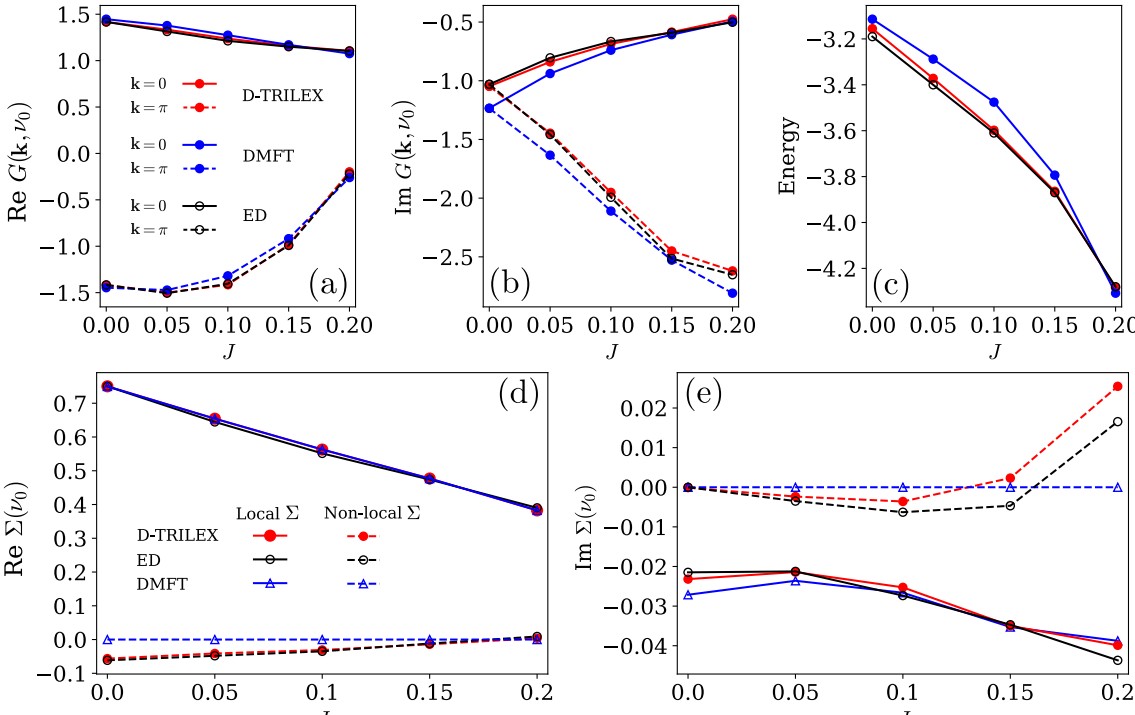

Figure 3: Panels (a) and (b) respectively show the real and the imaginary parts of the Green's function for the Hubbard-Kanamori dimer calculated for the frequency $\nu_0 = \pi/\beta$ at momenta $\mathbf{k} = 0$ (solid line) and $\mathbf{k} = \pi$ (dashed line). The panel (c) shows the average energy of the system. Panels (d) and (e) respectively show the real and the imaginary parts of the self-energy $\Sigma$ for the frequency $\nu_0 = \pi/\beta$. The local component is denoted by a think line, while the non-local component is represented by a dashed line. Non-local components are identically zero for DMFT, but are displayed for consistency. Results obtained using D-TRILEX (red), DMFT (blue), and ED (black) methods for different values of the Hund's coupling $J$. Model parameters for these calculations are $t = 0.2$, $U = 0.5$, $\beta = 10$, and $\mu = 0.75$, and are equal across all panels.

defined for $N_k = 2$ points in momentum space that correspond to $\mathbf{k} = 0$ (symmetric solution) and $\mathbf{k} = \pi$ (anti-symmetric solution). Based on this consideration, we can apply our multi-band D-TRILEX method, that is designed for solving periodic lattice models, to this benchmark system. The interacting part is considered in the Kanamori form (29) discussed above.

We chose the single-site two-orbital impurity problem of DMFT as the reference system for the D-TRILEX calculation. Since interorbital hoping processes are not taken into account, different orbitals do not hybridize. For the case of degenerate orbitals considered here it implies that the Green's function is diagonal in the orbital space and has identical components for both orbitals ($G_{ll'} = G\delta_{ll'}$). To compare the D-TRILEX result with the exact solution for the dimer problem we perform ED calculations using the pomerol package [148]. The total number of degrees of freedom for the two-orbital Hubbard-Kanamori dimer for the ED calculation is $2N_l N_{\text{imp}} = 8$ and the total number of states is $N_{\text{tot}} = 2^8 = 256$. This makes the ED calculation numerically inexpensive.

First, we focus on the effect of the Hund's exchange coupling $J$. To this aim we perform calculations for different values of $J$ fixing other model parameters to $t = 0.2$, $U = 0.5$, $\beta = 10$, and $\mu = 0.75$. A very similar set of model parameters for a single-orbital dimer problem was recently used in Ref. [99] to benchmark another diagrammatic extension of DMFT. In Fig.3,

we show the real (Re $G$, left panel) and imaginary (Im $G$, middle panel) parts of the Green's function produced by D-TRILEX (red), DMFT (blue), and ED (black) methods. We find that the D-TRILEX result for the Re $G$ lies on top of the exact solution in the whole range of values for the Hund's coupling considered here. DMFT is also rather accurate in calculating the real part of the Green's function, but the discrepancy between the DMFT and ED results is noticeable. The D-TRILEX solution for Im $G$ is very close to the one provided by ED, while the DMFT result becomes substantially different from the exact solution, especially for small values of $J$. A very good agreement between D-TRILEX and ED methods is also confirmed by analyzing the result for the average energy $\langle E \rangle$ (right panel in Fig. 3). The average energy for ED is obtained as $\langle E \rangle_{\text{ED}} = \sum_i (E_i - \mu) e^{-\beta(E_i - \mu)}$ where the index $i$ runs over the eigenstates of the system. The average energy in D-TRILEX is calculated using Eq. (103). The DMFT energy $\langle E \rangle_{\text{DMFT}}$ has been computed using the same formula (103) by setting $\tilde{\Sigma} = 0$ and $\tilde{\Pi} = 0$. We show that the mismatch in D-TRILEX and ED results for the energy is 1.1% ($\delta E = 0.034$) at $J = 0$ and decreases as $J$ increases. The largest difference between DMFT and ED results is found at $J = 0.1$ and amounts to 3.7% ($\delta E = 0.134$), which is approximately four times larger than the one of the D-TRILEX approach. Nevertheless, we observe that in this case DMFT is surprisingly close to the exact result. The reason is that for the considered set of model parameters the system lies very far away from half-filling, hence the non-local fluctuations between the two sites of the dimer are suppressed. This fact can be confirmed by looking at the self-energy $\Sigma$ shown in panels (d) and (e) of Fig. 3. The local contribution to the self-energy $2\Sigma^{\text{local}} = \Sigma(\mathbf{k} = 0) + \Sigma(\mathbf{k} = \pi)$ is dominant and is in a very good agreement among all three methods. The non-local part $2\Sigma^{\text{non-local}} = \Sigma(\mathbf{k} = 0) - \Sigma(\mathbf{k} = \pi)$, which is completely missing in DMFT, is relatively small and is also well reproduced by D-TRILEX approach.

At half-filling, DMFT ceases to be a good approximation. To illustrate that D-TRILEX is able to improve and even to cure a wrong behavior of the DMFT result, we perform calculations for $t = 0.5$ and $\beta = 10$ for different values of the Hubbard interaction $U$ for a fixed ratio $U/J = 4$. The chemical potential is set to $\mu = (3U - 5J)/2$ in order to ensure half-filling [149]. Panel (a) of Fig. 4 shows the imaginary part of the local Green's function as a function of the Matsubara frequency. The result is obtained in a weak ($U = 0.5$, dots) and strong coupling ($U = 2.0$, triangles) regimes of the interaction. At $U = 0.5$, the D-TRILEX result coincides with the exact solution in the whole frequency range. The DMFT result is also very accurate and only slightly deviates from the ED solution at lowest frequencies. This situation changes completely at $U = 2.0$, where the exact Im $G$ provided by ED is strongly reduced at low frequencies. Remarkably, DMFT does not capture this change and predicts approximately the same result for the Im $G$ for both values of the interaction. Instead, the D-TRILEX solution lies very close to exact result and reproduces the correct behavior of the Im $G$. To confirm this fact, we compute the normalized difference from the ED result for D-TRILEX and DMFT as

$$\delta\left[\text{Im}\, G(\nu_n)\right] = \text{Im}\left[G(\nu_n) - G_{\text{ED}}(\nu_n)\right] / \text{Im}\, G_{\text{ED}}(\nu_n). \tag{33}$$

The corresponding result obtained for three different frequencies as a function of $U$ is shown is the panel (b) of Fig. 4. We find that the normalized difference for DMFT is relatively large and drastically increases upon increasing the interaction strength. At $U = 2.0$, the Im $G(\nu)$ calculated at the zeroth and the first Matsubara frequency using DMFT is respectively almost two and 1.5 times larger than the exact result. On the contrary, the Im $G(\nu)$ of D-TRILEX lies very close to the ED result. Indeed, the normalized difference for D-TRILEX calculated for the first and the fifth frequency does not exceed 2%. The difference for D-TRILEX calculated for the zeroth frequency becomes larger than 2% at $U > 1.5$ and reaches the maximum value of 7.6% at $U = 2.0$. We find that the DMFT result strongly deviates from the considered benchmark at moderate and large values of $U$. By looking at the real part of the self-energy Re$\Sigma$ (panels (c) and (d) in Fig. 4), we can immediately understand the origin of the large mismatch

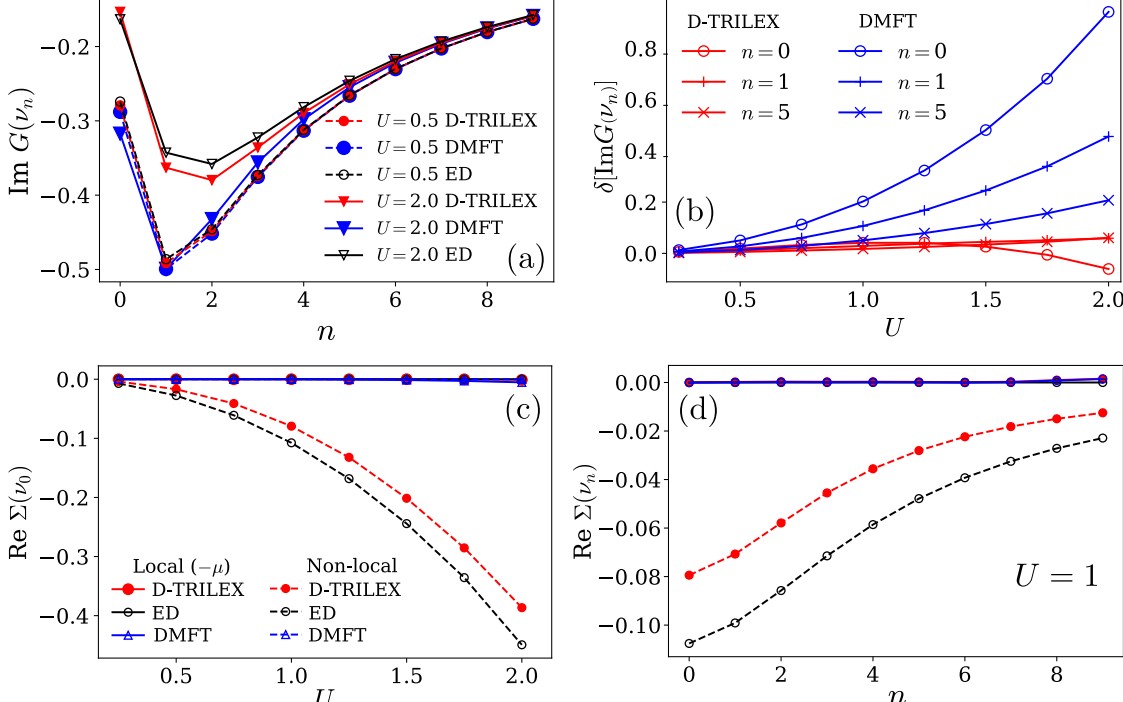

Figure 4: (a) Imaginary part of the local Green's function $\mathrm{Im}\,G(\nu_n)$ calculated as a function of the Mastubara frequency index $n$ for two values of the interaction $U = 0.5$ (dashed lines) and $U = 2.0$ (solid lines). The result is obtained at half-filling for $t = 0.5$, $\beta = 10$, and $J = U/4$ using the D-TRILEX (red), the DMFT (blue), and the ED (black) methods. (b) Normalized difference $\delta\left[\mathrm{Im}\,G(\nu_n)\right]$ (33) with respect to the ED solution calculated for D-TRILEX (red), the DMFT (blue) methods as a function of $U$. The result is obtained for three difference Matsubara frequencies with indices $n = 0$ (empty circles), $n = 1$ (pluses), and $n = 5$ (crosses). (c) Real part of the self-energy $\Sigma$ as a function of the interaction $U$ at the first Matsubara frequency $\nu_0$. (d) Real part of the self-energy $\Sigma(\nu_n)$ as a function of the Matsubara index obtained at $U = 1$. In panels (c) and (d), the local and non-local parts are shown and the chemical potential $\mu$ is subtracted from the local part.

between ED and DMFT. As a matter of fact, the real part of the self-energy at moderate to large $U$ is dominated by the non-local contributions (dashed lines), which are completely missing in DMFT, while local contributions (solid lines) are approximately zero. D-TRILEX does not exactly reproduce all the contributions to the non-local self-energy, as they correspond to roughly 25% of the value of self-energy at $U = 1$. However, it follows the same trend as the ED result and this ensures the correct behavior of the Green's function as $U$ is increased. We do not show the imaginary part of the self-energy, since it is at least an order of magnitude smaller than the real part in the whole range of parameters considered here.

In addition to single-particle quantities we calculate the charge and spin susceptibilities defined as $X^{\mathrm{ch/sp}} = -\sum_{ll'} X^{d/m}_{ll'l'}$. Fig. 5 shows the corresponding results for the static susceptibilities $X^{\mathrm{ch/sp}}(\mathbf{q}, \omega = 0)$ obtained at the $\mathbf{q} = \pi$ point. The susceptibilities at the $\mathbf{q} = 0$ point are very small in the whole range of considered parameters and are not shown here. In the left panel of Fig. 5, we illustrate the results for the half-filled Hubbard-Kanamori dimer considered above. We find that the susceptibilities of the D-TRILEX approach are in a very good agreement with the exact ED solution in the whole range of local interaction strength $0.25 \leq U \leq 2$. In the right panel, we demonstrate the dependence of the static charge and spin susceptibilities

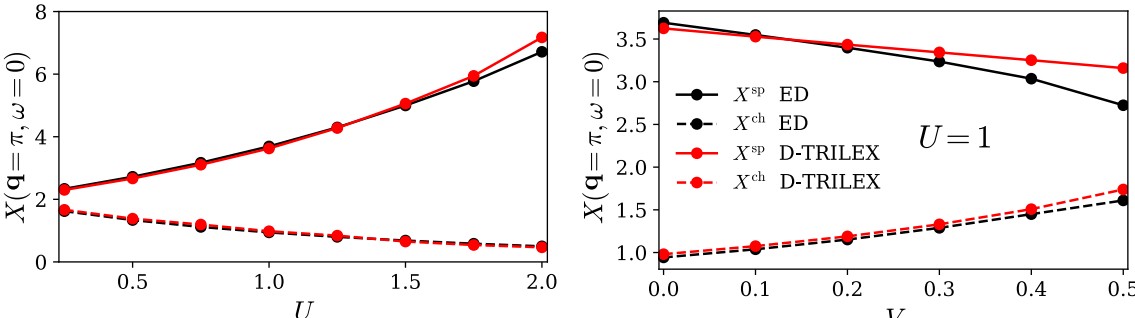

Figure 5: Spin (full lines) and charge (dashed lines) static ($\omega = 0$) susceptibility $X$ obtained at the $\mathbf{q} = \pi$ point for the half-filled system with model parameters $t = 0.5$, $J = U/4$, and $\beta = 10$. The left panel shows the result as a function of the local interaction strengths $U$ in the absence of the non-local interaction ($V = 0$). The right panel illustrates the susceptibility as a function of $V$ calculated for the fixed value of the local interaction $U = 1$.

on the value of the non-local interaction $V^d$ between electronic densities on neighboring sites $\langle i, j \rangle$ (1). More explicitly, we consider the non-local interaction in the form

$$\frac{V^d}{2} \sum_{ll',i \neq j} \rho^d_{i,ll} \rho^d_{j,l'l'} \,. \tag{34}$$

To simplify notations, in the following the superscript "$d$" for the non-local interaction is omitted. We find that in the presence of the non-local interaction the susceptibilities obtained using ED and D-TRILEX methods are nearly identical up to $V = 0.3$. Above that threshold, the D-TRILEX susceptibility starts to deviate from the exact ED result. At $V > 0.3$ the D-TRILEX spin susceptibility continues to decrease almost linearly with increasing the value of $V$, while the exact result shows a stronger non-linear damping. This trend continues also above $V = 0.5$, where the difference between the D-TRILEX result and the exact result continues to increase. This behavior can be explained by the fact that the strong non-local interaction favors either full or zero occupancy of a lattice site. This charge density wave instability strongly suppresses magnetic fluctuations. In this regime the D-TRILEX calculations break down, because they are performed on the basis of the DMFT impurity problem, which does not incorporate any effect of the non-local interaction. The inclusion of the bosonic hybridization function in the spirit of EDMFT could improve the result, because in this case some contributions of the non-local interaction would be taken into account in the impurity problem via the bosonic hybridization.

These findings show that D-TRILEX improves the DMFT results in all considered regimes. Additionally, D-TRILEX reproduces the trends observed in ED calculations in all the cases, even when DMFT fails. This fact suggests that for the considered system the difference between DMFT and ED mostly stems from non-local correlations that have the form accounted for in the D-TRILEX diagrams. These results are particularly remarkable taking into account that DMFT approximation is not very accurate in low dimensions, hence the DMFT impurity problem is probably not an optimal reference system for a diagrammatic expansion in this case.

## 5.2 Extended Hubbard model on a square lattice

Previous works on D-TRILEX [93–95] suggest that the method is able to account for the effect of the non-local interactions. However, no thorough benchmarking of the results for the extended Hubbard model has been performed so far. For this reason, in this work we investigate the performance of D-TRILEX in the case of a single-orbital extended Hubbard model on a

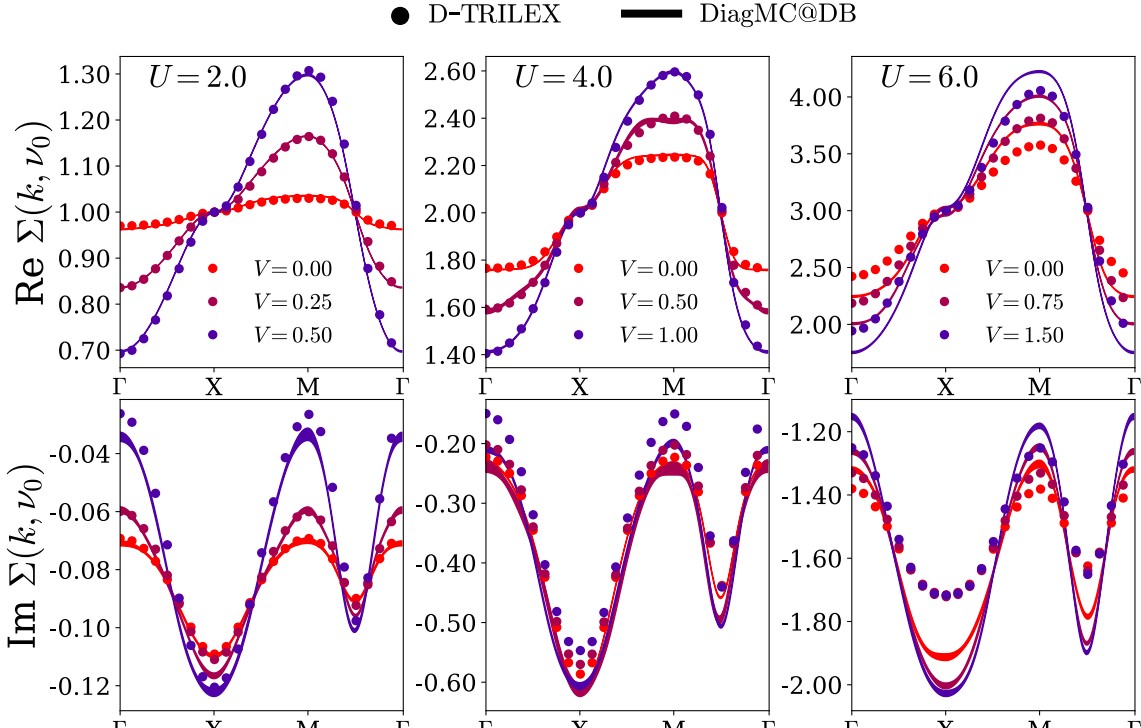

Figure 6: Real (top row) and imaginary (bottom row) parts of the lattice self-energy for the half-filled single-band extended Hubbard model on a square lattice. The result is obtained for $U = 2.0$ (left column), $U = 4.0$ (middle column), and $U = 6.0$ (right column) for three different values of the nearest-neighbor interaction $V = 0.0$ (light red), $V = U/8$ (dark red), and $V = U/4$ (purple). The D-TRILEX result is depicted by dots. The DiagMC@DB data is taken from Ref. [83] and is represented by solid lines with the width that corresponds to the estimated stochastic error.

square lattice with the local $U$ and the nearest-neighbor $V^d$ interactions between electronic densities (1). To simplify notations, in the following the superscript "$d$" for the non-local interaction is again omitted. The single-particle dispersion for this model for the case of a nearest-neighbor hopping amplitude reads

$$\epsilon_{\mathbf{k}} = -2t \left( \cos k_x + \cos k_y \right). \tag{35}$$

Similarly, the momentum-space representation for the non-local interaction is

$$V_{\mathbf{q}} = 2V \left( \cos q_x + \cos q_y \right). \tag{36}$$

We set the value of the nearest-neighbor hopping to $t = 1$, so that the half-bandwidth is $D = 4t = 4$. To benchmark our results, we compare the lattice self-energy $\Sigma$ calculated using D-TRILEX approach with the result of the dual boson diagrammatic Monte Carlo (DiagMC@DB) method presented in Ref. [83]. DiagMC@DB allows for the exact solution of an effective dual boson action (55), where the renormalized interaction is truncated at the two-particle level (50). Note that the dual boson action is derived within the exact analytical transformation of the initial lattice problem (1) (see Appendix A.1). In addition, diagrammatic Monte Carlo methods applied to dual theories show a very good agreement with the exact results [75,94] and the results of the cluster methods [150,151].

We perform calculations at half-filling for different strengths of the Hubbard interaction $U = 2$, $U = 4$, and $U = 6$. For each value of $U$ we consider three different values of the nearest-neighbor Coulomb interaction $V = 0$, $V = U/8$, and $V = U/4$. As in Ref. [83], for $U = 2$ and

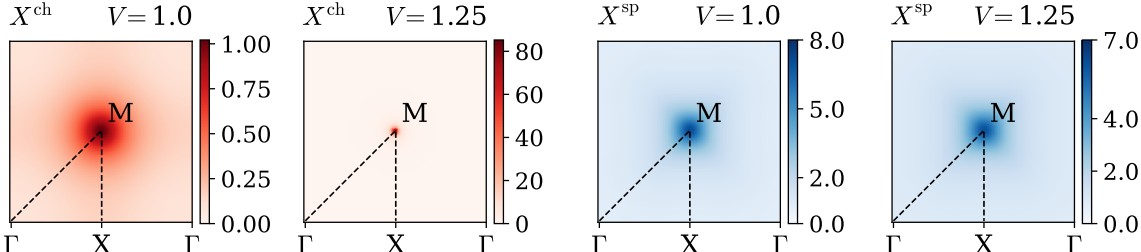

Figure 7: Static charge (red panels) and spin (blue panels) susceptibilities $X^{\text{ch/sp}}(\mathbf{q}, \omega = 0)$ in the Brillouin zone. Calculations are performed for the half-filled single-band extended Hubbard model on a square lattice for $U = 4$ and $\beta = 4$. The value of the non-local interaction $V$ is indicated above each panel.

$U = 4$ the temperature is set to $T = 0.25$, for $U = 6$ to $T = 0.50$. The obtained results for the lattice self-energy are shown in Fig. 6. We find that for the smallest value of the Hubbard interaction $U = 2$ the agreement between the two methods is almost perfect. A slight difference appears only in the imaginary part of the self-energy in the vicinity of the $X = (\pi, 0)$ point for $V = 0.25$ and near the $\Gamma = (0, 0)$ point for $V = 0.5$. When the interaction reaches the value of the half-bandwidth $U = 4$, the real part of the D-TRILEX self-energy remains very close to the DiagMC@DB result for all values of $V$ considered here. On the other hand, we observe a constant shift in the imaginary part of the self-energy that increases with the strength of the non-local interaction $V$. A constant but smaller shift was also observed between DF and DiagMC@DF results [75, 94], hence it does not seem to be a feature of only the D-TRILEX method. Finally, at $U = 6.0$ we observe that D-TRILEX does not agree with DiagMC@DB as accurately as for smaller values of the interaction. In the real part, the difference between the two methods is not very large and appears to be independent on the value of $V$. On the contrary, the imaginary part of the self-energy displays a rather large mismatch already at $V = 0$, and the agreement seems to become worse as $V$ increases. This result comes as no surprise and agrees with the findings of Refs. [75, 83, 94] that the ladder-like dual approximations become less accurate in the regime of strong magnetic fluctuations. The reason is that magnetic fluctuations become strongly non-linear close to a magnetic instability (see, e.g., Ref. [152]). This non-linear behavior originates from the mutual interplay between different bosonic modes as well as from an anharmonic fluctuation of the single mode itself. The description of these effects requires to consider much more complex diagrammatic structures that account for vertical (transverse) insertions of momentum- and frequency-dependent bosonic fluctuations, which are present in the DiagMC@DB approach but are not considered in ladder-like dual approximations including the D-TRILEX approach. However, despite the quantitative disagreement, at $U = 6.0$ D-TRILEX qualitatively captures the correct momentum dependence of the self-energy, which is completely missing in DMFT.

In addition to single-particle quantities, D-TRILEX also provides two-particle quantities, namely the susceptibility and the polarization operator of the lattice problem. These quantities are calculated as momentum- and frequency-dependent functions, which allows one to get the information about the full energy spectrum of the charge and spin excitations in the system. In Fig. 7, we show the static ($\omega = 0$) charge and spin susceptibilities $X^{\text{ch/sp}} = -X^{d/m}$ in the Brillouin zone (BZ) computed for the same model at $U = 4$, $T = 0.25$, and different values of the non-local interaction $V = 1.0$ and $V = 1.25$. Both, charge and spin susceptibilities display a maximum value at the $M = (\pi, \pi)$ point of the BZ, which signals that corresponding order parameters tend to have a checkerboard configuration on a square lattice. From the physical point of view, this means that in this parameter range the system has a tendency towards the charge density wave (CDW) and the antiferromagnetic (AFM) orderings. The

enormous increase in the value of the charge susceptibility indicates that the system is very close to the CDW transition point, that corresponds to a divergence of the charge susceptibility. On the contrary, the spin susceptibility does not change significantly, and its value is slightly reduced upon increasing $V$. This reduction is expectable, since in this particular case the spin fluctuations are screened by strong charge fluctuations.

It is commonly believed that in strongly-correlated systems the non-local interactions have to be treated in the framework of the extended DMFT by introducing a bosonic hybridization function in the impurity problem [126–130]. However, the diagrammatic expansion in dual theories can be performed for of an arbitrary reference system. In particular, the results of this section demonstrate that the D-TRILEX approach can accurately treat the non-local interactions on the basis of the DMFT impurity problem. The latter does not contain the bosonic hybridization function and thus is easier to solve numerically. In addition, in D-TRILEX the non-local collective electronic fluctuations are not restricted in the range, which is a big advantage over cluster extensions of DMFT. Moreover, the D-TRILEX method is able to capture the interplay between the collective electronic fluctuations in different channels through the self-consistent procedure that involves single-particle quantities. As a matter of fact, the bosonic propagators from all channels contribute to the self-energy. Therefore, a large value of the bosonic propagator in one channel considerably increases the value of the self-energy, hence it reduces the value of the Green's function. In turn, the reduced value of the Green's function leads to a smaller value of the dual polarization for app channels, which is the main ingredient for computing the physical susceptibility and the polarization operator. This feature is not provided by DMFT calculations of the susceptibility with dynamical vertex corrections [17,19,153] that are performed non-self-consistently.

## 5.3 Two-orbital Hubbard-Kanamori model

In this section we study the simplest multi-orbital system, a half-filled two-orbital Hubbard-Kanamori model on a square lattice. We consider the case when the orbitals have same $U$, $U'$, and $J$ values for the Kanamori interaction, but different bandwidths. This case is of interest because the two orbitals are thus characterized by a different effective interaction strength that can be defined as the ratio between the actual interaction and the width of the band. Our aim is to show that D-TRILEX is able to capture this basic feature of the model. For simplicity, we neglect the hybridization term between the two orbitals in the single-particle part of the Hamiltonian, although it can in principle be taken into account in our implementation. The dispersion for the corresponding orbital $l \in \{1, 2\}$ is given by the hoppings $t_l$ between the nearest-neighbor lattice sites $j$ and $j'$ on the square lattice. In momentum space, the electronic dispersion can be written as follows

$$\varepsilon_{\mathbf{k},ll'} = -2t_l \left( \cos k_x + \cos k_y \right) \delta_{ll'}. \tag{37}$$

We take $t_1 = 1$ and $t_2 = 0.75$, which corresponds to $D_1 = 4$ and $D_2 = 3$ values for the half-bandwidth $D_l$.

First, we calculate the local density of states (DOS) from the corresponding local part of the lattice Green's function by means of analytical continuation using the maximum entropy method implemented in the ana_cont package [154]. Fig. 8 shows the results obtained at the inverse temperature $\beta = 2$ for different values of the interaction $U = 4$, $U = 5$, and $U = 7$. Remaining parameters for the interaction are $U' = U/2$ and $J = U/4$. The top and bottom rows in this figure respectively show the DOS for the first ($l = 1$) and the second ($l = 2$) orbitals obtained using DMFT (blue) and D-TRILEX (red). We find that at $U = 4$ both DMFT and D-TRILEX results show a qualitatively similar behavior for the first orbital (top left panel). Thus, the DOS for the first orbital exhibits a quasi particle peak at Fermi energy ($E = 0$) and

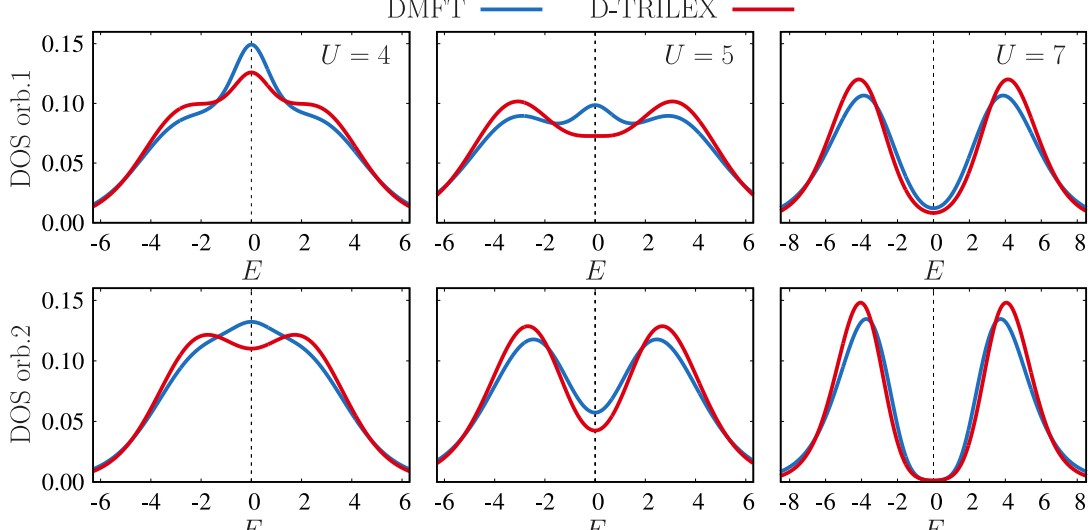

Figure 8: DOS for the two-orbital Hubbard-Kanamori model obtained using DMFT (blue line) and D-TRILEX (red line). The value of the interaction $U$ is indicated for each column. The other parameters for the interaction are $U' = U/2$ and $J = U/4$. The top and bottom rows display results for the orbitals $l = 1$ and $l = 2$, respectively. The half-bandwithds for these orbitals are $D_1 = 4$ and $D_2 = 3$.

two "shoulders" that reflect the strong Hubbard interaction. We observe that in D-TRILEX the quasi-particle peak is smaller and the sub-bands are more pronounced than in DMFT. These facts indicate that the D-TRILEX result lies closer to the Mott transition. A rather different but consistent behavior can be found in the DOS for the second orbital (bottom left panel). The D-TRILEX result shows that at $U = 4$ the quasiparticle peak is already destroyed, and the system starts to form a pseudogap at Fermi energy. However, DOS of DMFT still has a quasiparticle peak with shoulders that are smeared out by large temperature. At $U = 5$ (middle column) the quasiparticle peak in DOS of D-TRILEX disappears for both orbitals. In DMFT, the quasiparticle peak remains for the first orbital, but the DOS for the second orbital qualitatively agrees with the one of D-TRILEX and shows a pronounced pseudogap. Finally, at $U = 7$ DMFT and D-TRILEX agree qualitatively in the DOS for both orbitals. At this interaction strength the DOS for the first orbital still has a small spectral weight at Fermi energy, while the second orbital lies already in the Mott phase. Nevertheless, we find that D-TRILEX consistently predicts a smaller spectral weight at Fermi energy compared to DMFT for all considered values of the interaction. This result is in accordance with the fact that the non-local fluctuations at half-filling push the system closer to the insulating state [93].

We also calculate the susceptibility for the considered multi-orbital system. The charge, spin, and orbital components of the susceptibility can be obtained from the orbital-dependent D-TRILEX susceptibility as follows (see e.g. Ref. [147])

$$X^{\text{ch/sp}} = -\sum_{ll'} X^{d/m}_{ll'l'}, \qquad X^{\text{orb}} = -\sum_{ll'} X^{d}_{ll'll'} = -\sum_{ll'} X^{m}_{ll'll'}. \tag{38}$$

Fig. 9 shows the static lattice susceptibility $X(\mathbf{q}, \omega = 0)$ calculated along the high-symmetry path that connects $\Gamma = (0,0)$, $X = (0,\pi)$, and $M = (\pi,\pi)$ points in the Brillouin zone. The result is obtained for the charge (red), orbital (green), and spin (blue) components of the susceptibility for different values of the interaction $U$. We find that both, charge and orbital susceptibilities decrease upon increasing the interaction strength. At $U = 1$ these two susceptibilities reveal a peak at the M point. This momentum-structure indicates that the system

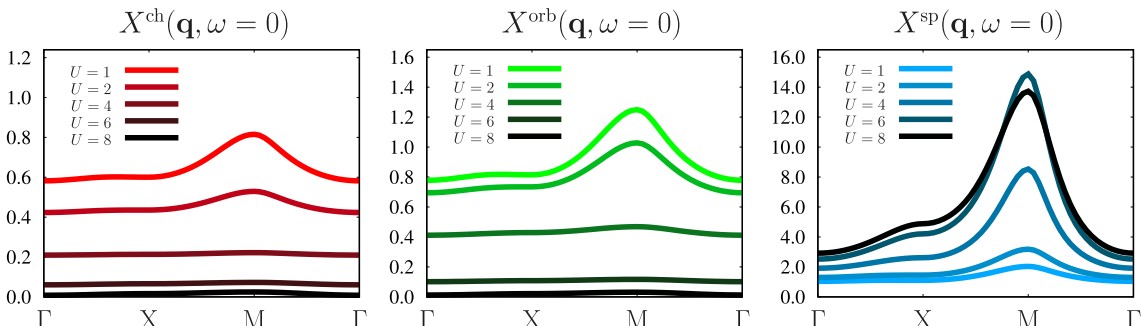

Figure 9: Charge (left), orbital (center) and spin (right) components of the static susceptibility $X(\mathbf{q}, \omega = 0)$ computed along the high-symmetry path in momentum space. Calculations are performed for the half-filled Hubbard-Kanamori model for different values of $U$ fixing other parameters to $\beta = 2$, $U' = U/2$, and $J = U/4$.

exhibits collective electronic fluctuations in the corresponding channel that are characterized by a $\mathbf{q} = (\pi, \pi)$ wave vector. Increasing the value of $U$ suppresses the peak at M point and consequently reduces the strength of the fluctuations. The spin susceptibility shows an opposite trend compared to $X^{\text{ch}}$ and $X^{\text{orb}}$. It increases with the interaction strength up to approximately $U = 6$ and then starts to decrease upon further increasing the interaction to $U = 8$. The spin susceptibility also shows a maximum at M point, but at strong interactions its value is much larger than the one of the charge and the orbital susceptibilities. This fact suggests that the considered model possesses well-developed antiferromagnetic (AFM) fluctuations that represent the main source of instability in the system.

## 5.4 Bilayer square lattice

As a final case study, we consider Hubbard model on a bilayer square lattice as a particular example of a multi-site system. The model Hamiltonian can be written in two equivalent forms with either one ($N_{\text{imp}} = 1$) or two ($N_{\text{imp}} = 2$) lattice sites in the unit cell. This fact makes the considered system ideal for testing and cross-validating implementations of multi-site calculations.

The single-particle term of the Hamiltonian with the two sites in the unit cell reads

$$\varepsilon_{\mathbf{k}, ll'}^{\text{2-site}} = -2t \left( \cos k_x + \cos k_y \right) \delta_{ll'} + 2t_\perp \sigma_{ll'}^x, \tag{39}$$

where $t$ and $t_\perp$ are respectively the intra- and interlayer hopping amplitudes. The index $l$ numerates the site within the unit cell, and $\sigma^x$ is the first Pauli matrix. In the limit $t \gg t_\perp$, the two layers are almost decoupled and the system behaves similarly to a single-layer square lattice Hubbard model. In the opposite limit $t \ll t_\perp$, the system behaves as a collection of dimers, where the two site in the same dimer belong to different layers. In this limit, the electrons have small probability to hop between neighboring dimers.

The introduced single-particle Hamiltonian can be diagonalized in the $\{l, l'\}$ space. The resulting dispersion has two bands that correspond to a tight-binding dispersion for a single-layer square lattice that is respectively shifted in energy by $\pm 2t_\perp$. We note that for $t_\perp = 2$ the relative shift between the bands becomes $\Delta E = 4t_\perp = 8$, which corresponds to the value of the bandwidth of each band. Therefore, at larger values of $t_\perp$ the two bands do not overlap and the non-interacting system becomes a band insulator. Formally, one can parametrize the two bands by introducing the label $k_z \in \{0, \pi\}$

$$\varepsilon_{\mathbf{k}}^{\text{1-site}} = -2 \left( t \cos k_x + t \cos k_y + t_\perp \cos k_z \right), \tag{40}$$

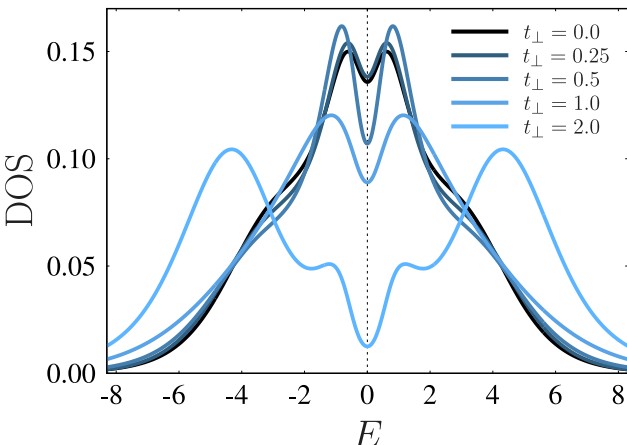

Figure 10: DOS for a bilayer square lattice obtained at $t = 1$, $U = 4$, $\beta = 4$ for different values of the interlayer hopping $t_\perp$. Lighter shades of blue indicate larger $t_\perp$.

in strict analogy with the transformation carried out in Section 5.1 for the Hubbard-Kanamori dimer. In this representation, the $k_z$ label plays a role of a $z$-component of momentum in the dispersion of a three-dimensional system that has one lattice site in the unit cell. Throughout the tests, we performed calculations using both the two-site and the single-site representations for the Hamiltonian and checked that the results identically coincide.

We perform calculations at half-filling for $t = 1$, $U = 4$, and $\beta = 4$, and investigate how the properties of the system change as a function of $t_\perp$. In order to understand the physics of this system, we compute both single- and two-particle observables, namely the electronic DOS (Fig. 10) and the static magnetic susceptibility (Fig. 11). In particular, we focus on the two independent components of the spin susceptibility $X_\parallel^{\mathrm{sp}}$ (left panel of Fig. 11) and $X_\perp^{\mathrm{sp}}$ (right panel of Fig. 11) that respectively describe magnetic fluctuations within and between the layers. In the two-site representation these components have the following form: $X_\parallel^{\mathrm{sp}} \equiv -X_{1111}^m = -X_{2222}^m$ and $X_\perp^{\mathrm{sp}} \equiv -X_{1122}^m = -X_{2211}^m$. These quantities can also be found in the single-site representation as: $2X_\parallel^{\mathrm{sp}} = X_{k_z=0}^{\mathrm{sp}} + X_{k_z=1}^{\mathrm{sp}}$ and $2X_\perp^{\mathrm{sp}} = X_{k_z=0} - X_{k_z=1}^{\mathrm{sp}}$. The charge susceptibility for the considered system is small and is not shown here.

At $t_\perp = 0$, which corresponds to the case of completely decoupled layers, the DOS exhibits a dip at the Fermi energy. A relatively large leading eigenvalue of the Bethe–Salpeter equation in the magnetic channel $\lambda = 0.8$ at the wave vector $\mathbf{q} = \mathrm{M}$ and the corresponding peak in the intralayer spin susceptibility $X_\parallel^{\mathrm{sp}}$ indicate that this dip signals the formation of a pseudogap due to strong AFM fluctuations. The interlayer component $X_\perp^{\mathrm{sp}}$ of the spin susceptibility is zero, since the two layers are decoupled. This result is in a perfect agreement with the behavior reported in Ref. [94] for a single-layer case. We find that for $t_\perp \leq 1$ the electronic spectral function is still dominated by the two peaks that lie close to the Fermi energy. The splitting between the two peaks increases with increasing $t_\perp$ in accordance with the results of Ref. [155] obtained for a bilayer Bethe lattice. This behavior can be partially attributed to the shift between the two non-interacting bands that is proportional to $t_\perp$. However, the value of this shift appears to be renormalized by electronic correlations. In addition, in this regime of $t_\perp$ magnetic fluctuations still play an important role, which is confirmed by relatively large values of both, inter- and intralayer components of the spin susceptibility. We find that increasing $t_\perp$ reduces the value of $X_\parallel^{\mathrm{sp}}$ and increases the value of $X_\perp^{\mathrm{sp}}$ at the M point. We attribute this behavior to the fact that fluctuations within the dimers become more and more important, which effectively reduces available degrees of freedom for in-plane magnetic fluc-

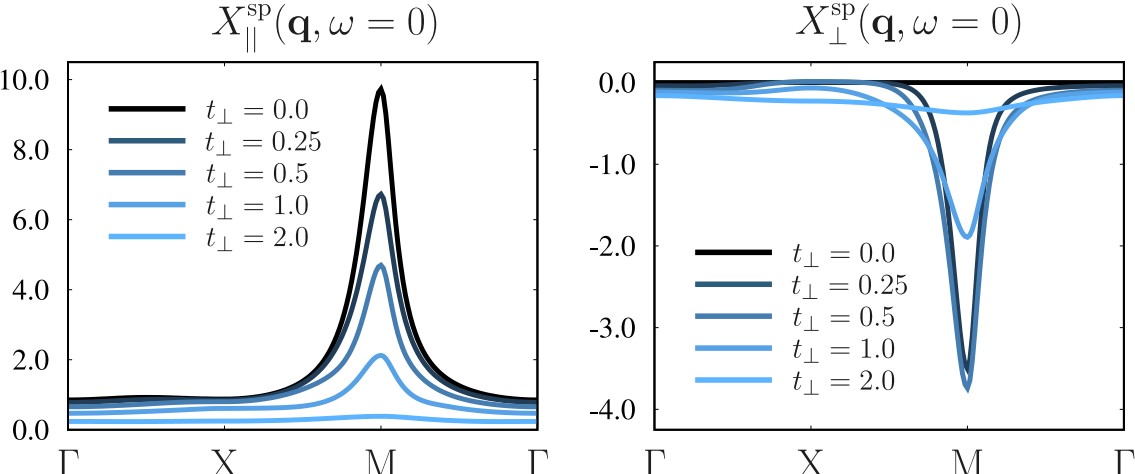

Figure 11: Interlayer $X_{\parallel}^{\mathrm{sp}}$ (left) and intralayer $X_{\perp}^{\mathrm{sp}}$ (right) components of the spin susceptibility obtained along the high symmetry path in momentum space for different values of the interlayer hopping $t_{\perp}$. Calculations are performed for the bilayer square lattice at $t = 1$, $U = 4$, and $\beta = 4$.

tuations. At $t_{\perp} = 2$, we observe a qualitative change in the DOS. In particular, it reveals a four-peak structure that consists in two small peaks at $E \simeq \pm 1$ and two large peaks at $E \simeq \pm 5$. The distance between the latter does not exactly coincide with the relative splitting between the two non-interacting bands $\Delta E = 8$. This shows that electronic correlations still play an important role in renormalizing the dispersion in this regime. However, a strong suppression of both components of the spin susceptibility indicates that this renormalization cannot be attributed to spatial magnetic fluctuations. The appearance of small peaks close to Fermi energy and a small spectral weight at Fermi energy indicate that at $t_{\perp} = 2$ the dimer physics in the system starts to dominate over the lattice physics. Thus, the splitting between the small peaks $\Delta E \simeq 2$ can be attributed to singlet fluctuations within the dimer that have energy of the exchange interaction $\mathcal{J} = 2t_{\perp}^2/U = 2$.

These findings are consistent with what has been observed previously for the case of a bilayer Bethe lattice [155, 156] and with the results of cluster DMFT studies for the bilayer square lattice [157]. In spite of the different geometric structures, the phase diagrams for the bilayer Bethe lattice and the bilayer square lattice appear to be both characterized by a metallic phase, a correlation-driven insulating phase and a band-insulating phase induced by the formation of dimers. Indeed, a transition from a metal to a band-insulator upon increasing $t_{\perp}$ was reported for small value of the interaction, while a transition from an AFM pseudogap regime to a band-insulator was found at larger interactions for both models. In our calculations, we find that D-TRILEX qualitatively reproduces the behaviour observed in the mentioned works capturing the expected crossover between these regimes. Hence, D-TRILEX can be seen as a reliable tool for describing the complex physics in the considered model. A more detailed analysis of the phase diagram is desirable and deserves further studies.

## 6 Conclusion

We introduced the D-TRILEX theory in a general multi-orbital and multi-site framework. This method is designed to tackle strongly-interacting electronic materials and allows for a self-consistent treatment of both single- and two-particle physical observables. The balanced diagrammatic structure of D-TRILEX accounts for the desired vertex corrections and the lead-

ing non-local collective electronic fluctuations, while remaining computationally feasible in a multi-band context. The method also allows for considering the frequency- and momentum-dependent interactions without introducing bosonic hybridizations in the reference system.

We provided a detailed derivation and description of all the constituent equations. We also illustrated the computational workflow that was used in our numerical implementation. Additionally, we discussed some hints that allowed us to improve the performances and the stability of the self-consistent procedure. Further, we investigated electronic correlations in four relevant model systems. They were chosen either to compare the performance of the method with existing benchmark results, or to illustrate specific capabilities of the developed approach. We focused on zero- and two-dimensional systems to show that D-TRILEX correctly accounts for the correlation effects even in low dimensions, where DMFT is not always expected to provide an optimal reference system. We have found that D-TRILEX shows a very good agreement with the benchmarks in all considered cases. Where benchmarks are not available, the results of the method reproduced general trends reported in previous studies of similar models.

Since the method can be formulated on the basis of an arbitrary reference system, it allows for several possibilities for further improvements of the results by tuning the parameters of the reference system, especially in situations where the DMFT impurity problem does not provide a good starting point for the D-TRILEX diagrammatic expansion. For instance, the reference system can be improved by considering a cluster of suitable size, or by introducing more appropriate fermionic and bosonic hybridization functions. In addition to that, the solution of the derived partially bosonized dual action (5) can be systematically improved by considering more elaborate diagrammatic contribution that are not taken into account in D-TRILEX [94].

Even though not included in our current computational scheme, one can also address the problem of superconductivity in the framework of the developed approach. In order to find the transition temperature to a superconducting state one can look at the divergence of the lattice susceptibility (72) in the corresponding particle-particle channel. To this aim one can either consider a suitable reference problem related to the desired superconducting order parameter (see, e.g., Refs. [56, 131, 158, 159]) and use the D-TRILEX form for the polarization operator in the particle-particle channel, or to additionally account for the scattering on the transverse momentum- and frequency-dependent bosonic fluctuations in the polarization operator in the particle-particle channel in the case of a single-site reference system [91, 160, 161]. Going inside the superconducting phase would require to introduce an anomalous component of the Green's function working in the Nambu space formalism similarly to what has been proposed in the framework of the TRILEX approach [91].

We believe that the ability of our approach to consistently account for the effect of the non-local collective electronic fluctuations could open a route towards an accurate description of a broad class of materials where multi-orbital effects play a crucial role. The use of our approach should especially be appealing in situations where the non-local correlations where so far neglected due to the lack of a computationally feasible method, and where the local approximations still represent the *state-of-the-art* solution of the problem.

# Acknowledgements

M.V. is thankful to Niklas Witt for the fruitful discussions. M.V., V.H., and A.I.L. acknowledge the support by the Cluster of Excellence "Advanced Imaging of Matter" of the Deutsche Forschungsgemeinschaft (DFG) - EXC 2056 - Project No. ID390715994, and by North-German Supercomputing Alliance (HLRN) under the Project No. hhp00042. V.H. and A.I.L. further acknowledge the support by the DFG through FOR 5249-449872909 (Project P8) and the European Research Council via Synergy Grant 854843-FASTCORR. The work of E.A.S. was

supported by the European Union's Horizon 2020 Research and Innovation programme under the Marie Skłodowska Curie grant agreement No. 839551 - 2DMAGICS.

**Author contributions**   M.V., V.H., A.I.L., and E.A.S. contributed to the development of the theory. M.V., J.K., and M.E.-N. contributed to the development of the numerical implementation. M.V. and V.H. contributed to the data collection. All the authors contributed equally to the discussion of the results and to writing of the manuscript.

**Data availability**   The data shown here and the program package used to obtain the results of this work is available from the corresponding authors upon reasonable request.

## A   Derivation of the multi-band D-TRILEX approach

In this Appendix we present the derivation of the multi-band D-TRILEX approach following Refs. [93, 94], where the method was introduced in the single-band framework.

### A.1   Effective partially bosonized dual action

We start with isolating the reference (impurity) problem from the initial lattice action (1). The action for the reference system

$$
\begin{aligned}
\mathcal{S}_{\text{imp}} = & -\sum_{\substack{\nu,\{l\}, \\ \sigma\sigma'}} c^*_{\nu\sigma l}\left[(i\nu+\mu)\delta_{\sigma\sigma'}\delta_{ll'} - \Delta^{\sigma\sigma'}_{\nu,ll'}\right]c_{\nu\sigma'l'} \\
& + \frac{1}{2}\sum_{\substack{q,\{k\}, \\ \{l\},\{\sigma\}}} U^{pp}_{l_1 l_2 l_3 l_4} c^*_{k,\sigma,l_1} c^*_{q-k,\sigma',l_2} c_{q-k',\sigma',l_4} c_{k',\sigma,l_3} + \frac{1}{2}\sum_{\omega,\{l\},\varsigma} Y^{\varsigma}_{\omega, l_1 l_2, l_3 l_4}\,\rho^{\varsigma}_{-\omega, l_1 l_2}\,\rho^{\varsigma}_{\omega, l_4 l_3} \\
& + \sum_{\omega,\{l\},\vartheta} Y^{\vartheta}_{\omega, l_1 l_2, l_3 l_4}\,\rho^{*\vartheta}_{\omega, l_1 l_2}\,\rho^{\vartheta}_{\omega, l_3 l_4}
\end{aligned}
\tag{41}
$$

contains momentum-independent parts of the lattice action (1). In addition to them, we introduce fermionic $\Delta^{\sigma\sigma'}_{\nu,ll'}$ and bosonic $Y^{r}_{\omega, l_1 l_2, l_3 l_4}$ hybridization function that usually enter the impurity problem of the (extended) DMFT [48, 126–130]. In Eq. (41) these quantities are written in a general frequency $\nu$ ($\omega$), band $l$, spin $\sigma$, and channel $r \in \{\varsigma,\vartheta\}$ dependent form. These hybridization functions are usually determined according to some self-consistency condition and their form depends on the choice of the reference system and on the initial problem. The composite variables that are coupled to the bosonic hybridization functions are defined in the main text. The corresponding densities for the particle-particle channel are following

$$
\begin{aligned}
n^{s}_{q, l_1 l_2} &= \frac{1}{2}\sum_{k}\left(c_{q-k,\downarrow l_2}c_{k\uparrow l_1} - c_{q-k,\uparrow l_2}c_{k\downarrow l_1}\right), & n^{*s}_{q, l_1 l_2} &= \frac{1}{2}\sum_{k}\left(c^*_{k\uparrow l_1}c^*_{q-k,\downarrow l_2} - c^*_{k\downarrow l_1}c^*_{q-k,\uparrow l_2}\right), \\
n^{t0}_{q, l_1 l_2} &= \frac{1}{2}\sum_{k}\left(c_{q-k,\downarrow l_2}c_{k\uparrow l_1} + c_{q-k,\uparrow l_2}c_{k\downarrow l_1}\right), & n^{*t0}_{q, l_1 l_2} &= \frac{1}{2}\sum_{k}\left(c^*_{k\uparrow l_1}c^*_{q-k,\downarrow l_2} + c^*_{k\downarrow l_1}c^*_{q-k,\uparrow l_2}\right), \\
n^{t+}_{q, l_1 l_2} &= \frac{1}{\sqrt{2}}\sum_{k}c_{q-k,\uparrow l_2}c_{k\uparrow l_1}, & n^{*t+}_{q, l_1 l_2} &= \frac{1}{\sqrt{2}}\sum_{k}c^*_{k\uparrow l_1}c^*_{q-k,\uparrow l_2}, \\
n^{t-}_{q, l_1 l_2} &= \frac{1}{\sqrt{2}}\sum_{k}c_{q-k,\downarrow l_2}c_{k\downarrow l_1}, & n^{*t-}_{q, l_1 l_2} &= \frac{1}{\sqrt{2}}\sum_{k}c^*_{k\downarrow l_1}c^*_{q-k,\downarrow l_2}.
\end{aligned}
\tag{42}
$$

The remaining part of the lattice action reads

$$
\mathcal{S}_{\text{rem}} = \sum_{k,\{l\}} \sum_{\sigma\sigma'} c^*_{k\sigma l} \tilde{\varepsilon}^{\sigma\sigma'}_{k,ll'} c_{k\sigma'l'} + \frac{1}{2} \sum_{q,\{l\},\varsigma} \tilde{V}^{\varsigma}_{q,l_1 l_2,l_3 l_4} \rho^{\varsigma}_{-q,l_1 l_2} \rho^{\varsigma}_{q,l_4 l_3} + \sum_{q,\{l\},\vartheta} \tilde{V}^{\vartheta}_{q,l_1 l_2,l_3 l_4} \rho^{*\vartheta}_{q,l_1 l_2} \rho^{\vartheta}_{q,l_3 l_4},
$$
(43)

where $\tilde{\varepsilon}^{\sigma\sigma'}_{k,ll'} = \varepsilon^{\sigma\sigma'}_{\mathbf{k},ll'} - \Delta^{\sigma\sigma'}_{\nu,ll'}$ and $\tilde{V}^{r}_{q,l_1 l_2,l_3 l_4} = V^{r}_{q,l_1 l_2,l_3 l_4} - Y^{r}_{\omega,l_1 l_2,l_3 l_4}$. Let us perform following Hubbard-Stratonovich transformations for the partition function of $\mathcal{S}_{\text{rem}}$

$$
\exp\left\{ -\sum_{k,\{l\}} \sum_{\sigma\sigma'} c^*_{k\sigma l} \tilde{\varepsilon}^{\sigma\sigma'}_{k,ll'} c_{k\sigma'l'} \right\} =
$$
(44)

$$
\mathcal{D}_f \int D[f^*,f] \exp\left\{ \sum_{k,\{l\}} \sum_{\sigma\sigma'} \left( f^*_{k\sigma l} \left[ \tilde{\varepsilon}^{-1}_k \right]^{\sigma\sigma'}_{ll'} f_{k\sigma'l'} - f^*_{k\sigma l} c_{k\sigma l} - c^*_{k\sigma l} f_{k\sigma l} \right) \right\},
$$

$$
\exp\left\{ -\frac{1}{2} \sum_{q,\{l\},\varsigma} \rho^{\varsigma}_{-q,l_1 l_2} \tilde{V}^{\varsigma}_{q,l_1 l_2,l_3 l_4} \rho^{\varsigma}_{q,l_4 l_3} \right\} =
$$
(45)

$$
\mathcal{D}_\varphi \int D[\varphi^{\varsigma}] \exp\left\{ \sum_{q,\{l\},\varsigma} \left( \frac{1}{2} \varphi^{\varsigma}_{-q,l_1 l_2} \left[ \left( \tilde{V}^{\varsigma}_q \right)^{-1} \right]_{l_1 l_2,l_3 l_4} \varphi^{\varsigma}_{q,l_4 l_3} - \varphi^{\varsigma}_{-q,l_1 l_2} \rho^{\varsigma}_{q,l_2 l_1} \right) \right\},
$$

$$
\exp\left\{ -\sum_{q,\{l\},\vartheta} \rho^{*\vartheta}_{q,l_1 l_2} \tilde{V}^{\vartheta}_{q,l_1 l_2,l_3 l_4} \rho^{\vartheta}_{q,l_3 l_4} \right\} =
$$
(46)

$$
\mathcal{D}_\varphi \int D[\varphi^{*\vartheta},\varphi^{\vartheta}] \exp\left\{ \sum_{q,\{l\},\vartheta} \left( \varphi^{*\vartheta}_{q,l_1 l_2} \left[ \left( \tilde{V}^{\vartheta}_q \right)^{-1} \right]_{l_1 l_2,l_3 l_4} \varphi^{\vartheta}_{q,l_3 l_4} - \varphi^{*\vartheta}_{q,l_1 l_2} \rho^{\vartheta}_{q,l_1 l_2} - \rho^{*\vartheta}_{q,l_1 l_2} \varphi^{\vartheta}_{q,l_1 l_2} \right) \right\}.
$$

The terms $\mathcal{D}_f = -\det\left[ \tilde{\varepsilon}_k \right]$ and $\mathcal{D}_\varphi^{-1} = -\sqrt{\det\left[ \tilde{V}_q \right]}$ can be neglected, because they do not affect the calculation of expectation values. After these transformations the lattice action takes the following form

$$
\mathcal{S}' = -\sum_{k,\{l\}} \sum_{\sigma\sigma'} f^*_{k\sigma l} \left[ \tilde{\varepsilon}^{-1}_k \right]^{\sigma\sigma'}_{ll'} f_{k\sigma'l'} + \sum_{k,\sigma,l} \left\{ \left( f^*_{k\sigma l} + \eta^*_{k\sigma l} \right) c_{k\sigma l} + c^*_{k\sigma l} \left( f_{k\sigma l} + \eta_{k\sigma l} \right) \right\} + \mathcal{S}_{\text{imp}}
$$

$$
-\sum_{q,\{l\},\vartheta} \left\{ \varphi^{*\vartheta}_{q,l_1 l_2} \left[ \left( \tilde{V}^{\vartheta}_q \right)^{-1} \right]_{l_1 l_2,l_3 l_4} \varphi^{\vartheta}_{q,l_3 l_4} - \left( \varphi^{*\vartheta}_{q,l_1 l_2} + j^{*\vartheta}_{q,l_1 l_2} \right) \rho^{\vartheta}_{q,l_1 l_2} \right.
$$

$$
-\rho^{*\vartheta}_{q,l_1 l_2} \left( \varphi^{\vartheta}_{q,l_1 l_2} + j^{\vartheta}_{q,l_1 l_2} \right) \left. \right\} - \sum_{q,\{l\},\varsigma} \left\{ \frac{1}{2} \varphi^{\varsigma}_{-q,l_1 l_2} \left[ \left( \tilde{V}^{\varsigma}_q \right)^{-1} \right]_{l_1 l_2,l_3 l_4} \varphi^{\varsigma'}_{q,l_4 l_3} \right.
$$

$$
-\left( \varphi^{\varsigma}_{-q,l_1 l_2} + j^{\varsigma}_{-q,l_1 l_2} \right) \rho^{\varsigma}_{q,l_2 l_1} \left. \right\},
$$
(47)

where we additionally introduced source fields $\eta^{(*)}$ and $j^{(*)}$ for fermionic $c^{(*)}$ and composite $\rho^{(*)}$ variables, respectively. Below, these fields will be used to derive the connection between the dual and lattice quantities. Now, we shift fermionic $f^{(*)} \to \hat{f}^{(*)} = f^{(*)} - \eta^{(*)}$ and bosonic $\varphi^{(*)} \to \hat{\varphi}^{(*)} = \varphi^{(*)} - j^{(*)}$ variables to decouple the source fields from original Grassmann variables $c^{(*)}$. After that the lattice action becomes

$$
\mathcal{S}' = -\sum_{k,\{l\}} \sum_{\sigma\sigma'} \hat{f}^*_{k\sigma l} \left[ \tilde{\varepsilon}^{-1}_k \right]^{\sigma\sigma'}_{ll'} \hat{f}_{k\sigma'l'} + \sum_{k,\sigma,l} \left( f^*_{k\sigma l} c_{k\sigma l} + c^*_{k\sigma l} f_{k\sigma l} \right)
$$

$$
+\sum_{q,\{l\},r} \left( \varphi^{\varsigma}_{-q,l_1 l_2} \rho^{\varsigma}_{q,l_2 l_1} + \varphi^{*\vartheta}_{q,l_1 l_2} \rho^{\vartheta}_{q,l_1 l_2} + \rho^{*\vartheta}_{q,l_1 l_2} \varphi^{\vartheta}_{q,l_1 l_2} \right) + \mathcal{S}_{\text{imp}}
$$

$$
-\frac{1}{2} \sum_{q,\{l\},\varsigma} \hat{\varphi}^{\varsigma}_{-q,l_1 l_2} \left[ \left( \tilde{V}^{\varsigma}_q \right)^{-1} \right]_{l_1 l_2,l_3 l_4} \hat{\varphi}^{\varsigma}_{q,l_4 l_3} - \sum_{q,\{l\},\vartheta} \hat{\varphi}^{*\vartheta}_{q,l_1 l_2} \left[ \left( \tilde{V}^{\vartheta}_q \right)^{-1} \right]_{l_1 l_2,l_3 l_4} \hat{\varphi}^{\vartheta}_{q,l_3 l_4}.
$$
(48)

At this step we can integrate out the reference problem $\mathcal{S}_{\text{imp}}$ by explicitly taking the path integral over fermionic $c^{(*)}$ variables

$$
\int D[c^*,c] \exp\left\{ -\mathcal{S}_{\text{imp}} - \sum_{k,ll',\sigma\sigma'} \left( f^*_{k\sigma l} \left[ B_\nu^{-1} \right]^{\sigma\sigma'}_{ll'} c_{k\sigma'l'} + c^*_{k\sigma l} \left[ B_\nu^{-1} \right]^{\sigma\sigma'}_{ll'} f_{k\sigma'l'} \right) \right.
$$

$$
\left. - \sum_{q,\{l\},\{r\}} \left( \varphi^\varsigma_{-q,l_1l_2} [\alpha_\omega^{-1}]^{\varsigma\varsigma'}_{l_1l_2,l_3l_4} \rho^{\varsigma'}_{q,l_4l_3} + \varphi^{*\vartheta}_{q,l_1l_2} [\alpha_\omega^{-1}]^{\vartheta\vartheta'}_{l_1l_2,l_3l_4} \rho^{\vartheta'}_{q,l_3l_4} + \rho^{*\vartheta}_{q,l_1l_2} [\alpha_\omega^{-1}]^{\vartheta\vartheta'}_{l_1l_2,l_3l_4} \varphi^{\vartheta'}_{q,l_3l_4} \right) \right\} =
$$

$$
\mathcal{Z}_{\text{imp}} \exp\left\{ - \sum_{k,\{l\},\{\sigma\}} f^*_{k\sigma_1 l_1} \left[ B_\nu^{-1} \right]^{\sigma_1\sigma_2}_{l_1l_2} g^{\sigma_2\sigma_3}_{\nu,l_2l_3} \left[ B_\nu^{-1} \right]^{\sigma_3\sigma_4}_{l_3l_4} f_{k\sigma_4 l_4} \right.
$$

$$
- \frac{1}{2} \sum_{q,\{l\},\{\varsigma\}} \varphi^{\varsigma_1}_{-q,l_1l_2} [\alpha_\omega^{-1}]^{\varsigma_1\varsigma_2}_{l_1l_2,l_1'l_2'} \chi^{\varsigma_2\varsigma_3}_{\omega,l_1'l_2',l_3'l_4'} [\alpha_\omega^{-1}]^{\varsigma_3\varsigma_4}_{l_3'l_4',l_3l_4} \varphi^{\varsigma_4}_{l_4l_3}
$$

$$
\left. - \sum_{q,\{l\},\{\vartheta\}} \varphi^{*\vartheta_1}_{q,l_1l_2} [\alpha_\omega^{-1}]^{\vartheta_1\vartheta_2}_{l_1l_2,l_1'l_2'} \chi^{\vartheta_2\vartheta_3}_{\omega,l_1'l_2',l_3'l_4'} [\alpha_\omega^{-1}]^{\vartheta_3\vartheta_4}_{l_3'l_4',l_3l_4} \varphi^{\vartheta_4}_{q,l_3l_4} - \tilde{\mathcal{F}}[f,\varphi] \right\}. \tag{49}
$$

$\mathcal{Z}_{\text{imp}}$, $g^{\sigma\sigma'}_{\nu,ll'}$ and $\chi^{rr'}_{\omega,l_1l_2,l_3l_4}$ are respectively the partition function, the Green's function, and the susceptibility of the reference (impurity) problem. Note that in Eq. (49) we additionally rescaled the fermionic $f^{(*)}$ and bosonic $\varphi^{(*)}$ fields by the parameters $B_\nu^{-1}$ and $\alpha_\omega^{-1}$, respectively. These parameters will be determined below. Following the standard approximation used in dual theories [71–75, 77–80, 82, 83, 93, 94], the interaction part of the action $\tilde{\mathcal{F}}[f,\varphi]$ is truncated at the level of the two-particle correlations functions and reads

$$
\tilde{\mathcal{F}}[f,\varphi] \simeq \sum_{q,\{k\}} \sum_{\{\nu\},\{l\}} \sum_{\{\sigma\},\varsigma/\vartheta} \times
$$

$$
\times \left\{ \frac{1}{4} [\Gamma_{\nu\nu'\omega}]^{\sigma_1\sigma_2\sigma_3\sigma_4}_{l_1l_2l_3l_4} f^*_{k\sigma_1 l_1} f_{k+q,\sigma_2 l_2} f^*_{k'+q,\sigma_4 l_4} f_{k'\sigma_3 l_3} + \Lambda^{\sigma\sigma'\varsigma}_{\nu\omega,l_1,l_2,l_3l_4} f^*_{k\sigma l_1} f_{k+q,\sigma',l_2} \varphi^\varsigma_{q,l_4l_3} \right.
$$

$$
\left. + \frac{1}{2} \left( \Lambda^{\sigma\sigma'\vartheta}_{\nu\omega,l_1,l_2,l_3l_4} f^*_{k\sigma l_1} f^*_{q-k,\sigma',l_2} \varphi^\vartheta_{q,l_3l_4} + \Lambda^{*\sigma\sigma'\vartheta}_{\nu\omega,l_1,l_2,l_3l_4} \varphi^{*\vartheta}_{q,l_3l_4} f_{q-k,\sigma',l_2} f_{k\sigma l_1} \right) \right\}. \tag{50}
$$

Here, $\Gamma$ is the four-point (fermion-fermion) vertex function, which is explicitly defined in Eq. (92). The three-point (fermion-boson) vertex functions $\Lambda^{\sigma\sigma'r}_{\nu\omega}$ have the following form

$$
\Lambda^{\sigma_1\sigma_2\varsigma}_{\nu\omega,l_1,l_2,l_3l_4} = \sum_{\{\sigma'\},\{l'\},\varsigma'} \left\langle c_{\nu\sigma_1'l_1'} c^*_{\nu+\omega,\sigma_2'l_2'} \rho^{\varsigma'}_{-\omega,l_3'l_4'} \right\rangle \left[ B_\nu^{-1} \right]^{\sigma_1'\sigma_1}_{l_1'l_1} \left[ B_{\nu+\omega}^{-1} \right]^{\sigma_2'\sigma_2}_{l_2'l_2} \left[ \alpha_\omega^{-1} \right]^{\varsigma'\varsigma}_{l_3'l_4'l_3l_4}, \tag{51}
$$

$$
\Lambda^{\sigma_1\sigma_2\vartheta}_{\nu\omega,l_1,l_2,l_3l_4} = \sum_{\{\sigma'\},\{l'\},\vartheta'} \left\langle c_{\nu\sigma_1'l_1'} c_{\omega-\nu,\sigma_2'l_2'} \rho^{*\vartheta'}_{\omega,l_3'l_4'} \right\rangle \left[ B_\nu^{-1} \right]^{\sigma_1'\sigma_1}_{l_1'l_1} \left[ B_{\omega-\nu}^{-1} \right]^{\sigma_2'\sigma_2}_{l_2'l_2} \left[ \alpha_\omega^{-1} \right]^{\vartheta'\vartheta}_{l_3'l_4'l_3l_4}, \tag{52}
$$

$$
\Lambda^{*\sigma_1\sigma_2\vartheta}_{\nu\omega,l_1,l_2,l_3l_4} = \sum_{\{\sigma'\},\{l'\},\vartheta'} \left\langle \rho^{\vartheta'}_{\omega,l_3'l_4'} c^*_{\omega-\nu,\sigma_2'l_2'} c^*_{\nu\sigma_1'l_1'} \right\rangle \left[ B_\nu^{-1} \right]^{\sigma_1'\sigma_1}_{l_1'l_1} \left[ B_{\omega-\nu}^{-1} \right]^{\sigma_2'\sigma_2}_{l_2'l_2} \left[ \alpha_\omega^{-1} \right]^{\vartheta'\vartheta}_{l_3'l_4'l_3l_4}. \tag{53}
$$

If the scaling parameters are chosen as $B^{\sigma\sigma'}_{\nu,ll'} = g^{\sigma\sigma'}_{\nu,ll'}$ and

$$
\alpha^{rr'}_{\omega,l_1l_2,l_1'l_2'} = \delta_{l_1,l_1'} \delta_{l_2,l_2'} \delta_{rr'} + \sum_{l_3,l_4} \tilde{U}^r_{\omega,l_1l_2,l_3l_4} \chi^{rr'}_{\omega,l_3l_4,l_1'l_2'}, \tag{54}
$$

where $\tilde{U}^r_{\omega,l_1l_2,l_3l_4} = U^r_{l_1l_2,l_3l_4} + Y^r_{\omega,l_1l_2,l_3l_4}$ is the bare interaction of the reference system and $U^r$ is defined in Eqs. (10)–(13), then $\Lambda^{\sigma\sigma'r}_{\nu\omega}$ takes the usual form of the three-point vertex of the reference system [80, 83, 93, 94, 162]. However, in this case the vertex contains the inverse

of the impurity Green's function $g_{\nu,ll'}^{\sigma\sigma'}$ and consequently has a big numerical noise at large frequencies. At the same time, inverting $\alpha_\omega^{rr'}$ does not lead to the numerical noise due to the presence of a delta-function term in Eq. (54). For this reason in the current implementation of the D-TRILEX method, we keep $\alpha_\omega^{rr'}$ in the form of Eq. (54) and set $B_{\nu,ll'}^{\sigma\sigma'} = \delta_{\sigma\sigma'}\delta_{ll'}$.

It is important to mention that in the most available impurity solvers based on continuous time quantum Monte Carlo method [141–144], the two-particle quantities of the reference system are defined in imaginary time $\tau$ with a different order of the operators compared to our case. For instance, in w2dynamics package [145] the time-ordered two-particle correlation functions are computed as $G_{abcd} = \langle T_\tau c_a c_b^\dagger c_c c_d^\dagger \rangle$ [163]. Here, $T_\tau$ is the imaginary-time ordering operator and the latin indices $a = \{l_a, \sigma_a, \tau_a\}$ describe the orbital, spin, and imaginary-time dependence. Often, the correlation functions are directly measured in the Matsubara space as a function of $\omega$ (or $\nu$ and $\omega$), by performing a Fourier transform on the fly. In order to exploit these correlation functions in D-TRILEX, they have to be recast in the form used in Eqs. (51)–(53) for the vertex functions and in the form of the susceptibility $\chi_{\omega,l_1l_2,l_3l_4}^{\varsigma\varsigma'} = -\langle \rho_{\omega,l_2l_1}^\varsigma \rho_{-\omega,l_3l_4}^{\varsigma'} \rangle$. Taking into account that $\rho_{\omega,l_1l_2}^r = n_{\omega,l_1l_2}^r - \langle n_{\omega,l_1l_2}^r \rangle$, where the densities $n_\omega^r$ are defined in Eqs. (4) and (42), the quantities required for constructing D-TRILEX diagrammatic expansion can be obtained by subtracting the disconnected parts from the corresponding correlation functions of the reference system. Note also that the fermionic operators in $G_{abcd}$ have to be placed in a desired order by applying commutation relations, which may also lead to additional contributions to the disconnected terms.

After integrating out the reference system $\mathcal{S}_{\text{imp}}$, the action takes the form of the dual boson problem [80,83,93,94]

$$
\begin{aligned}
\tilde{\mathcal{S}} = &-\sum_{k,ll'}\sum_{\sigma\sigma'} \hat{f}_{k\sigma l}^* [\tilde{\varepsilon}_k^{-1}]_{ll'}^{\sigma\sigma'} \hat{f}_{k\sigma'l'} + \sum_{k,ll'}\sum_{\sigma\sigma'} f_{k\sigma l}^* g_{\nu,ll'}^{\sigma\sigma'} f_{k\sigma'l'} + \tilde{\mathcal{F}}[f,\varphi] \\
&-\frac{1}{2}\sum_{q,\{l\},\{\varsigma\}}\left\{\hat{\varphi}_{-q,l_1l_2}^\varsigma \left[(\tilde{V}_q^\varsigma)^{-1}\right]_{l_1l_2,l_3l_4} \hat{\varphi}_{q,l_4l_3}^\varsigma - \varphi_{-q,l_1l_2}^{\varsigma_1}[\alpha_\omega^{-1}]_{l_1l_2,l_1'l_2'}^{\varsigma_1\varsigma_2} \chi_{q,l_1'l_2',l_3'l_4'}^{\varsigma_2\varsigma_3}[\alpha_\omega^{-1}]_{l_3'l_4',l_3l_4}^{\varsigma_3\varsigma_4} \varphi_{q,l_4l_3}^{\varsigma_4}\right\} \\
&-\sum_{q,\{l\},\{\vartheta\}}\left\{\hat{\varphi}_{q,l_1l_2}^{*\vartheta} \left[(\tilde{V}_q^\vartheta)^{-1}\right]_{l_1l_2,l_3l_4} \hat{\varphi}_{q,l_3l_4}^\vartheta - \varphi_{q,l_1l_2}^{*\vartheta_1}[\alpha_\omega^{-1}]_{l_1l_2,l_1'l_2'}^{\vartheta_1\vartheta_2} \chi_{\omega,l_1'l_2',l_3'l_4'}^{\vartheta_2\vartheta_3}[\alpha_\omega^{-1}]_{l_3'l_4',l_3l_4}^{\vartheta_3\vartheta_4} \varphi_{q,l_3l_4}^{\vartheta_4}\right\}.
\end{aligned}
\tag{55}
$$

Note that after rescaling bosonic variables one gets $\hat{\varphi}^{(*)} = \varphi^{(*)}\alpha^{-1} - j^{(*)}$. In order to eliminate the four-point vertex function $\Gamma^{ph}$ from the theory, we add and subtract the following terms

$$
\frac{1}{2}\sum_{q,\{l\}}\sum_{\varsigma\varsigma'} \varphi_{-q,l_1l_2}^\varsigma \left[\bar{w}_\omega^{-1}\right]_{l_1l_2,l_3l_4}^{\varsigma\varsigma'} \varphi_{q,l_4l_3}^{\varsigma'} + \sum_{q,\{l\}}\sum_{\vartheta\vartheta'} \varphi_{q,l_1l_2}^{*\vartheta} \left[\bar{w}_\omega^{-1}\right]_{l_1l_2,l_3l_4}^{\vartheta\vartheta'} \varphi_{q,l_3l_4}^{\vartheta'},
\tag{56}
$$

from the dual action (55). At this step, $\bar{w}_\omega$ are introduced as arbitrary quantities. Further, they will be adjusted to obtain the partially bosonized approximation for the four-point vertex function (see Appendix B). The dual boson action becomes

$$
\begin{aligned}
\tilde{\mathcal{S}} = &-\text{Tr}_{\sigma,l}\sum_k \left\{\hat{f}_k^* \tilde{\varepsilon}_k^{-1}\hat{f}_k - f_k^* g_\nu f_k\right\} + \tilde{\mathcal{F}}[f,\varphi] + \frac{1}{2}\text{Tr}_{\varsigma,l}\sum_q \varphi_{-q}\bar{w}_\omega^{-1}\varphi_q + \text{Tr}_{\vartheta,l}\sum_q \varphi_q^*\bar{w}_\omega^{-1}\varphi_q \\
&-\text{Tr}_{\vartheta,l}\sum_q\left\{\varphi_q^*\alpha_\omega^{-1}\left(\tilde{V}_q^{-1} - \chi_\omega + \alpha_\omega\bar{w}_\omega^{-1}\alpha_\omega\right)\alpha_\omega^{-1}\varphi_q + j_q^*\tilde{V}_q^{-1}j_q - \varphi_q^*\alpha_\omega^{-1}\tilde{V}_q^{-1}j_q - j_q^*\tilde{V}_q^{-1}\alpha_\omega^{-1}\varphi_q\right\} \\
&-\text{Tr}_{\varsigma,l}\sum_q\left\{\frac{1}{2}\varphi_{-q}\alpha_\omega^{-1}\left(\tilde{V}_q^{-1} - \chi_\omega + \alpha_\omega\bar{w}_\omega^{-1}\alpha_\omega\right)\alpha_\omega^{-1}\varphi_q + \frac{1}{2}j_{-q}\tilde{V}_q^{-1}j_q - \varphi_{-q}\alpha_\omega^{-1}\tilde{V}_q^{-1}j_q\right\},
\end{aligned}
\tag{57}
$$

where we explicitly isolated the terms that contain the bosonic source fields $j^{(*)}$. To shorten the expression, we omitted band, spin, and channel indices that can be easily restored using the fact that all multiplications in Eq. (57) are performed in the matrix form. Now, we perform the following Hubbard-Stratonovich transformations

$$\exp\left\{\frac{1}{2}\text{Tr}_{\varsigma,l}\sum_q \varphi_{-q}\alpha_\omega^{-1}\left[\tilde{V}_q^{-1}-\chi_\omega+\alpha_\omega\bar{w}_\omega^{-1}\alpha_\omega\right]\alpha_\omega^{-1}\varphi_q\right\}=\mathcal{D}_b\int D[b^\varsigma]\times$$

$$\times\exp\left\{-\text{Tr}_{\varsigma,l}\sum_q\left(\frac{1}{2}b_{-q}\bar{w}_\omega^{-1}\alpha_\omega\left[\tilde{V}_q^{-1}-\chi_\omega+\alpha_\omega\bar{w}_\omega^{-1}\alpha_\omega\right]^{-1}\alpha_\omega\bar{w}_\omega^{-1}b_q-\varphi_{-q}\bar{w}_\omega^{-1}b_q\right)\right\} \quad (58)$$

$$\exp\left\{\text{Tr}_{\vartheta,l}\sum_q \varphi_q^*\alpha_\omega^{-1}\left[\tilde{V}_q^{-1}-\chi_\omega+\alpha_\omega\bar{w}_\omega^{-1}\alpha_\omega\right]\alpha_\omega^{-1}\varphi_q\right\}=\mathcal{D}_b\int D[b^\vartheta]\times$$

$$\times\exp\left\{-\text{Tr}_{\vartheta,l}\sum_q\left(b_q^*\bar{w}_\omega^{-1}\alpha_\omega\left[\tilde{V}_q^{-1}-\chi_\omega+\alpha_\omega\bar{w}_\omega^{-1}\alpha_\omega\right]^{-1}\alpha_\omega\bar{w}_\omega^{-1}b_q-\varphi_q^*\bar{w}_\omega^{-1}b_q-b_q^*\bar{w}_\omega^{-1}\varphi_q\right)\right\},$$

$$(59)$$

where the terms $\mathcal{D}_b^{-1}=\sqrt{\det\left[\bar{w}\alpha^{-1}\left[\tilde{V}^{-1}-\chi+\alpha\bar{w}^{-1}\alpha\right]\alpha^{-1}\bar{w}\right]}$ can again be neglected, because they also do not affect the calculation of correlation functions. The dual action becomes

$$\tilde{\mathcal{S}}'=-\text{Tr}_{\sigma,l}\sum_k\left\{\hat{f}_k^*\tilde{\varepsilon}_k^{-1}\hat{f}_k-f_k^*g_\nu f_k\right\}+\tilde{\mathcal{F}}[f,\varphi]$$

$$+\text{Tr}_{\vartheta,l}\sum_q b_q^*\bar{w}_\omega^{-1}\alpha_\omega\left[\tilde{V}_q^{-1}-\chi_\omega+\alpha_\omega\bar{w}_\omega^{-1}\alpha_\omega\right]^{-1}\alpha_\omega\bar{w}_\omega^{-1}b_q-\text{Tr}_{\vartheta,l}\sum_q j_q^*\tilde{V}_q^{-1}j_q$$

$$+\frac{1}{2}\text{Tr}_{\varsigma,l}\sum_q b_{-q}\bar{w}_\omega^{-1}\alpha_\omega\left[\tilde{V}_q^{-1}-\chi_\omega+\alpha_\omega\bar{w}_\omega^{-1}\alpha_\omega\right]^{-1}\alpha_\omega\bar{w}_\omega^{-1}b_q-\frac{1}{2}\text{Tr}_{\varsigma,l}\sum_q j_{-q}\tilde{V}_q^{-1}j_q$$

$$+\text{Tr}_{\vartheta,l}\sum_q \varphi_q^*\bar{w}_\omega^{-1}\varphi_q-\text{Tr}_{\vartheta,l}\sum_q\left\{\varphi_q^*\bar{w}_\omega^{-1}\left(b_q-\bar{w}_\omega\alpha_\omega^{-1}\tilde{V}_q^{-1}j_q\right)+\left(b_q^*-j_q^*\tilde{V}_q^{-1}\alpha_\omega^{-1}\bar{w}_\omega\right)\bar{w}_\omega^{-1}\varphi_q\right\}$$

$$+\frac{1}{2}\text{Tr}_{\varsigma,l}\sum_q \varphi_{-q}\bar{w}_\omega^{-1}\varphi_q-\text{Tr}_{\varsigma,l}\sum_q \varphi_{-q}\bar{w}_\omega^{-1}\left(b_q-\bar{w}_\omega\alpha_\omega^{-1}\tilde{V}_q^{-1}j_q\right). \quad (60)$$

We shift bosonic variables as $b^{(*)}\to\hat{b}^{(*)}=b^{(*)}+\bar{w}\alpha^{-1}\tilde{V}^{-1}j^{(*)}$ to decouple the sources $j^{(*)}$ from

the dual bosonic fields $\varphi^{(*)}$. After that the fields $\varphi^{(*)}$ can be integrated out as

$$
\int D[\varphi^{\varsigma}] \exp\left\{ -\frac{1}{2} \mathrm{Tr}_{\varsigma,l} \sum_q \varphi_{-q} \bar{w}_\omega^{-1} \varphi_q + \mathrm{Tr}_{\varsigma,l} \sum_q \left( b_{-q} \bar{w}_\omega^{-1} - \sum_{k,\sigma\sigma'} f_{k\sigma}^* f_{k+q,\sigma'} \Lambda_{\nu\omega}^{\sigma\sigma'} \right) \varphi_q \right\} =
$$

$$
\mathcal{Z}_\varphi \exp\left\{ \frac{1}{2} \mathrm{Tr}_{\varsigma,l} \sum_q b_{-q} \bar{w}_\omega^{-1} b_q - \mathrm{Tr}_{\varsigma,l} \sum_{q,k} \sum_{\sigma\sigma'} \Lambda_{\nu\omega}^{\sigma\sigma'} f_{k\sigma}^* f_{k+q,\sigma'} b_q \right.
$$

$$
\left. + \frac{1}{2} \mathrm{Tr}_{\varsigma,l} \sum_{q,\{\sigma\}} \sum_{k,k'} \Lambda_{\nu\omega}^{\sigma_1\sigma_2} \bar{w}_\omega \Lambda_{\nu'+\omega,-\omega}^{\sigma_4\sigma_3} f_{k\sigma_1}^* f_{k+q,\sigma_2} f_{k'+q,\sigma_4}^* f_{k'\sigma_3} \right\}, \tag{61}
$$

$$
\int D[\varphi^{*\vartheta}, \varphi^\vartheta] \exp\left\{ -\mathrm{Tr}_{\vartheta,l} \sum_q \varphi_q^* \bar{w}_\omega^{-1} \varphi_q + \mathrm{Tr}_{\vartheta,l} \sum_q \left( b_q^* \bar{w}_\omega^{-1} \varphi_q + \varphi_q^* \bar{w}_\omega^{-1} b_q \right) \right.
$$

$$
\left. -\frac{1}{2} \mathrm{Tr}_{\vartheta,l} \sum_{q,k} \sum_{\sigma\sigma'} \left( f_{k\sigma}^* f_{q-k,\sigma'}^* \Lambda_{\nu\omega}^{\sigma\sigma'} \varphi_q + \varphi_q^* \Lambda_{\nu\omega}^{*\sigma\sigma'} f_{q-k,\sigma'} f_{k\sigma} \right) \right\} =
$$

$$
\mathcal{Z}_\varphi \exp\left\{ \mathrm{Tr}_{\vartheta,l} \sum_q b_q^* \bar{w}_\omega^{-1} b_q - \frac{1}{2} \mathrm{Tr}_{\vartheta,l} \sum_{q,k} \sum_{\sigma\sigma'} \left( f_{k\sigma}^* f_{q-k,\sigma'}^* \Lambda_{\nu\omega}^{\sigma\sigma'} b_q + b_q^* \Lambda_{\nu\omega}^{*\sigma\sigma'} f_{q-k,\sigma'} f_{k\sigma} \right) \right.
$$

$$
\left. + \frac{1}{4} \mathrm{Tr}_{\vartheta,l} \sum_{q,\{\sigma\}} \sum_{k,k'} \Lambda_{\nu\omega}^{\sigma_1\sigma_2} \bar{w}_\omega \Lambda_{\nu'\omega}^{\sigma_3\sigma_4} f_{k\sigma_1}^* f_{q-k,\sigma_2}^* f_{q-k',\sigma_4} f_{k'\sigma_3} \right\}, \tag{62}
$$

where $\mathcal{Z}_\varphi$ is a partition function of the Gaussian part of the bosonic action. As discussed in Appendix B and Refs. [93,94], quartic terms that appear at the last lines of Eqs. (61) and (62) approximately cancel the fermion-fermion ($\Gamma$) part of the interaction (50) if the $\bar{w}_\omega^{rr'}$ quantities are taken in the form of Eq. (96). As the result, the problem reduces to a partially bosonized dual action

$$
\tilde{\mathcal{S}}_{fb} = -\mathrm{Tr}_{\sigma,l} \sum_k \left\{ \hat{f}_k^* \tilde{\varepsilon}_k^{-1} \hat{f}_k - f_k^* g_\nu f_k \right\}
$$

$$
+ \frac{1}{2} \mathrm{Tr}_{\varsigma,l} \sum_q \hat{b}_{-q} \bar{w}_\omega^{-1} \alpha_\omega \left[ \tilde{V}_q^{-1} - \chi_\omega + \alpha_\omega \bar{w}_\omega^{-1} \alpha_\omega \right]^{-1} \alpha_\omega \bar{w}_\omega^{-1} \hat{b}_q - \frac{1}{2} \mathrm{Tr}_{\varsigma,l} \sum_q j_{-q} \tilde{V}_q^{-1} j_q
$$

$$
- \frac{1}{2} \mathrm{Tr}_{\varsigma,l} \sum_q b_{-q} \bar{w}_\omega^{-1} b_q + \mathrm{Tr}_{\varsigma,l} \sum_{q,k} \sum_{\sigma\sigma'} \Lambda_{\nu\omega}^{\sigma\sigma'} f_{k\sigma}^* f_{k+q,\sigma'} b_q
$$

$$
+ \mathrm{Tr}_{\vartheta,l} \sum_q \hat{b}_q^* \bar{w}_\omega^{-1} \alpha_\omega \left[ \tilde{V}_q^{-1} - \chi_\omega + \alpha_\omega \bar{w}_\omega^{-1} \alpha_\omega \right]^{-1} \alpha_\omega \bar{w}_\omega^{-1} \hat{b}_q - \mathrm{Tr}_{\vartheta,l} \sum_q j_q^* \tilde{V}_q^{-1} j_q
$$

$$
- \mathrm{Tr}_{\vartheta,l} \sum_q b_q^* \bar{w}_\omega^{-1} b_q + \frac{1}{2} \mathrm{Tr}_{\vartheta,l} \sum_{q,k} \sum_{\sigma\sigma'} \left( f_{k\sigma}^* f_{q-k,\sigma'}^* \Lambda_{\nu\omega}^{\sigma\sigma'} b_q + b_q^* \Lambda_{\nu\omega}^{*\sigma\sigma'} f_{q-k,\sigma'} f_{k\sigma} \right), \tag{63}
$$

that upon neglecting fermionic $\eta^{(*)}$ and bosonic $j^{(*)}$ source takes the simple form shown in Eq. (5) of the main text. The bare fermionic Green's function for this action is defined in Eq. (7). The bare dual bosonic propagator reads

$$
\tilde{\mathcal{W}}_{q,l_1l_2,l_3l_4}^{r_1r_2} = \sum_{\{r'\},\{l'\}} \alpha_{\omega,l_1l_2,l_1'l_2'}^{r_1r_1'} \left[ \left( \tilde{V}_q^{-1} - \chi_\omega \right)^{-1} \right]_{l_1'l_2',l_3'l_4'}^{r_1'r_2'} \alpha_{\omega,l_3'l_4',l_3l_4}^{r_2'r_2} + \bar{w}_{\omega,l_1l_2,l_3l_4}^{r_1r_2}. \tag{64}
$$

Substituting the explicit expression (96) for the $\bar{w}_\omega^{rr'}$ quantities leads for the final form for the bare bosonic propagator shown in Eq. (8).

## A.2 Physical quantities from the dual space

The source fields introduced in the previous Appendix allows one to derive expressons for the correlation functions of the initial lattice problem (1) even though the original Grassman variables $c^{(*)}$ have been already integrated out. By definition, the fermionic Green's function can be found as

$$G^{\sigma\sigma'}_{k,ll'} = -\langle c_{k\sigma l} c^*_{k\sigma' l'}\rangle = -\frac{1}{\mathcal{Z}}\frac{\partial^2 \mathcal{Z}}{\partial \eta^*_{k\sigma l}\partial \eta_{k\sigma' l'}}, \qquad (65)$$

where $\mathcal{Z}$ is the partition function of the problem. Taking into account that the source fields enter only the partially bosonized dual action (63), the lattice Green's function becomes

$$
\begin{aligned}
G^{\sigma\sigma'}_{k,ll'} &= -\left[\tilde{\varepsilon}^{-1}_k\right]^{\sigma\sigma'}_{ll'} - \sum_{l_1 l_2}\sum_{\sigma_1\sigma_2}\left[\tilde{\varepsilon}^{-1}_k\right]^{\sigma\sigma_1}_{ll_1}\langle f_{k\sigma_1 l_1}f^*_{k\sigma_2 l_2}\rangle_{\tilde{\mathcal{S}}_{fb}}\left[\tilde{\varepsilon}^{-1}_k\right]^{\sigma_2\sigma'}_{l_2 l'} \\
&= -\left[\tilde{\varepsilon}^{-1}_k\right]^{\sigma\sigma'}_{ll'} + \sum_{l_1 l_2}\sum_{\sigma_1\sigma_2}\left[\tilde{\varepsilon}^{-1}_k\right]^{\sigma\sigma_1}_{ll_1}\tilde{G}^{\sigma_1\sigma_2}_{kl_1 l_2}\left[\tilde{\varepsilon}^{-1}_k\right]^{\sigma_2\sigma'}_{l_2 l'}.
\end{aligned}
\qquad (66)
$$

Using the Dyson equation for the dressed dual fermionic Green's function (15) and substituting the explicit form for the bare dual fermionic Green's function (7) allows to get the following relation for the lattice Green's function

$$\left[G^{-1}_k\right]^{\sigma\sigma'}_{ll'} = \left[(g_\nu + \tilde{\Sigma}_k)^{-1}\right]^{\sigma\sigma'}_{ll'} + \Delta^{\sigma\sigma'}_{\nu,ll'} - \varepsilon^{\sigma\sigma'}_{\mathbf{k},ll'}, \qquad (67)$$

shown in Eq. (20) of the main text. Expression (21) for the lattice self-energy can then be obtained straightforwardly using the standard Dyson equation for the lattice Green's function (22).

The lattice susceptibilities can be obtained in a similar way as

$$X^{\varsigma\varsigma'}_{q,l_1 l_2,l_3 l_4} = -\langle \rho^\varsigma_{q,l_2 l_1}\rho^{\varsigma'}_{-q,l_3 l_4}\rangle = -\frac{1}{\mathcal{Z}}\frac{\partial^2 \mathcal{Z}}{\partial j^\varsigma_{-q,l_1 l_2}\partial j^{\varsigma'}_{q,l_4 l_3}}, \qquad (68)$$

$$X^{\vartheta\vartheta'}_{q,l_1 l_2,l_3 l_4} = -\langle \rho^\vartheta_{q,l_1 l_2}\rho^{*\vartheta'}_{q,l_3 l_4}\rangle = -\frac{1}{\mathcal{Z}}\frac{\partial^2 \mathcal{Z}}{\partial j^{*\vartheta}_{q,l_1 l_2}\partial j^{\vartheta'}_{q,l_3 l_4}}. \qquad (69)$$

The part of the action that depends on the bosonic sources is in the following form

$$
\begin{aligned}
\tilde{\mathcal{S}}_j &= \frac{1}{2}\operatorname{Tr}_{\varsigma,l}\sum_q j_{-q}C_q j_q + \operatorname{Tr}_{\varsigma,l}\sum_q j_{-q}R_q b_q \\
&+ \operatorname{Tr}_{\vartheta,l}\sum_q j^*_q C_q j_q + \operatorname{Tr}_{\vartheta,l}\sum_q\left(j^*_q R_q b_q + j^*_q R_q b_q\right),
\end{aligned}
\qquad (70)
$$

where the terms $C_q$ and $R_q$ explicitly read

$$
\begin{aligned}
C_q &= \tilde{V}^{-1}_q\left[\tilde{V}^{-1}_q - \chi_\omega + \alpha_\omega \bar{w}^{-1}_\omega \alpha_\omega\right]^{-1}\tilde{V}^{-1}_q - \tilde{V}^{-1}_q, \\
R_q &= \tilde{V}^{-1}_q\left[\tilde{V}^{-1}_q - \chi_\omega + \alpha_\omega \bar{w}^{-1}_\omega \alpha_\omega\right]^{-1}\alpha_\omega \bar{w}^{-1}_\omega.
\end{aligned}
\qquad (71)
$$

Therefore, the susceptibility takes the form

$$X^{r_1 r_2}_{q,l_1 l_2,l_3 l_4} = C^{r_1\varsigma_2}_{q,l_1 l_2,l_3 l_4} + \sum_{\{l'\},\{r'\}}R^{r_1 r'_1}_{q,l_1 l_2,l'_1 l'_2}\tilde{W}^{r'_1 r'_2}_{q,l'_1 l'_2,l'_3 l'_4}R^{r'_2 r_2}_{q,l'_3 l'_4,l_3 l_4}. \qquad (72)$$

Importantly, the terms $C_q$ and $R_q$ are not divergent. For this reason, the divergence of the susceptibility $X_q$ and of the renormalized dual interaction $\tilde{W}_q$ occurs at the same time. After some algebra, the expression (72) can be drastically simplified, and the inverse susceptibility takes the form of a standard Dyson equation

$$\left[X_q^{-1}\right]_{l_1 l_2, l_3 l_4}^{rr'} = \left[\Pi_q^{-1}\right]_{l_1 l_2, l_3 l_4}^{rr'} - \left(U_{l_1 l_2, l_3 l_4}^{r} + V_{q, l_1 l_2, l_3 l_4}^{r}\right)\delta_{rr'}, \tag{73}$$

with the following polarization operator of the lattice problem

$$\Pi_{q, l_1 l_2, l_3 l_4}^{rr'} = \Pi_{\omega, l_1 l_2, l_3 l_4}^{\text{imp}\,rr'} + \sum_{l'l'', r_1} \tilde{\Pi}_{q, l_1 l_2, l'l''}^{rr_1}\left[\left(\mathbb{1} + \bar{u}\cdot\tilde{\Pi}_q\right)^{-1}\right]_{l'l'', l_3 l_4}^{r_1 r'}. \tag{74}$$

The $\bar{u}_{l_1 l_2, l_3 l_4}^{r}$ term is defined in Appendix B. It is important to emphasize that the derived relations for the Green's function (67) and the susceptibility (73) do not depend on the particular approximation used to obtain the dual self-energy $\tilde{\Sigma}$ and the dual polarization operator $\tilde{\Pi}$.

### A.3 General form for D-TRILEX diagrams

In the D-TRILEX approach the self-energy $\tilde{\Sigma}$ and the polarization operator $\tilde{\Pi}$ are obtained self-consistently from the following functional that corresponds to the partially bosonized dual action (5)

$$\begin{aligned}
\Phi[\tilde{G}, \tilde{W}, \Lambda] = &-\frac{1}{2}\sum_{q,k}\sum_{\{l\},\{\sigma\}}\sum_{\varsigma\varsigma'}\Lambda_{\nu\omega, l_1, l_2, l_3 l_4}^{\sigma_1 \sigma_2 \varsigma}\tilde{G}_{k+q, l_2 l_8}^{\sigma_2 \sigma_8}\tilde{G}_{k, l_7 l_1}^{\sigma_7 \sigma_1}\tilde{W}_{q, l_3 l_4, l_5 l_6}^{\varsigma\varsigma'}\Lambda_{\nu+\omega, -\omega, l_8, l_7, l_6 l_5}^{\sigma_8 \sigma_7 \varsigma'} \\
&+\frac{1}{2}\sum_{k,k'}\sum_{\{l\},\{\sigma\}}\sum_{\varsigma\varsigma'}\Lambda_{\nu, \omega=0, l_1, l_2, l_3 l_4}^{\sigma_1 \sigma_2 \varsigma}\tilde{G}_{k, l_2 l_1}^{\sigma_2 \sigma_1}\tilde{G}_{k', l_7 l_8}^{\sigma_7 \sigma_8}\tilde{\mathcal{W}}_{q=0, l_3 l_4, l_5 l_6}^{\varsigma\varsigma'}\Lambda_{\nu', \omega=0, l_8, l_7, l_6 l_5}^{\sigma_8 \sigma_7 \varsigma'}.
\end{aligned} \tag{75}$$

Note that the contribution of the particle-particle $\vartheta$ channel to the introduced functional is neglected, because it is negligibly small in the D-TRILEX approximation [94]. The self-energy and the polarization operator can be found by varying the functional as

$$\tilde{\Sigma}_{k, ll'}^{\sigma\sigma'} = \left.\frac{\partial \Phi[\tilde{G}, \tilde{W}, \Lambda]}{\partial \tilde{G}_{k, l'l}^{\sigma'\sigma}}\right|_{\tilde{W},\Lambda}, \qquad \tilde{\Pi}_{q, l_1 l_2, l_3 l_4}^{\varsigma\varsigma'} = -2\left.\frac{\partial \Phi[\tilde{G}, \tilde{W}, \Lambda]}{\partial \tilde{W}_{q,, l_3 l_4, l_1 l_2}^{\varsigma'\varsigma}}\right|_{\tilde{G},\Lambda}. \tag{76}$$

This results in the following explicit expressions shown in Eqs. (18), (17), and (19) of the main text and illustrated in Fig. 1

$$\begin{aligned}
\tilde{\Sigma}_{k, l_1 l_7}^{\sigma_1 \sigma_7} = &-\sum_{q,\varsigma\varsigma'}\sum_{\{l\},\{\sigma\}}\Lambda_{\nu\omega, l_1, l_2, l_3 l_4}^{\sigma_1 \sigma_2 \varsigma}\tilde{G}_{k+q, l_2 l_8}^{\sigma_2 \sigma_8}\tilde{W}_{q, l_3 l_4, l_5 l_6}^{\varsigma\varsigma'}\Lambda_{\nu+\omega, -\omega, l_8, l_7, l_6 l_5}^{\sigma_8 \sigma_7 \varsigma'} \\
&+\sum_{k',\varsigma\varsigma'}\sum_{\{l\},\{\sigma\}}\Lambda_{\nu, \omega=0, l_1, l_7, l_3 l_4}^{\sigma_1 \sigma_7 \varsigma}\tilde{\mathcal{W}}_{q=0, l_3 l_4, l_5 l_6}^{\varsigma\varsigma'}\Lambda_{\nu', \omega=0, l_8, l_2, l_6 l_5}^{\sigma_8 \sigma_2 \varsigma'}\tilde{G}_{k', l_2 l_8}^{\sigma_2 \sigma_8},
\end{aligned} \tag{77}$$

$$\tilde{\Pi}_{q, l_1 l_2, l_7 l_8}^{\varsigma\varsigma'} = \sum_{k}\sum_{\{l\},\{\sigma\}}\Lambda_{\nu+\omega, -\omega, l_4, l_3, l_2 l_1}^{\sigma_4 \sigma_3 \varsigma}\tilde{G}_{k, l_3 l_5}^{\sigma_3 \sigma_5}\tilde{G}_{k+q, l_6 l_4}^{\sigma_6 \sigma_4}\Lambda_{\nu\omega, l_5, l_6, l_7 l_8}^{\sigma_5 \sigma_6 \varsigma'}. \tag{78}$$

Here, $\tilde{G}_k^{\sigma\sigma'}$ and $W_q^{\varsigma\varsigma'}$ are the dressed fermionic and bosonic propagators of the partially bosonized dual problem (5) that can be found using Dyson equations (15) and (16), respectively.

# B  Partially bosonized approximation for the four-point vertex

In this Appendix we show the partially bosonized approximation for the four-point vertex function of the reference system for the multi-band case. This approximation allows us to eliminate the four-point vertex from the theory by introducing $\bar{w}$ terms in the lattice action according to Eq. (56). First, we rewrite the interaction of the reference system (41) in the particle-hole representation

$$\frac{1}{2}\sum_{\omega,\{\nu\}}\sum_{\{l\},\{\sigma\}}U^{pp}_{l_1l_2l_3l_4}c^*_{\nu\sigma l_1}c^*_{\omega-\nu,\sigma'l_2}c_{\omega-\nu',\sigma'l_4}c_{\nu'\sigma l_3}=$$

$$\frac{1}{2}\sum_{\omega,\{\nu\}}\sum_{\{l\},\{\sigma\}}U^{pp}_{l_1l_2l_3l_4}c^*_{\nu\sigma l_1}c_{\nu'\sigma l_3}c^*_{\omega-\nu,\sigma'l_2}c_{\omega-\nu',\sigma'l_4}-\frac{1}{2}\sum_{\nu,\{l\},\sigma}U^{pp}_{l_1l_2l_3l_4}c^*_{\nu\sigma l_1}c_{\nu\sigma l_4}\delta_{l_2l_3}=$$

$$\frac{1}{2}\sum_{\omega,\{\nu\}}\sum_{\{l\},\{\sigma\}}U^{ph}_{l_1l_2l_3l_4}c^*_{\nu\sigma l_1}c_{\nu+\omega,\sigma l_2}c^*_{\nu'+\omega,\sigma'l_4}c_{\nu'\sigma'l_3}-\frac{1}{2}\sum_{\nu,\{l\},\sigma}U^{pp}_{l_1l_2l_3l_4}c^*_{\nu\sigma l_1}c_{\nu\sigma l_4}\delta_{l_2l_3}, \tag{79}$$

which results in the following relation $U^{ph}_{l_1l_2l_3l_4}=U^{pp}_{l_1l_4l_3l_2}$. Now, let us antisymmetrize the interaction as (quadratic terms in Grassmann $c^{(*)}$ variables are neglected for simplicity)

$$\frac{1}{2}\sum_{\omega,\{\nu\}}\sum_{\{l\},\{\sigma\}}U^{ph}_{l_1l_2l_3l_4}c^*_{\nu\sigma l_1}c_{\nu+\omega,\sigma l_2}c^*_{\nu'+\omega,\sigma'l_4}c_{\nu'\sigma'l_3}=$$

$$\frac{1}{8}\sum_{\omega,\{\nu\}}\sum_{\{l\},\{\sigma\}}\Gamma^{0d}_{l_1l_2l_3l_4}\left(c^*_{\nu\sigma l_1}c_{\nu+\omega,\sigma l_2}\right)\left(c^*_{\nu'+\omega,\sigma',l_4}c_{\nu',\sigma',l_3}\right)+$$

$$\frac{1}{8}\sum_{\omega,\{\nu\}}\sum_{\{l\},\{\sigma\}}\Gamma^{0m}_{l_1l_2l_3l_4}\left(c^*_{\nu\sigma_1 l_1}\vec{\sigma}_{\sigma_1\sigma_2}c_{\nu+\omega,\sigma_2 l_2}\right)\left(c^*_{\nu'+\omega,\sigma_4 l_4}\vec{\sigma}_{\sigma_4\sigma_3}c_{\nu'\sigma_3 l_3}\right), \tag{80}$$

in order to obtain the expressions for the bare four-point vertex functions of the reference system in the particle-hole channel

$$\Gamma^{0d}_{l_1l_2l_3l_4}=2U^{ph}_{l_1l_2l_3l_4}-U^{ph}_{l_1l_3l_2l_4}=2U^{pp}_{l_1l_4l_3l_2}-U^{pp}_{l_1l_4l_3l_2}, \qquad \Gamma^{0m}_{l_1l_2l_3l_4}=-U^{ph}_{l_1l_3l_2l_4}=-U^{pp}_{l_1l_4l_3l_2}. \tag{81}$$

Alternatively, the interaction can also be antisymmetrized in the particle-particle channel

$$\frac{1}{2}\sum_{\omega,\{\nu\}}\sum_{\{l\},\{\sigma\}}U^{ph}_{l_1l_2l_3l_4}c^*_{\nu\sigma l_1}c_{\nu+\omega,\sigma l_2}c^*_{\nu'+\omega,\sigma'l_4}c_{\nu'\sigma'l_3}=$$

$$\frac{1}{4}\sum_{\omega,\{\nu\},\{l\}}\Gamma^{0s}_{l_1l_2l_3l_4}\left(c^*_{\nu\uparrow l_1}c^*_{\omega-\nu\downarrow l_2}-c^*_{\nu\downarrow l_1}c^*_{\omega-\nu,\uparrow l_2}\right)\left(c_{\omega-\nu',\downarrow l_4}c_{\nu'\uparrow l_3}-c_{\omega-\nu',\uparrow l_4}c_{\nu'\downarrow l_3}\right)+$$

$$\frac{1}{4}\sum_{\omega,\{\nu\},\{l\}}\Gamma^{0t}_{l_1l_2l_3l_4}\left(c^*_{\nu\uparrow l_1}c^*_{\omega-\nu,\downarrow l_2}+c^*_{\nu\downarrow l_1}c^*_{\omega-\nu,\uparrow l_2}\right)\left(c_{\omega-\nu',\downarrow l_4}c_{\nu'\uparrow l_3}+c_{\omega-\nu',\uparrow l_4}c_{\nu'\downarrow l_3}\right)+$$

$$\frac{1}{2}\sum_{\omega,\{\nu\},\{l\}}\Gamma^{0t}_{l_1l_2l_3l_4}\left\{\left(c^*_{\nu\uparrow l_1}c^*_{\omega-\nu,\uparrow l_2}\right)\left(c_{\omega-\nu',\uparrow l_4}c_{\nu'\uparrow l_3}\right)+\left(c^*_{\nu\downarrow l_1}c^*_{\omega-\nu,\downarrow l_2}\right)\left(c_{\omega-\nu',\downarrow l_4}c_{\nu'\downarrow l_3}\right)\right\}, \tag{82}$$

which gives the corresponding bare vertex functions

$$\Gamma^{0s}_{l_1l_2l_3l_4}=\frac{1}{2}\left(U^{ph}_{l_1l_3l_4l_2}+U^{ph}_{l_1l_4l_3l_2}\right)=\frac{1}{2}\left(U^{pp}_{l_1l_2l_3l_4}+U^{pp}_{l_1l_2l_4l_3}\right), \tag{83}$$

$$\Gamma^{0t}_{l_1l_3l_4l_2}=\frac{1}{2}\left(U^{ph}_{l_1l_3l_4l_2}-U^{ph}_{l_1l_4l_3l_2}\right)=\frac{1}{2}\left(U^{pp}_{l_1l_2l_3l_4}-U^{pp}_{l_1l_2l_4l_3}\right). \tag{84}$$

Note that the obtained expressions (81), (83) and (84) coincide with the standard definition for the vertex functions in FLEX approach (see e.g. Ref. [125]).

One can also formally rewrite the interaction of the reference system (41) in the channel representation as

$$\frac{1}{2} \sum_{\omega,\{\nu\}} \sum_{\{l\},\{\sigma\}} U^{ph}_{l_1 l_2 l_3 l_4} c^*_{\nu\sigma l_1} c_{\nu+\omega,\sigma l_2} c^*_{\nu'+\omega,\sigma' l_4} c_{\nu'\sigma' l_3} +$$

$$\frac{1}{2} \sum_{\omega,\{l\},\varsigma} Y^\varsigma_{\omega,l_1 l_2,l_3 l_4} \rho^\varsigma_{-\omega,l_1 l_2} \rho^\varsigma_{\omega,l_4 l_3} + \sum_{\omega,\{l\},\vartheta} Y^\vartheta_{\omega,l_1 l_2,l_3 l_4} \rho^{*\vartheta}_{\omega,l_1 l_2} \rho^\vartheta_{\omega,l_3 l_4} =$$

$$\frac{1}{2} \sum_{\omega,\{l\},\varsigma} \tilde{U}^\varsigma_{\omega,l_1 l_2,l_3 l_4} \rho^\varsigma_{-\omega,l_1 l_2} \rho^\varsigma_{\omega,l_4 l_3} + \sum_{q,\{l\},\vartheta} \tilde{U}^\vartheta_{\omega,l_1 l_2,l_3 l_4} \rho^{*\vartheta}_{q,l_1 l_2} \rho^\vartheta_{q,l_3 l_4}. \tag{85}$$

In order to determine the bare interaction $\tilde{U}^r_{\omega,l_1 l_2,l_3 l_4} = U^r_{l_1 l_2,l_3 l_4} + Y^r_{\omega,l_1 l_2,l_3 l_4}$ of the reference system for every $r$ channel, we antisymmetrize the expression (85). Following the works [93, 94], the bare four-point vertex functions (81) take the following form

$$\left[\Gamma^{0d}_{\nu\nu'\omega}\right]_{l_1 l_2 l_3 l_4} = 2\tilde{U}^d_{\omega,l_1 l_2,l_3 l_4} - \tilde{U}^d_{\nu'-\nu,l_1 l_3,l_2 l_4} - 3\tilde{U}^m_{\nu'-\nu,l_1 l_3,l_2 l_4} + \tilde{U}^s_{\omega+\nu+\nu',l_1 l_4,l_3 l_2} - 3\tilde{U}^t_{\omega+\nu+\nu',l_1 l_4,l_3 l_2}$$
$$= 2U^{ph}_{l_1 l_2 l_3 l_4} - U^{ph}_{l_1 l_3 l_2 l_4} + o(Y), \tag{86}$$

$$\left[\Gamma^{0m}_{\nu\nu'\omega}\right]_{l_1 l_2 l_3 l_4} = 2\tilde{U}^m_{\omega,l_1 l_2,l_3 l_4} + \tilde{U}^m_{\nu'-\nu,l_1 l_3,l_2 l_4} - \tilde{U}^d_{\nu'-\nu,l_1 l_3,l_2 l_4} - \tilde{U}^s_{\omega+\nu+\nu',l_1 l_4,l_3 l_2} - \tilde{U}^t_{\omega+\nu+\nu',l_1 l_4,l_3 l_2}$$
$$= -U^{ph}_{l_1 l_3 l_2 l_4} + o(Y). \tag{87}$$

Using the idea of Ref. [93] we associate the static parts $U^{ph}$ of the vertex functions (86) and (87) with the longitudinal contributions $\tilde{U}^\varsigma_\omega$. After doing that, we immediately get

$$U^d_{l_1 l_2,l_3 l_4} = \frac{1}{2}\left(2U^{ph}_{l_1 l_2 l_3 l_4} - U^{ph}_{l_1 l_3 l_2 l_4}\right) = \frac{1}{2}\left(2U^{pp}_{l_1 l_4 l_2 l_3} - U^{pp}_{l_1 l_4 l_3 l_2}\right), \tag{88}$$

$$U^m_{l_1 l_2,l_3 l_4} = -\frac{1}{2}U^{ph}_{l_1 l_3 l_2 l_4} = -\frac{1}{2}U^{pp}_{l_1 l_4 l_3 l_2}. \tag{89}$$

The same procedure can be performed for the particle-particle channel. Since the variables $\rho^{(*)\vartheta}_\omega$ are already defined in the antisymmetrized form (42), the bare interaction in the particle-particle channel simply coincides with the bare vertex defined in Eqs. (83) and (83)

$$U^s_{l_1 l_2,l_3 l_4} = \frac{1}{2}\left(U^{ph}_{l_1 l_3 l_4 l_2} + U^{ph}_{l_1 l_4 l_3 l_2}\right) = \frac{1}{2}\left(U^{pp}_{l_1 l_2 l_3 l_4} + U^{pp}_{l_1 l_2 l_4 l_3}\right), \tag{90}$$

$$U^t_{l_1 l_2,l_3 l_4} = \frac{1}{2}\left(U^{ph}_{l_1 l_3 l_4 l_2} - U^{ph}_{l_1 l_4 l_3 l_2}\right) = \frac{1}{2}\left(U^{pp}_{l_1 l_2 l_3 l_4} - U^{pp}_{l_1 l_2 l_4 l_3}\right). \tag{91}$$

As in previous works on D-TRILEX method [93, 94], the exact four-point vertex function of the reference problem is defined as follows

$$\left[\Gamma_{\nu\nu'\omega}\right]^{\sigma_1\sigma_2\sigma_3\sigma_4}_{l_1 l_2 l_3 l_4} = \sum_{\{l'\},\{\sigma'\}} \left\langle c_{\nu\sigma'_1 l'_1} c^*_{\nu+\omega,\sigma'_2 l'_2} c^*_{\nu'\sigma'_3 l'_3} c_{\nu'+\omega,\sigma'_4 l'_4}\right\rangle_{\text{connected}} \times$$
$$\times \left[B^{-1}_\nu\right]^{\sigma'_1\sigma_1}_{l'_1 l_1} \left[B^{-1}_{\nu+\omega}\right]^{\sigma'_2\sigma_2}_{l'_2 l_2} \left[B^{-1}_{\nu'}\right]^{\sigma'_3\sigma_3}_{l'_3 l_3} \left[B^{-1}_{\nu'+\omega}\right]^{\sigma'_4\sigma_4}_{l'_4 l_4}. \tag{92}$$

The conventional definition for the four-point vertex corresponds to the choice $B^{\sigma\sigma'}_{\nu,ll'} = g^{\sigma\sigma'}_{\nu,ll'}$. Following the derivation presented in Refs. [93, 94], in the multi-band case the partially approximation for the fermion-fermion vertex function reads

$$\left[\Gamma_{\nu\nu'\omega}\right]^{\sigma_1\sigma_2\sigma_3\sigma_4}_{l_1 l_2 l_3 l_4} = \sum_{\{r\}} \left\{\left[M^{\varsigma\varsigma'}_{\nu\nu'\omega}\right]^{\sigma_1\sigma_2\sigma_3\sigma_4}_{l_1 l_2 l_3 l_4} - \left[M^{\varsigma\varsigma'}_{\nu,\nu+\omega,\nu'-\nu}\right]^{\sigma_1\sigma_3\sigma_2\sigma_4}_{l_1 l_3 l_2 l_4} - \left[M^{\vartheta\vartheta'}_{\nu,\nu',\omega+\nu+\nu'}\right]^{\sigma_1\sigma_4\sigma_3\sigma_2}_{l_1 l_4 l_3 l_2}\right\}, \tag{93}$$

where the partially bosonized collective electronic fluctuations in different channels are

$$\left[M^{\varsigma\varsigma'}_{\nu\nu'\omega}\right]^{\sigma_1\sigma_2\sigma_3\sigma_4}_{l_1l_2l_3l_4} = \sum_{\{l'\}} \Lambda^{\sigma_1\sigma_2\varsigma}_{\nu\omega,l_1,l_2,l_1'l_2'} \bar{w}^{\varsigma\varsigma'}_{\omega,l_1'l_2',l_3'l_4'} \Lambda^{\sigma_4\sigma_3\varsigma'}_{\nu'+\omega,-\omega,l_4,l_3,l_4'l_3'}, \tag{94}$$

$$\left[M^{\vartheta\vartheta'}_{\nu\nu'\omega}\right]^{\sigma_1\sigma_2\sigma_3\sigma_4}_{l_1l_2l_3l_4} = \sum_{\{l'\}} \Lambda^{\sigma_1\sigma_2\vartheta}_{\nu\omega,l_1,l_2,l_1'l_2'} \bar{w}^{\vartheta\vartheta'}_{\omega,l_1'l_2',l_3'l_4'} \Lambda^{*\sigma_3\sigma_4\vartheta'}_{\nu'\omega,l_3,l_4,l_3'l_4'}. \tag{95}$$

The bosonic fluctuation that connects the three-point vertices in Eqs. (94) and (95) is

$$\bar{w}^{rr'}_{\omega,l_1l_2,l_3l_4} = w^{rr'}_{\omega,l_1l_2,l_3l_4} - \bar{u}^r_{l_1l_2l_3l_4}\delta_{rr'}. \tag{96}$$

It corresponds to the renormalized interaction $w^{rr'}$ of the reference system that can be obtained from the corresponding susceptibility as

$$w^{rr'}_{\omega,l_1l_2,l_3l_4} = \tilde{U}^r_{l_1l_2,l_3l_4}\delta_{rr'} + \sum_{\{l'\}} \tilde{U}^r_{l_1l_2,l_1'l_2'} \chi^{rr'}_{\omega,l_1'l_2',l_3'l_4'} \tilde{U}^{r'}_{l_3'l_4',l_3l_4}. \tag{97}$$

As has been discussed in Refs. [93, 94], the choice (10)–(13) for the bare interaction $U^r$ in different $r$ channels provides the best possible partially bosonized approximation for the four-point vertex function (93). However, this choice leads to the double-counting of the bare interaction of the reference problem. This double-counting is removed by the $\bar{u}^r$ term that enters Eq. (96). The expression for this term can be obtained in the same way as in Refs. [93, 94], and for the multi-band case explicitly reads

$$\bar{u}^{\varsigma}_{l_1l_2,l_3l_4} = \frac{1}{2}U^{\varsigma}_{l_1l_2,l_3l_4}, \qquad \bar{u}^{\vartheta}_{l_1l_2,l_3l_4} = U^{\vartheta}_{l_1l_2,l_3l_4}. \tag{98}$$

## C Density and energy from D-TRILEX calculations

The density for each site-orbital index $l$ can be calculated using the regular formula, based on the Green's function

$$\langle n_{l\sigma}\rangle = \frac{1}{2} + \sum_k \mathrm{Re}\,(G_k)^{\sigma\sigma}_{ll} = \frac{1}{2} + \frac{1}{\beta}\sum_\nu \mathrm{Re}\left(G^{\mathrm{loc}}_\nu\right)^{\sigma\sigma}_{ll}, \tag{99}$$

and the total density is simply $\langle n\rangle = \sum_{l\sigma}\langle n_{l\sigma}\rangle$. In our calculations, we employed a fitting of the first few even orders in $1/\nu$ of the tail of the local Green's function $G^{\mathrm{loc}}_\nu$ to get an accurate result.

Another important single-particle observable of the system is the average energy of the system. It can be obtained as the expectation value of the Hamiltonian $H$ corresponding to action (1) of the system, $\langle E\rangle_{\mathrm{tot}} = \langle H\rangle$. The total energy $\langle E\rangle_{\mathrm{tot}} = \langle E\rangle_{\mathrm{kin}} + \langle E\rangle_{\mathrm{pot}}$ can be split into a single-particle kinetic contribution $\langle E\rangle_{\mathrm{kin}}$ and a potential energy $\langle E\rangle_{\mathrm{pot}}$ coming from the interaction. The kinetic part corresponds to

$$\langle E\rangle_{\mathrm{kin}} = \sum_{\substack{k,\{l\},\\\sigma\sigma'}} \left[\varepsilon^{\sigma\sigma'}_{\mathbf{k},ll'} - \mu\delta_{ll'}\delta_{\sigma\sigma'}\right]\langle c^*_{k\sigma l}c_{k\sigma'l'}\rangle = \sum_{\substack{k,\{l\},\\\sigma\sigma'}} \left(\varepsilon^{\sigma\sigma'}_{\mathbf{k},ll'} - \mu\delta_{ll'}\delta_{\sigma\sigma'}\right)G^{\sigma\sigma'}_{k,ll'}. \tag{100}$$

The potential contribution is the expectation value of the interacting terms

$$\langle E\rangle_{\mathrm{pot}} = \frac{1}{2}\sum_{q,\{k\}}\sum_{\{l\},\{\sigma\}} U^{pp}_{l_1l_2l_3l_4}\left\langle c^*_{k\sigma l_1}c^*_{q-k,\sigma'l_2}c_{q-k',\sigma'l_4}c_{k'\sigma l_3}\right\rangle$$
$$+ \frac{1}{2}\sum_{q,\{l\},\varsigma} V^{\varsigma}_{q,l_1l_2l_3l_4}\left\langle\rho^{\varsigma}_{-q,l_1l_2}\rho^{\varsigma}_{q,l_4l_3}\right\rangle, \tag{101}$$

where we neglected particle-particle contributions. After recasting the interaction in the particle-hole representation (79), we can use the relation (68) and the definition of $\rho^{\varsigma}_{q,ll'}$ to replace the expectation values of four operators with the quantities computed in D-TRILEX

$$\langle n^{\varsigma}_{q,l_2 l_1} n^{\varsigma}_{-q,l_3 l_4} \rangle = -X^{\varsigma}_{q,l_1 l_2, l_3 l_4} + \langle n^{\varsigma}_{q,l_2 l_1} \rangle \langle n^{\varsigma}_{-q,l_3 l_4} \rangle. \tag{102}$$

Following these replacements and excluding unphysical term due to Pauli principle, the final expression for the potential energy in the Kanamori approximation for the interaction reads

$$\begin{aligned}
\langle E \rangle_{\text{pot}} = &-\frac{1}{2} \sum_{q,\{l\}} U^{ph}_{l_3 l_4 l_1 l_2} \left[ X^d_q \right]_{l_1 l_2 l_3 l_4} + \frac{1}{2} \sum_{\{l\}} U^{ph}_{l_3 l_4 l_1 l_2} \langle n^d_{l_2 l_1} \rangle \langle n^d_{l_3 l_4} \rangle \\
&+ \frac{1}{4} \sum_{q,l} U^{ph}_{llll} \left[ X^d_q \right]_{llll} + \frac{1}{4} \sum_{q,l} U^{ph}_{llll} \left[ X^z_q \right]_{llll} - \frac{1}{4} \sum_{q,l} U^{ph}_{llll} \langle n^d_{ll} \rangle^2 \\
&- \frac{1}{2} \sum_{l_1,l_2} (1 - \delta_{l_1,l_2}) \left\{ U^{ph}_{l_1 l_2 l_1 l_2} \langle n^d_{l_1 l_1} \rangle - \frac{1}{2} U^{ph}_{l_1 l_2 l_2 l_1} \left( X^{ch}_{l_2 l_1 l_1 l_2} + X^z_{l_2 l_1 l_1 l_2} - \frac{1}{2} \langle n^d_{l_1 l_2} \rangle^2 \right) \right\} \\
&- \frac{1}{2} \sum_{q,\{l\}} \left[ V^{ch}_{\mathbf{q}} \right]_{l_3 l_4 l_1 l_2} \left[ X^{ch}_q \right]_{l_1 l_2 l_3 l_4} - \frac{3}{2} \sum_{q,\{l\}} \left[ V^z_{\mathbf{q}} \right]_{l_3 l_4 l_1 l_2} \left[ X^z_q \right]_{l_1 l_2 l_3 l_4}. \tag{103}
\end{aligned}$$

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
