# Peer review of "Multi-band D-TRILEX approach to materials with strong electronic correlations"

_SciPost Physics, doi:SciPost Phys. 13, 036 (2022)_

## Round 1 · Referee Report · Anonymous (Referee 1) · 2022-5-17

Strengths

1 - presentation of the formalism is thorough
2 - numerical benchmarks are convincing
3 - the method presented truly appears to have good prospects for application

Weaknesses

1 - it is difficult to extract the main points from the paper (introduction and conclusion sections are very short compared to the overall length)
2 - the paper presents very little new physical insight
3 - the paper presents very few new technical and conceptual ideas (the generalization of the D-TRILEX to the multiband case appears rather straightforward)

Report

The authors give a thorough account of the D-TRILEX formalism, developed for a rather general, multi-band lattice model. Details of the implementation are discussed, and the method is tested on several examples, and compared directly to other methods (DMFT, DiagMC@DB). In cases where no benchmark data is available, the physical picture emerging in the tests is analyzed in some detail and is found to be consistent with the findings of previous works.

The subject of the paper is important. The development of methods that capture non-local correlations in 2D lattice models falls under central tasks in condensed matter theory. Added to that, the ability to treat multiband models may prove essential for the description of numerous materials of interest. In my view, the work presented in this paper is very well motivated.

The numerical results appear very convincing: Figs.2-4 demonstrate that D-TRILEX performs consistently better than DMFT and gives results which are comparable to (I imagine) much more expensive DiagMC@DB. The amount of data presented also indicates that the computations are, indeed, feasible and relatively cheap.

However, I find that some important points are not sufficiently emphasized. As far as I understand, one of the big advantages of the D-TRILEX method is that it does not require retarded interactions in the impurity problem (unlike TRILEX). There are no mentions of this in neither the introduction or the abstract. In the conclusions, it is stated that even frequency-dependent interactions in the original model would not change this desirable property of the D-TRILEX method. This should be more elaborated as it may not be immediately obvious to a casual reader.

I also find that the benchmark in the case of the extended Hubbard model (Section 5.2) is out of place. There have been previous implementations of the D-TRILEX method which could have treated this same model (e.g. Ref. 91). I do not understand why the multiband generalization is needed for this benchmark. It would have made more sense to focus on actual multiband models in the main text. At least, the authors should have emphasized that Fig.4 does not actually test any of the new generality of D-TRILEX developed in this work.

I also find a general lack of discussion regarding the systematic errors in the theory. One of the main touted strengths of D-TRILEX is that the expression for the self-energy and polarization in D-TRILEX satisfy symmetries that are not satisfied in standard TRILEX. In standard TRILEX, the expressions Σ=λGWΛ and P=λGGΛ (featuring only one renormalized vertex Λ) allow for a straightforward formulation of systematic corrections to the theory: In the cluster TRILEX, the expressions for Σ and P are of the same form as in the single-site theory, and as Nimp, one recoveres the exact result . It is not immediately clear that the single-site D-TRILEX can be systematically improved towards the exact theory. What is the Nimp limit of D-TRILEX? I think that in the introductions and conclusion, the properties of D-TRILEX should be discussed in terms of control parameters and sources of systematic error. Is there a series of systematic improvements to D-TRILEX which leads to the exact solution of a model? Can the extended Hubbard model (discussed in Section 5.2) be rewritten in a 2-site supercell notation and then solved straightforwardly by D-TRILEX as a multiband model? If possible, such benchmark would certainly be of great interest for the readers, and would greatly strenghthen the paper. In fact, in the introduction it is stated that the authors formulate D-TRILEX "in a multi-orbital and multi-site framework", yet no actual benchmarks with Nimp>1 were given, to my understanding.

In general, I would suggest that the authors expand the introduction and conclusions so as to rely less on the reader's ability to sift through very long formalism. Main points should be accessible in a quick reading of the paper.

Requested changes

1 - in the introduction and conclusions, emphasize more and elaborate on the properties of the impurity problem. If no retarded interactions are needed in the impurity problem, why is that? Under what conditions is this no longer true?

2 - explain better the motivation for the benchmark in the case of the extended Hubbard model. What does this have to do with the multiband generalization of D-TRILEX? What about the multi-site (Nimp>1) solution for the extended Hubbard model?

3 - discuss more the systematic errors, and possibility for systematic improvements. How does the formalism change if we improve the dual boson starting point? Explain what is the Nimp limit of the method.

4 - Discuss more the relationship of D-TRILEX to TRILEX and other methods. Citation of Phys. Rev. B 96, 104504 (2017) in the context of TRILEX and GW+DMFT is due, as it showcases the versatility of those methods. Can D-TRILEX be formulated in Nambu space to treat superconductivity (it is, in a way, a 2-orbital calculation)?

5 - Discuss more the stability of the Dyson equation in the second paragraph of page 11. Have you encountered problems with diverging bosons? Is there a general problem in reaching low temperatures? Can one suppress ordering instabilities? Can one apply D-TRILEX in ordered (say, AFM) phases?

5 - below Eq.2 define γk,ll for the sake of completeness.

6 - last paragraph before Section 6, fix "the complex physics in the considered this model"

---

## Round 1 · Referee Report · Anonymous (Referee 2) · 2022-5-22

Report

The authors present an extension of the D-TRILEX numerical method for many body systems introduced in Ref. [91] for a single-band model to the more generic case of multi-orbital systems. After introducing the formalism in Sections 2 and 3 (with the support of the Appendices) and discussing the details of the numerical implementation in Sec. 4, the authors showed applications of the method to several cases, benchmarking it when possible to exact solutions or less approximated methods.
Although there are no conceptual advances with respect to the single orbital case, the article reports technical advances due to numerical difficulties for the theoretical treatment of more complex multi-orbital systems.

The benchmark with the Kanamori dimer is particularly interesting. In fact, given the extreme simplicity of it, the dimer has an exact solution that can be used to validate the method. In section 5.1., it is shown how DMFT fails to predict the correct value of the imaginary part of the local self-energy at low frequency while the D-TRILEX gives the correct result.
However, the benchmark with the exact solution is not complete. In fact, the authors should show how the non-local components of the self-energy, that are missing in DMFT, compare with the exact solution. At the same time, it would be important to check that also the susceptibilities calculated with D-TRILEX compare well with the exact solution.
Furthermore, since the method can treat non-local interactions, it would be instructive to show a comparison with the dimer in presence of non-local interactions as well.

In section 5.2 the authors apply the D-TRILEX to the extended Hubbard model on a square lattice and compare the results to the ones obtained using DIAGMC@DB in Ref.[82]. While for weak-to-intermediate values of the onsite repulsion U the agreement between the two methods is good, their predictions start to deviate for larger values of the Hubbard U. While the authors report this problem they do not really discuss in too much detail why it is happening and how this could be overcome.
In these works Phys. Rev. B 99, 235106 (2019) and Phys. Rev. B 104, 235128 (2021) efficient ways for calculating vertex functions in the strong coupling limit have been put forward reproducing the asymptotic power law behavior of the Neel Temperature in three dimensions for large values of U. The authors should cite these papers and comment if it is possible to use approximate forms for the vertices to obtain more accurate results at stronger couplings.

As a suggestion to improve the readability of the paper, I recommend the authors to add a figure where they show schematically the workflow of the D-TRILEX discussed in section 4.

I find the manuscript interesting enough and I recommend its publication after my comments have been properly addressed.

---

## Round 2 · Referee Report · Anonymous · 2022-7-2

Report
The authors have addressed the criticisms raised by the referees substantively.
The revised version of the paper includes additional benchmark data which strengthen confidence in the applicability of the proposed D-TRILEX method.
There are now additional explanations that make the paper more readable, and the main points more easily accessible. The additional comments are written concerning
- the meaning of $N_\mathrm{imp}$ and the distinction between cluster impurity models and lattices with inequivalent sites
- the possibility and the role of including bosonic hybridization functions
- systematic improvements towards exact theory (in terms of better reference problems and more complete diagrammatic extensions)
- prospects for further generalizations (ordered phases, superconductivity)
The authors also further elaborate on the divergence of bosonic propagators and the strategy to stabilize the calculation. The issue is now presented fairly and in a way that is accessible to a non-specialist.
The added Fig. 2 nicely summarizes the workflow of the method, which greatly improves the readability of the paper.
Concerning my request from the previous review:
> 5 - below Eq.2 define →γk,ll′ for the sake of completeness.
> <<A: This quantity is already defined few lines after Eq.(2).>>
Here I meant that one should write the Rashba form of SOC with an equation, rather than to refer to literature.
Overall, the revised version of the paper is excellent, and I recommend publication in SciPost Physics.

---

## Round 2 · Referee Report · Anonymous · 2022-7-4

Report
The authors added important information that was missing in the previous version showing further limitations of the method, in particular when a non-local interaction is added.
However, I am not satisfied with the benchmark of the self-energy results in the case of the dimer.
The authors have chosen a set of parameters where the non-local component of the self energy is way smaller than the local one. For this set of parameters, DMFT is already a good approximation for the dimer and the benchmark is useless in this case. The authors should use the same parameters that yields a substantial deviation between ED and DMFT results as shown in Fig.3 when U = t.
Also, I am not particularly impressed by the explanation of why this method should work at strong coupling.
First, numerical results point to the opposite direction.
Second, even if this is an expansion around the strong coupling limit,
it really does not imply that the method would yield accurate results at the two-particle level at strong coupling. In fact, the papers I have suggested to be referenced address this topic thoroughly: DIFFERENT strong coupling expansions can yield DIFFERENT values of the Neel critical temperature in 3D and sometimes completely destroy antiferromagnetism (AFM). The right approximation schemes are the one that stabilize AFM starting from the atomic limit (where AFM is absent) and yield the right critical temperature.
A natural question that arises at this point, would D-Trilex yield the correct value of T-Neel at strong coupling in 3D? Is out there a reference showing this?
Also, even if D-Trilex does not use the four-point vertex function, DB technique does, therefore having reliable approximate schemes to calculate the four-point vertex function would reduce the computational complexity for numerical calculation of strongly correlated systems in regime where D-Trilex does not work.
I think these are subtle and important aspects that the authors missed and could leave for future work, but I truly believe that mentioning them in the discussion section and adding the suggested references would enrich their work.

---

## Round 2 · Author Response

Reply to Anonymous Report 1
Report
The authors give a thorough account of the D-TRILEX formalism, developed for a rather general, multi-band lattice model. Details of the implementation are discussed, and the method is tested on several examples, and compared directly to other methods (DMFT, DiagMC@DB). In cases where no benchmark data is available, the physical picture emerging in the tests is analyzed in some detail and is found to be consistent with the findings of previous works.
The subject of the paper is important. The development of methods that capture non-local correlations in 2D lattice models falls under central tasks in condensed matter theory. Added to that, the ability to treat multiband models may prove essential for the description of numerous materials of interest. In my view, the work presented in this paper is very well motivated.
The numerical results appear very convincing: Figs.2-4 demonstrate that D-TRILEX performs consistently better than DMFT and gives results which are comparable to (I imagine) much more expensive DiagMC@DB. The amount of data presented also indicates that the computations are, indeed, feasible and relatively cheap.
<<A: We thank the Referee for the in-depth analysis of our work, as well as for the useful suggestions and comments.>>
However, I find that some important points are not sufficiently emphasized. As far as I understand, one of the big advantages of the D-TRILEX method is that it does not require retarded interactions in the impurity problem (unlike TRILEX). There are no mentions of this in neither the introduction or the abstract. In the conclusions, it is stated that even frequency-dependent interactions in the original model would not change this desirable property of the D-TRILEX method. This should be more elaborated as it may not be immediately obvious to a casual reader.
<<A: We think that a more correct statement would be that one of the advantages of the D-TRILEX method is that it can be formulated on the basis of an arbitrary interacting reference problem. The reason is that in our approach the splitting between the impurity part and the rest of the lattice action can be introduced arbitrarily by adding and subtracting the fermionic and bosonic hybridization functions, as highlighted in the derivation of the partially bosonized theory (Appendix A.1). For this reason, even if the original problem contains the non-local interaction, the reference problem does not necessarily have to (but, in principle, can) contain the corresponding frequency-dependent bosonic hybridization. To our knowledge, our ideology if different from the one of GW+EDMFT approach, where the frequency-dependent bosonic hybridization function is always considered when taking into account the non-local interaction. The TRILEX approach is derived is a similar way as the GW+EDMFT method. For instance, in Ref.[PRB 97, 155145 (2018)] the authors say ``This simplified TRILEX version can be seen as a GW+EDMFT-like scheme which, however, can treat simultaneously both charge and spin fluctuations''. Nevertheless, we don't know whether taking into account the bosonic hybridization is necessary in TRILEX when the non-local interaction is considered. This question should be addressed to the authors of the TRILEX method. We comment on the use of the bosonic hybridization function in the paragraph after Eq. (4) when discussing different reference systems.>>
I also find that the benchmark in the case of the extended Hubbard model (Section 5.2) is out of place. There have been previous implementations of the D-TRILEX method which could have treated this same model (e.g. Ref. 91). I do not understand why the multiband generalization is needed for this benchmark. It would have made more sense to focus on actual multiband models in the main text. At least, the authors should have emphasized that Fig.4 does not actually test any of the new generality of D-TRILEX developed in this work.
<<A: Previous works on D-TRILEX [PRB 100, 205115 (2019), PRB 103, 245123 (2021), npj Comput. Mater. 8, 118 (2022)] indeed suggest that the method is able to account for the effect of the non-local interactions. However, no thorough benchmarking of the results for the extended Hubbard model has been performed so far. For this reason, in this work we decided also to investigate the performance of D-TRILEX in the case of a single-orbital extended Hubbard model on a square lattice with the local U and the nearest-neighbor V interactions between electronic densities. The aim of the current work is to provide a comprehensive discussion of the D-TRILEX method and show how the method performs against exact methods. For this reason, we believe that this Section fits well within the scope of the manuscript, especially since the numerical calculations in the entire work including the ones for the single-band extended Hubbard model have been performed using the code implementation discussed in this manuscript. We have clarified this point at the beginning of the Section 5.2 of the revised manuscript.>>
I also find a general lack of discussion regarding the systematic errors in the theory. One of the main touted strengths of D-TRILEX is that the expression for the self-energy and polarization in D-TRILEX satisfy symmetries that are not satisfied in standard TRILEX. In standard TRILEX, the expressions Σ=λGWΛ and P=λGGΛ (featuring only one renormalized vertex Λ) allow for a straightforward formulation of systematic corrections to the theory: In the cluster TRILEX, the expressions for Σ and P are of the same form as in the single-site theory, and as Nimp→∞, one recoveres the exact result . It is not immediately clear that the single-site D-TRILEX can be systematically improved towards the exact theory. What is the Nimp→∞ limit of D-TRILEX? I think that in the introductions and conclusion, the properties of D-TRILEX should be discussed in terms of control parameters and sources of systematic error. Is there a series of systematic improvements to D-TRILEX which leads to the exact solution of a model?
<<A: We acknowledge that this is an important point to address. First of all, as we write at the beginning of Section 3, ``the introduced effective fermion-boson action (5) allows for the calculation of observable quantities by using diagrammatic techniques. This goal can be achieved either by performing approximated diagrammatic expansions or by applying exact numerical methods such as the diagrammatic Monte Carlo (DiagMC) scheme. In this section we discuss the simplest diagrammatic approximation, namely the D-TRILEX method, that represents a feasible approach for actual calculations in the multi-band case.'' For this reason, the developed approach allows one to systematically improve the theory by including more diagrammatic contributions. For instance, one can consider much more complex diagrammatic structures that account for vertical (transverse) insertions of momentum- and frequency-dependent bosonic fluctuations, which are present in the DiagMC@DB approach but are not considered in ladder-like dual approximations including the D-TRILEX approach (this sentence is added to the revised main text). In this case, the polarization operator of D-TRILEX will also take the form P=λGGΛ, where Λ will be a vertex renormalized by the transverse fluctuations. However, it should be pointed out that these diagrammatic contributions can be considered only by sacrificing the simplicity of the computational scheme proposed here. In fact, the D-TRILEX approach has one of the most complex diagrammatic structures that still allow for feasible numerical calculations in the multi-orbital case. Another systematic improvement can be achieved by ``tuning the parameters of the reference system, especially in situations where the DMFT impurity problem does not provide a good starting point for the D-TRILEX diagrammatic expansion. For instance, the reference system can be improved by considering a cluster of suitable size, or by introducing more appropriate fermionic and bosonic hybridization functions.'' This discussion has been added to Conclusions. Regarding the N→∞ limit, if we consider a cluster of size Nl (with some requirements on symmetry) as the reference system, taking Nl →∞ also leads to the exact solution in our method. The reason behind this statement is that D-TRILEX is derived as a diagrammatic expansion in terms of dual Green's functions Gd=G-g on top of an exactly solved reference system. Here, G is a cluster DMFT Green's function and g is the Green's function of the reference system. In the limit of Nl →∞, the reference system coincides with the lattice problem. This implies that G=g and also means that in the considered limit all diagrammatic contributions in D-TRILEX vanish.>>
Can the extended Hubbard model (discussed in Section 5.2) be rewritten in a 2-site supercell notation and then solved straightforwardly by D-TRILEX as a multiband model? If possible, such benchmark would certainly be of great interest for the readers, and would greatly strenghthen the paper. In fact, in the introduction it is stated that the authors formulate D-TRILEX "in a multi-orbital and multi-site framework", yet no actual benchmarks with Nimp>1 were given, to my understanding.
<<A: Yes, the extended Hubbard model on a square lattice can be rewritten in a supercell notation and solved in the same way as a multi-band system. However, we think that these calculations would not add any new results. In fact, a similar calculation has already been performed for a bilayer square lattice (Section 5.4). In that case we considered two equivalent representations for the model, namely with either one (Nimp = 1) or two (Nimp = 2) lattice sites in the unit cell, and showed that the results for these two identically coincide. As to the second part of the question, we realized that we did not clearly state what we mean by the expression ``multi-site'', so we decided to add the following clarification after Eq.(28): ``In this context, Nimp is the number of independent impurities in the unit cell of the reference problem. Note that the case of Nimp > 1 corresponds to a collection of impurities, as explained in Ref.[PhysRevB.97.115150], and not to a cluster of Nimp sites. If the impurities are all identical, then the reference system reduces to a single site impurity problem. If some of them are different, it is sufficient to solve an impurity problem only for the non-equivalent ones. In the multi-impurity case, fluctuations between the impurities are taken into account diagrammatically in the framework of D-TRILEX approach. On the other hand, a cluster reference system corresponds to a multi-orbital problem with Nimp =1. In this case, Nl is the total number of orbitals and sites of the considered cluster. The separation between orbitals and sites that we introduce is useful to reduce the computational complexity when addressing problems with several atoms in the unit cells.''>>
In general, I would suggest that the authors expand the introduction and conclusions so as to rely less on the reader's ability to sift through very long formalism. Main points should be accessible in a quick reading of the paper.
Requested changes
1 - in the introduction and conclusions, emphasize more and elaborate on the properties of the impurity problem. If no retarded interactions are needed in the impurity problem, why is that? Under what conditions is this no longer true?
<<A: As we explain above, our method relies on the arbitrary splitting between the reference problem and the rest of the action. In this case, considering the frequency dependent hybridization function is not mandatory, but still can be done if needed for a particular calculation. As we demonstrate in Section 5.2, ``the D-TRILEX approach can accurately treat the non-local interactions on the basis of the DMFT impurity problem. The latter does not contain the bosonic hybridization function and thus is easier to solve numerically.'' At the same time, performing additional calculations for the susceptibility for the dimer problem presented in Section 5.1 we find that the largest mismatch between the D-TRILEX and the exact results appears only for a very strong non-local interaction V>0.3U. We argue there that ``the inclusion of the bosonic hybridization function in the spirit of EDMFT could improve the result, because in this case some contributions of the non-local interaction would be taken into account in the impurity problem via the bosonic hybridization.''>>
2 - explain better the motivation for the benchmark in the case of the extended Hubbard model. What does this have to do with the multiband generalization of D-TRILEX? What about the multi-site (Nimp > 1)solution for the extended Hubbard model?
<<A: This question has been addressed above and the corresponding changes to the text have been added.>>
3 - discuss more the systematic errors, and possibility for systematic improvements. How does the formalism change if we improve the dual boson starting point? Explain what is the Nimp→∞ limit of the method.
<<A: This question is also addressed above.>>
4 - Discuss more the relationship of D-TRILEX to TRILEX and other methods. Citation of Phys. Rev. B 96, 104504 (2017) in the context of TRILEX and GW+DMFT is due, as it showcases the versatility of those methods. Can D-TRILEX be formulated in Nambu space to treat superconductivity (it is, in a way, a 2-orbital calculation)?
<<A: We thank the Referee for pointing to the missing citation. The corresponding discussion has been added to Conclusions and reads: ``Even though not included in our current computational scheme, one can also address the problem of superconductivity in the framework of the developed approach. In order to find the transition temperature to a superconducting state one can look at the divergence of the lattice susceptibility (72) in the corresponding particle-particle channel. To this aim one can either consider a suitable reference problem related to the desired superconducting order parameter (see, e.g., Refs.PRB 94, 125133 (2016), PRB 100, 024510 (2019), PRB 101, 045119 (2020), npj Quantum Mater. 7, 50 (2022)]) and use the D-TRILEX form for the polarization operator in the particle-particle channel, or to additionally account for the scattering on the transverse momentum- and frequency-dependent bosonic fluctuations in the polarization operator in the particle-particle channel in the case of a single-site reference system [PRB 90, 235132 (2014), PRB 96, 104504 (2017), PRB 99, 041115 (2019)]. Going inside the superconducting phase would require to introduce an anomalous component of the Green's function working in the Nambu space formalism similarly to what has been proposed in the framework of the TRILEX approach [PRB 96, 104504 (2017)].''>>
5 - Discuss more the stability of the Dyson equation in the second paragraph of page 11. Have you encountered problems with diverging bosons? Is there a general problem in reaching low temperatures? Can one suppress ordering instabilities? Can one apply D-TRILEX in ordered (say, AFM) phases?
<<A: Yes, we have issues with the convergence of this equation when we are close to a phase boundary. In the current implementation of the method we cannot go to dynamically broken symmetry phases, so we experience a divergence at the transition point. We added a more detailed discussion of strategies to avoid the divergence in low-dimensional systems, where Mermin-Wagner theorem forbids certain forms of symmetry breaking. We expect to being able to partially cure this issue by considering cluster or a multi-impurity reference problems. We would like to point out that this issue is not restricted to D-TRILEX. Most of the diagrammatic methods exhibit this issue in one way or another.>>
5 - below Eq.2 define →γk,ll′ for the sake of completeness.
<<A: This quantity is already defined few lines after Eq.(2).>>
6 - last paragraph before Section 6, fix ``the complex physics in the considered this model''.
<<A: The misprint has been corrected.>>
Reply to Anonymous Report 2
The authors present an extension of the D-TRILEX numerical method for many body systems introduced in Ref. [91] for a single-band model to the more generic case of multi-orbital systems. After introducing the formalism in Sections 2 and 3 (with the support of the Appendices) and discussing the details of the numerical implementation in Sec. 4, the authors showed applications of the method to several cases, benchmarking it when possible to exact solutions or less approximated methods. Although there are no conceptual advances with respect to the single orbital case, the article reports technical advances due to numerical difficulties for the theoretical treatment of more complex multi-orbital systems.
<<A: We thank the Referee for reviewing our manuscript as well as for the useful comments and suggestions.>>
The benchmark with the Kanamori dimer is particularly interesting. In fact, given the extreme simplicity of it, the dimer has an exact solution that can be used to validate the method. In section 5.1., it is shown how DMFT fails to predict the correct value of the imaginary part of the local self-energy at low frequency while the D-TRILEX gives the correct result. However, the benchmark with the exact solution is not complete. In fact, the authors should show how the non-local components of the self-energy, that are missing in DMFT, compare with the exact solution.
<<A: We thank the Referee for suggesting this addition to our manuscript. Indeed, the self-energy directly carries the information about the local part included in DMFT and the neglected non-local part. We added two panels in Figure 3 (old Figure 2) showing the corresponding results for the self-energy.>>
At the same time, it would be important to check that also the susceptibilities calculated with D-TRILEX compare well with the exact solution. Furthermore, since the method can treat non-local interactions, it would be instructive to show a comparison with the dimer in presence of non-local interactions as well.
<<A: Based on this suggestion, we added the results for the static charge and spin susceptibilities obtained for different values of the local U and the non-local V interactions to the revised manuscript. They are presented in Figure 5. We find that the D-TRILEX susceptibility matches very well with the exact result for small-to-moderate non-local interaction strength and breaks down at large non-local interactions.>>
In section 5.2 the authors apply the D-TRILEX to the extended Hubbard model on a square lattice and compare the results to the ones obtained using DIAGMC@DB in Ref.[82]. While for weak-to-intermediate values of the onsite repulsion U the agreement between the two methods is good, their predictions start to deviate for larger values of the Hubbard U. While the authors report this problem they do not really discuss in too much detail why it is happening and how this could be overcome.
<<A: The reason for the deviation from the exact result at larger values of U is that in this regime the system lies close to a magnetic instability, where magnetic fluctuations become strongly non-linear (see, e.g., Ref. [PRB 102, 224423 (2020)]). This non-linear behavior originates from the mutual interplay between different bosonic modes as well as from an anharmonic fluctuation of the single mode itself. The description of these effects requires to consider much more complex diagrammatic structures that account for vertical (transverse) insertions of momentum- and frequency-dependent bosonic fluctuations, which are present in the DiagMC@DB approach but are not considered in ladder-like dual approximations including the D-TRILEX approach. This point is clarified in the revised manuscript. We want to stress, however, that the method is exact in the large-U regime, as it is an expansion around an impurity that represents the exact solution in that limit. Indeed, in work mentioned by the Referee, it is shown that the method performs better when the interaction strength is larger than the bandwidth (U=10-12).>>
In these works Phys. Rev. B 99, 235106 (2019) and Phys. Rev. B 104, 235128 (2021) efficient ways for calculating vertex functions in the strong coupling limit have been put forward reproducing the asymptotic power law behavior of the Neel Temperature in three dimensions for large values of U. The authors should cite these papers and comment if it is possible to use approximate forms for the vertices to obtain more accurate results at stronger couplings.
<<A: The weaknesses of the D-TRILEX method and the mismatches with the exact results are not related to the calculation of impurity vertices. These vertices are computed exactly using continuous-time quantum Monte Carlo method within the w2dynamics software package. Furthermore, these papers seem not related in any way to our work as they rely on the calculation of the four-point vertex function, which is completely absent in our approach. In D-TRILEX, the four-point vertex functions are eliminated from the theory by using a special path-integral (Hubbard-Stratonovich) transformation that generates a partially bosonized approximation for the four-point vertex function. This approximation was first introduced by some of us in a set of works in a single-orbital case [PRL 121, 037204 (2018), PRB 99, 115124 (2019), PRB 100, 205115 (2019), PRB 103, 245123 (2021)], and in the current work was extended to a multi-orbital case.>>
As a suggestion to improve the readability of the paper, I recommend the authors to add a figure where they show schematically the workflow of the D-TRILEX discussed in section 4.
<<A: The corresponding figure has been added to the text as a new Figure 2.>>
I find the manuscript interesting enough and I recommend its publication after my comments have been properly addressed.
<<A: We thank again the Referee for the useful suggestions. >>

---

## Round 2 · List of Changes

1. The following text has been added to introduction:
``The introduced approach provides a consistent formulation of a diagrammatic expansion on the basis of an arbitrary interacting reference problem.
In particular, considering a finite cluster as the reference problem allows one to combine the diagrammatic and cluster ways of taking into account the non-local correlation effects within the multi-band D-TRILEX computational scheme.''
2. In Section 2 we added a comment on the bosonic hybridization function:
``It is also possible to build the D-TRILEX diagrammatic expansion on the basis of the impurity problem of the extended dynamical mean field theory (EDMFT)[126–130] by introducing a bosonic hybridization function (see Appendix A.1).
The latter accounts for the effect of the non-local interaction on the local electronic correlations and could play an important role when the non-local interactions are strong.''
3. In Section 2 we also added a comment on the use of clusters, clarifying the limit N →∞:
``The limit of an infinite plaquette as a reference system corresponds to the exact solution of the problem.
For this reason, we expect the accuracy of the D-TRILEX method to improve with enlarging the cluster similarly to what has been shown for the TRILEX approach [92].
Indeed, as the spatial size of the reference problem is increased, the range of electronic correlations that are treated within the exactly-solved cluster reference problem is also increased.
Additionally, using a cluster reference system allows for the study of broken symmetry phases.
In this regard, instead of viewing the cluster methods and the multi-band D-TRILEX theory as competing approaches, one could consider D-TRILEX as a method to improve the cluster solution of the problem by diagrammatically adding long-range correlations that are not captured by a finite cluster when the computational costs prevent a further increase of the cluster's size.''
4. We added Figure 2 with the computational workflow in Section 4.
5. In Section 4.2, we added a clarification on the meaning of Nimp and the distinction with a cluster calculation:
``In this context, Nimp is the number of independent impurities in the unit cell of the reference problem.
Note that the case of Nimp > 1 corresponds to a collection of impurities, as explained in Ref.[122], and not to a cluster of Nimp sites.
If the impurities are all identical, then the reference system reduces to a single site impurity problem.
If some of them are different, it is sufficient to solve an impurity problem only for the non-equivalent ones.
In the multi-impurity case, fluctuations between the impurities are taken into account diagrammatically in the framework of D-TRILEX approach.
On the other hand, a cluster reference system corresponds to a multi-orbital problem with Nimp = 1.
In this case, Nl is the total number of orbitals and sites of the considered cluster.
The separation between orbitals and sites that we introduce is useful to reduce the computational complexity when addressing problems with several atoms in the unit cells.''
6. In Section 4.2, we also added a more detailed explanation of the divergence of the bosonic Dyson's equation:
``The stability of the bosonic Dyson equation (16) can be problematic in regimes of parameters, where one or more of the eigenvalues of the quantity Π·W become equal or larger than 1.
In particular, this happens when the system is close to a phase transition
or if the correlation length in some channel of instability exceeds a critical value.
This issue appears in similar forms in other diagrammatic extensions of DMFT (see, e.g., Ref. [138]). … ''
``Of course, no rescaling is expected to work in the presence of the symmetry breaking due to a true phase transition.
The latter case should be addressed using a suitable cluster or multi-impurity reference problem.''
7. In Figure 3 (old Figure 2), we added panels (d) and (e) showing the results for the real and imaginary parts of the local/non-local self-energy.
8. We added a new Figure 4 that shows the comparison between the D-TRILEX and ED susceptibilties as a function of the local interaction (left panel) and the non-local interaction (right panel). The corresponding discussin of the figure is also added in the text.
\item We clarify our motivation to include the discussion of the single-band extended Hubbard model at the beginning of Section 5.2.
9. In Section 5.2, we also discuss the role of more complex diagrammatic contributions:
``The reason is that magnetic fluctuations become strongly non-linear close to a magnetic instability (see, e.g., Ref. [152]).
This non-linear behavior originates from the mutual interplay between different bosonic modes as well as from an anharmonic fluctuation of the single mode itself.
The description of these effects requires to consider much more complex diagrammatic structures that account for vertical (transverse) insertions of momentum- and frequency-dependent bosonic fluctuations, which are present in the DiagMC@DB approach but are not considered in ladder-like dual approximations including the D-TRILEX approach.''
10. In Conclusion, we added an outlook on the study of symmetry broken phases, as for instance superconductivity:
``Since the method can be formulated on the basis of an arbitrary reference system, it allows for several possibilities for further improvements of the results by tuning the parameters of the reference system, especially in situations where the DMFT impurity problem does not provide a good starting point for the D-TRILEX diagrammatic expansion.
For instance,
the reference system can be improved by considering a cluster of suitable size, or by introducing more appropriate fermionic and bosonic hybridization functions.
In addition to that, the solution of the derived partially bosonized dual action (5) can be systematically improved by considering more elaborate diagrammatic contribution that are not taken into account in D-TRILEX [94].
Even though not included in our current computational scheme, one can also address the problem of superconductivity in the framework of the developed approach.
In order to find the transition temperature to a superconducting state one can look at the divergence of the lattice susceptibility (72) in the corresponding particle-particle channel.
To this aim one can either consider a suitable reference problem related to the desired superconducting order parameter (see, e.g., Refs.[56, 131, 158, 159]) and use the D-TRILEX form for the polarization operator in the particle-particle channel, or to additionally account for the scattering on the transverse momentum- and frequency-dependent bosonic fluctuations in the polarization operator in the particle-particle channel in the case of a single-site reference system[90, 160, 161].
Going inside the superconducting phase would require to introduce an anomalous component of the Green's function working in the Nambu space formalism similarly to what has been proposed in the framework of the TRILEX approach [91].''

---

## Round 3 · Referee Report · Anonymous (Referee 2) · 2022-7-21

Report

The authors have improved the paper by properly answering my points and adding essential information to their manuscript.

However, they did not understand and did not address my criticism about the validity of D-Trilex at strong coupling at the two-particle level.

I think that my criticism could be reformulated in a single simpler question that is:

Is D-Trilex exact in the limit of infinite dimensions at the two-particle level?

If this is the case, D-Trilex would most likely yield good results in 3D when compared to other methods, at least qualitatively.
In fact, DMFT already guarantees the correct power law behaviour of the Neel Temperature at strong coupling in infinite dimensions. In three dimensions, the self-consistency of the D-Trilex method will most certainly yield a reduction of the critical temperature value from the one calculated in the limit of infinite dimensions.

If this is not the case, it is not obvious to me wether D-Trilex would even yield sound results from a qualitative point of view, e.g. power law behaviour of Neel
temperature at strong coupling. In fact, I brought to the authors' attention approximation schemes based on a strong coupling expansion that completely suppress Neel temperature even in the limit of infinite dimensions.
However, D-Trilex could still yield the correct power law behaviour even if it is not exact in infinite dimensions, but it is not obvious and it should be explicitly shown (in future work) as it has been done in PRB 99, 235106 (2019) for an approximation to the vertex function similar to the one the authors used that have been introduced in ref. PRB 100, 155149 (2019).
Having a single point in the phase diagram compared with another method is encouraging but not enough to answer this question.

With this said, and as I already stated in my previous report, I do not expect and I do not ask the authors to do further calculations because such a numerical study could be address in future work.

Anyhow the authors should answer my point and discuss it in their manuscript.
  • validity: -
  • significance: -
  • originality: -
  • clarity: -
  • formatting: -
  • grammar: -

Author:  Matteo Vandelli  on 2022-08-23  [id 2745]

(in reply to Report 1 on 2022-07-21)

We would like to reply to the Referee in order not to leave the questions unanswered.

First of all, we would like to emphasize that the strong coupling limit and the limit of infinite dimensions are two different limits. For this reason, the question about the «validity of D-TRILEX at strong coupling at the two-particle level» cannot be reformulated as «Is D-TRILEX exact in the limit of infinite dimensions at the two-particle level?»

In our replies to similar questions raised by the Referee in the two previous review rounds, we have already explained why the D-TRILEX method is exact in the weak and strong coupling limits (independently on dimension). To complement our previous arguments, we would like to mention that the diagrammatic expansion in the D-TRILEX theory is performed in terms of the three-point vertex functions that are connected by the interaction line W, and the bare dual Green’s functions. In the weak- and strong coupling limits this diagrammatic expansion is a perturbative expansion. Indeed, at weak coupling the small parameter in the diagrammatic expansion is the interaction. In the strong coupling limit the bare dual Greens function, which is purely non-local, is small, because electrons in the system are strongly localized.

As we said previously, we agree with the Referee that different strong coupling expansions can yield different results for the two-particle quantities. However, we believe that the diagrammatic expansion performed in D-TRILEX is consistent, because it is formulated on the basis of the partially bosonized dual action that is derived by means of the exact transformations or controlled approximations. In fact, there are only two approximations that are used when deriving the action for the D-TRILEX theory, namely we neglect the local vertex functions that are higher order than the two-particle (three-point and four-point) ones and we replace the four-point vertex function by its partially-bosoniszed approximation. The first approximation is a standard approximation that is widely used in many DMFT-based diagrammatic expansions including dual fermion/boson theories [Phys. Rev. B 77, 033101 (2008); Ann. Phys. 327(5), 1320 (2012)], DГA method [Phys. Rev. B 75, 045118 (2007)], and TRILEX approach [Phys. Rev. B 100, 205115 (2019); PRB 103, 245123 (2021)]. The second approximation has been tested in detail in a single-orbital case, which showed its validity in a wide range of interaction strengths from weak to strong coupling regimes [PRB 103, 245123 (2021)]. Regarding the validity of the D-TRILEX method at strong coupling at the two particle level, it is worth noting that the leading term (the second-order in the dual Green’s function) that contributes to the susceptibility in the strong coupling limit is exactly the polarization operator of D-TRILEX. More elaborate diagrammatic contributions already have four dual Green’s function in their stricture. In addition, as we point out in the current work, the D-TRILEX polarization operator has the same structure as the expression for the exchange interaction between magnetic densities, and this expression gives the correct result for the exchange interaction ~t2/U in the atomic limit [PRL 121, 037204 (2018)]. This observation also shows that the D-TRILEX diagrammatic expansion is correct at strong coupling at the two-particle level.

Now, let us comment on the d-infinity limit. In this limit, the D-TRILEX method is exact at the single-particle level if the DMFT reference problem is considered. However, D-TRILEX is not exact at the two particle level, because it uses a partially bosonized approximation instead of the exact four-point vertex function. However, we would like to point out that there is no correlation between the exactness of the theory in the limit of infinite dimensions and the accuracy of the theory in finite dimensions. Let us consider DMFT as a particular example proposed by the Referee. DMFT is exact in d-infinity limit at both single- and two-particle levels. However, it does not predict the AFM phase boundary accurately enough for the case of a simple cubic lattice. For instance, at t=1, U=8, where D-TRILEX and the exact Determinant Diagrammatic Monte Carlo give the TNéel = 0.33 value for the Neel temperature (see discussion in the previous review round), DMFT predicts a much higher value TNéel = 0.45 for the AFM phase boundary (see, e.g., Refs. [PRB 92, 144409 (2015); PRB 94, 115117 (2016)]). The t=1, U=8 point is very close to the top of the AFM dome, which means that D-TRILEX accurately predicts the Neel temperature in the most challenging regime of very strong magnetic fluctuations in addition to the exact results that the theory provides in the weak and strong-coupling limits as argued above.

This discussion about the different limits of the theory has been added to main text after Eq. (19) and at the end of Section. 3.2.

---

## Round 3 · Author Response

Reply to the second review of the Referee 1.

The authors have addressed the criticisms raised by the referees substantively. The revised version of the paper includes additional benchmark data which strengthen confidence in the applicability of the proposed D-TRILEX method. There are now additional explanations that make the paper more readable, and the main points more easily accessible. The additional comments are written concerning

  • the meaning of Nimp and the distinction between cluster impurity models and lattices with inequivalent sites

  • the possibility and the role of including bosonic hybridization functions

  • systematic improvements towards exact theory (in terms of better reference problems and more complete diagrammatic extensions)

  • prospects for further generalizations (ordered phases, superconductivity)

The authors also further elaborate on the divergence of bosonic propagators and the strategy to stabilize the calculation. The issue is now presented fairly and in a way that is accessible to a non-specialist. The added Fig. 2 nicely summarizes the workflow of the method, which greatly improves the readability of the paper. Concerning my request from the previous review:

5 - below Eq.2 define →γk,ll′ for the sake of completeness. <<A: This quantity is already defined few lines after Eq.(2).>> Here I meant that one should write the Rashba form of SOC with an equation, rather than to refer to literature. Overall, the revised version of the paper is excellent, and I recommend publication in SciPost Physics.

<<A: We thank the Referee for this nice words and recommendation to accept our work for publication. Concerning the definition of the spin-orbit coupling (SOC): In our work the SOC is described by a momentum-dependent quantity γk,ll′, which is a Fourier transform of effective spin-dependent hopping amplitudes between lattice sites. These effective spin-dependent hoppings can be obtained by diagonalizing a single-particle part of the Hamiltonian that contains the local term ~L·S, which describes the coupling of the orbital angular momentum L to the spin polarization of electrons S=c†σc (for details see, e.g., Ref. [120] – PRB 52, 10239 (1995)). Since the resulting term γk,ll′ is non-local, this form of the SOC is usually referred to as the Rashba form. The precise expression for γk,ll′ depends on the particular system of interest, and for this reason we cannot write down an explicit formula which is valid in general for all the systems that can be studied using our method. To clarify this point, we modified the corresponding sentence in the text of the manuscript. Now it reads: “The non-diagonal contribution in spin space γk,ll′ describes the effect of the external magnetic field and of the spin-orbit coupling (SOC), that is usually expressed in the Rashba form [119]. The latter corresponds to a Fourier transform of effective spin-dependent hopping amplitudes [120].” >>

Reply to the second review of the Referee 2.

The authors added important information that was missing in the previous version showing further limitations of the method, in particular when a non-local interaction is added. However, I am not satisfied with the benchmark of the self-energy results in the case of the dimer. The authors have chosen a set of parameters where the non-local component of the self energy is way smaller than the local one. For this set of parameters, DMFT is already a good approximation for the dimer and the benchmark is useless in this case. The authors should use the same parameters that yields a substantial deviation between ED and DMFT results as shown in Fig.3 when U = t.

<<A: The corresponding results for the non-local self-energy have been added to Fig. 4. These new results show that the non-local part of the D-TRILEX self-energy also follows the trend of the exact result and it is by far the largest contribution to the self-energy for this kind of systems. >>

Also, I am not particularly impressed by the explanation of why this method should work at strong coupling. First, numerical results point to the opposite direction.

<<A: We do not understand what numerical results the Referee is referring to. As we wrote in our reply to a similar question of the Referee in the previous review round: “We want to stress, however, that the [D-TRILEX] method is exact in the large-U regime, as it is an expansion around an impurity that represents the exact solution in that limit. Indeed, in work mentioned by the Referee [PRB 103, 245123 (2021)], it is shown that the method performs better when the interaction strength is larger than the bandwidth (U=10-12).” As a matter of fact, Fig. 3 of PRB 103, 245123 (2021) clearly demonstrates that the accuracy of D-TRILEX method is worst in the regime between U=6 and U=8 and improves at the interaction strengths starting from U=8. >>

Second, even if this is an expansion around the strong coupling limit, it really does not imply that the method would yield accurate results at the two-particle level at strong coupling. In fact, the papers I have suggested to be referenced address this topic thoroughly: DIFFERENT strong coupling expansions can yield DIFFERENT values of the Neel critical temperature in 3D and sometimes completely destroy antiferromagnetism (AFM). The right approximation schemes are the one that stabilize AFM starting from the atomic limit (where AFM is absent) and yield the right critical temperature. A natural question that arises at this point, would D-Trilex yield the correct value of T-Neel at strong coupling in 3D? Is out there a reference showing this?

<<A: It is not surprising that different approximations lead to different results for the Néel critical temperature (or for any other quantity of interest) in some intermediate regime between the weak-coupling and the atomic limit. Therefore, we do not fully understand what the Referee means with “the right Néel temperature” and “right approximation schemes”. If we use DMFT as a reference point, we are guaranteed that our expansion will work in some vicinity of the atomic limit. The same holds true for the other choices of the interacting reference problem that are exact in the atomic limit. As any approximation can in principle give a different value for the Néel temperature, we think that comparing D-TRILEX results for the susceptibility with non-exact results presented in the two works [PRB 99, 235106 (2019) and PRB 104, 235128 (2021)] mentioned by the Referee does not add any value to our work. Indeed, in these two papers the Néel temperatures are obtained by calculating the DMFT-like susceptibilities. However, in the 3D case considered there, the DMFT susceptibility does not correspond by any means to the exact susceptibility of the problem. Additionally, we would like to stress again that no approximation for the three-point vertex functions is utilized in D-TRILEX, contrarily to what is done in the works [PRB 99, 235106 (2019) and PRB 104, 235128 (2021)].

That said, we would like to provide some arguments on why D-TRILEX provides reasonably accurate results also for two-particle quantities. First of all, in our approach, single- and two- particle quantities are obtained self-consistently from the same functional introduced in Eq. (75). As a consequence, one can expect the accuracy of the D-TRILEX method to be comparable between the single- and two-particle levels. Secondly, as explicitly demonstrated in Ref. [PRB 103, 245123 (2021)], the D‑TRILEX diagrammatic expansion effectively takes into account only longitudinal fluctuations in the two-particle Bethe-Salpeter equation (BSE) considered in dual theories and also when calculating the DMFT susceptibility (see Fig. 1 in Ref. [PRB 103, 245123 (2021)]). These longitudinal fluctuations represent the leading contribution to BSE as we explicitly demonstrated in Ref. [PRB 103, 245123 (2021)] by means of exact diagrammatic Monte Carlo calculations. Therefore, D-TRILEX should be in a good agreement with the other dual theories and the DMFT result for the susceptibility when non-longitudinal fluctuations can be neglected.

Based on the same argument, we expect our method to give accurate results for the Néel temperature for the single-band Hubbard model on a cubic lattice. In order to support this claim, we performed calculations for the Néel temperature for the half-filled single-band Hubbard model on a cubic lattice at t=1 and U=8. We found a Néel temperature TNéel = 0.33, which is in very good agreement with exact results obtained using Determinant Diagrammatic Monte Carlo (DDMC) and Quantum Monte carlo (QMC) approaches, as well as using the ladder dual fermion method. All these results are nicely collected in a recent work on DiagMC [arXiv:2112.15209 (2021)]. A more thorough analysis of the Néel transition in a cubic lattice will be addressed in a separate work and does not fit within our discussion of the method beyond the single-band Hubbard model. >>

Also, even if D-Trilex does not use the four-point vertex function, DB technique does, therefore having reliable approximate schemes to calculate the four-point vertex function would reduce the computational complexity for numerical calculation of strongly correlated systems in regime where D-Trilex does not work. I think these are subtle and important aspects that the authors missed and could leave for future work, but I truly believe that mentioning them in the discussion section and adding the suggested references would enrich their work.

<<A: The current work does not address the DB theory, except as an intermediate step to derive the D-TRILEX method. As a consequence, we believe that the discussion of approximations for the calculation of the four-point vertex is not strictly related with the content of our work. Going beyond the simple D-TRILEX diagrammatics would indeed necessarily involve dealing with the four-point vertex functions, which will unavoidably ruin the simplicity of the computational scheme proposed in our work. However, a detailed discussion of the multi-band DB theory and/or much more complex extensions of D-TRILEX theory is beyond the scope of this work. >>

---

## Round 3 · List of Changes

1. We clarified the meaning of the Rashba form of the SOC in the introduction.
  2. We introduced two new panels in Fig.4, showing the local and non-local self-energy as a function of interaction and frequency.
  3. We added a description of the new content of Fig.4 in Sec. 5.1.

---

## Editorial Decision

published